# Distributed Stochastic Algorithms for High-rate Streaming Principal Component Analysis

**Haroon Raja**  *haroon.raja@rutgers.edu*
*Department of Electrical and Computer Engineering*
*Rutgers University–New Brunswick, Piscataway, NJ 08854 USA*

**Waheed U. Bajwa**  *waheed.bajwa@rutgers.edu*
*Department of Electrical and Computer Engineering*
*Department of Statistics*
*Rutgers University–New Brunswick, Piscataway, NJ 08854 USA*

**Reviewed on OpenReview:** *https: // openreview. net/ forum? id= CExeDOjpB6*

## Abstract

This paper considers the problem of estimating the principal eigenvector of a covariance matrix from independent and identically distributed data samples in streaming settings. The streaming rate of data in many contemporary applications can be high enough that a single processor cannot finish an iteration of existing methods for eigenvector estimation before a new sample arrives. This paper formulates and analyzes a distributed variant of the classical Krasulina's method (D-Krasulina) that can keep up with the high streaming rate of data by distributing the computational load across multiple processing nodes. The analysis improves upon the one in (Balsubramani et al., 2013) for the original Krasulina's method and shows that—under appropriate conditions—D-Krasulina converges to the principal eigenvector in an order-wise optimal manner; i.e., after receiving $M$ samples across all nodes, its estimation error can be $O(1/M)$. In order to reduce the network communication overhead, the paper also develops and analyzes a mini-batch extension of D-Krasulina, which is termed DM-Krasulina. The analysis of DM-Krasulina shows that it can also achieve order-optimal estimation error rates under appropriate conditions, even when some samples have to be discarded within the network due to communication latency. Finally, experiments are performed over synthetic and real-world data to validate the convergence behaviors of D-Krasulina and DM-Krasulina in high-rate streaming settings.

## 1 Introduction

Dimensionality reduction and feature learning methods such as *principal component analysis* (PCA), sparse PCA, independent component analysis, and autoencoder form an important component of any machine learning pipeline. For data lying in a $d$-dimensional space, such methods try to find the $k \ll d$ variables/features that are most relevant for solving an application-specific task (e.g., classification, regression, estimation, data compression, etc.). The focus of this work is on PCA, where the objective is to compute $k$-features that capture most of the variance in data. The proliferation of *big data* (both in terms of dimensionality and number of samples) has resulted in an increased interest in developing new algorithms for PCA due to the fact that classical numerical solutions (e.g., power iteration and Lanczos method (Golub & Van Loan, 2012)) for computing eigenvectors of symmetric matrices do not scale well with high dimensionality and large sample sizes. The main interest in this regard has been on developing algorithms that are cheap in terms of both memory and computational requirements as a function of dimensionality and number of data samples.

In addition to high dimensionality and large number of samples, another defining characteristic of modern data is their streaming nature in many applications; examples of such applications include the internet-of-things, high-frequency trading, meteorology, video surveillance, autonomous vehicles, social media analytics,

etc. Several stochastic methods have been developed in the literature to solve the PCA problem in streaming settings (Krasulina, 1969; Oja & Karhunen, 1985; Sanger, 1989; Warmuth & Kuzmin, 2007; Zhang & Balzano, 2016). These methods operate under the *implicit* assumption that the data arrival rate is slow enough so that each sample can be processed before the arrival of the next one. But this may not be true for many modern applications involving high-rate streaming data. To overcome this obstacle corresponding to high-rate streaming data, this paper proposes and analyzes distributed and distributed, mini-batch variants of the classical Krasulina's method (Krasulina, 1969). Before providing details of the proposed methods and their relationship to prior work, we provide a brief overview of the streaming PCA problem.

## 1.1  Principal Component Analysis (PCA) from Streaming Data

For data lying in $\mathbb{R}^d$, PCA learns a $k$-dimensional subspace with maximum data variance. Let $\mathbf{x} \in \mathbb{R}^d$ be a random vector that is drawn from some unknown distribution $\mathcal{P}_x$ with zero mean and $\mathbf{\Sigma}$ covariance matrix. For the constraint set $\mathcal{V} := \{\mathbf{V} \in \mathbb{R}^{d \times k} : \mathbf{V}^\mathrm{T}\mathbf{V} = \mathbf{I}\}$, we can pose PCA as the following constrained optimization problem:

$$\mathbf{Q}^* := \arg\max_{\mathbf{V} \in \mathcal{V}} \mathbb{E}_{\mathcal{P}_x}\Big\{ \mathsf{Tr}(\mathbf{V}^\mathrm{T}\mathbf{x}\mathbf{x}^\mathrm{T}\mathbf{V}) \Big\}, \tag{1}$$

where $\mathsf{Tr}(.)$ denotes the trace operator. The solution for the *statistical risk maximization* problem (1) is the matrix $\mathbf{Q}^*$ with top $k$ eigenvectors of $\mathbf{\Sigma}$. In practice, however, (1) cannot be solved in its current form since $\mathcal{P}_x$ is unknown. But if we have $T$ data samples, $\{\mathbf{x}_t\}_{t=1}^T$, drawn independently from $\mathcal{P}_x$, then we can accumulate these data samples to calculate the sample covariance matrix as:

$$\bar{\mathbf{A}}_T := \frac{1}{T} \sum_{t=1}^T \mathbf{A}_t, \tag{2}$$

where $\mathbf{A}_t := \mathbf{x}_t\mathbf{x}_t^\mathrm{T}$. Instead of solving (1), we can now solve an *empirical risk maximization* problem

$$\mathbf{Q} := \arg\max_{\mathbf{V} \in \mathcal{V}} \mathsf{Tr}(\mathbf{V}^\mathrm{T}\bar{\mathbf{A}}_T\mathbf{V}) = \arg\max_{\mathbf{V} \in \mathcal{V}} \frac{1}{T} \sum_{t=1}^T \mathsf{Tr}(\mathbf{V}^\mathrm{T}\mathbf{A}_t\mathbf{V}). \tag{3}$$

In principle, we can solve (3) by computing the *singular value decomposition* (SVD) of sample covariance $\bar{\mathbf{A}}_T$. But this is a computationally intensive task that requires $O(d^3)$ multiplications and that has a memory overhead of $O(d^2)$. In contrast, the goal in high-dimensional PCA problems is often to have $O(d^2k)$ computational complexity and $O(dk)$ memory complexity (Li et al., 2016).

More efficient (and hence popular) approaches for PCA use methods such as the power/orthogonal iteration and Lanczos method (Golub & Van Loan, 2012, Chapter 8). Although these methods improve overall computational complexity of PCA to $O(d^2k)$, they still have memory requirements on the order of $O(d^2)$. In addition, these are *batch* methods that require computing the sample covariance matrix $\bar{\mathbf{A}}_T$, which results in $O(d^2T)$ multiplication operations. Further, in streaming settings where the goal is real-time decision making from data, it is infeasible to compute $\bar{\mathbf{A}}_T$. Because of these reasons, stochastic approximation methods such as Krasulina's method (Krasulina, 1969) and Oja's rule (Oja & Karhunen, 1985) are often favored for the PCA problem. Both these are simple and extremely efficient algorithms, achieving $O(d)$ computational and memory complexity per iteration, for computing the principal eigenvector (i.e., $k = 1$) of a covariance matrix in streaming settings. Recent years in particular have seen an increased popularity of these algorithms and we will discuss these recent advances in Section 1.3.

Both Oja's rule and Krasulina's method share many similarities. In this paper, we focus on Krasulina's method with the understanding that our findings can be mapped to Oja's rule through some tedious but straightforward calculations. Using $t$ for algorithmic iteration, Krasulina's method estimates the top eigenvector by processing one data sample in each iteration as follows:[1]

$$\mathbf{v}_t = \mathbf{v}_{t-1} + \gamma_t \left( \mathbf{x}_t\mathbf{x}_t^\mathrm{T}\mathbf{v}_{t-1} - \frac{\mathbf{v}_{t-1}^\mathrm{T}\mathbf{x}_t\mathbf{x}_t^\mathrm{T}\mathbf{v}_{t-1}\mathbf{v}_{t-1}}{\|\mathbf{v}_{t-1}\|_2^2} \right), \tag{4}$$

---

[1]In contrast, the iterate of Oja's rule is given by $\mathbf{v}_t = \mathbf{v}_{t-1} + \gamma_t \left( \mathbf{x}_t\mathbf{x}_t^\mathrm{T}\mathbf{v}_{t-1} - \mathbf{v}_{t-1}^\mathrm{T}\mathbf{x}_t\mathbf{x}_t^\mathrm{T}\mathbf{v}_{t-1}\mathbf{v}_{t-1} \right).$

where $\gamma_t$ denotes the step size. Going forward, we will be using $\mathbf{A}_t$ in place of $\mathbf{x}_t\mathbf{x}_t^{\mathrm{T}}$ in expressions such as (4) for notational compactness. In practice, however, one should neither explicitly store $\mathbf{A}_t$ nor explicitly use it for calculation purposes.

Note that one can interpret Krasulina's method as a solution to an optimization problem. Using Courant–Fischer Minimax Theorem (Golub & Van Loan, 2012, Theorem 8.1.2), the top eigenvector computation (i.e., *1-PCA*, which is the $k = 1$ version of (1)) can be posed as the following optimization problem:

$$\mathbf{q}_1 := \arg\min_{\mathbf{v}\in\mathbb{R}^d} f(\mathbf{v}) = \arg\min_{\mathbf{v}\in\mathbb{R}^d} \frac{-\mathbf{v}^{\mathrm{T}}\mathbf{A}_t\mathbf{v}}{\|\mathbf{v}\|_2^2}. \tag{5}$$

In addition, the gradient of the function $f(\mathbf{v})$ defined in (5) is:

$$\nabla f(\mathbf{v}) = \frac{1}{\|\mathbf{v}\|_2^2}\left(-\mathbf{A}_t\mathbf{v} + \frac{(\mathbf{v}^{\mathrm{T}}\mathbf{A}_t\mathbf{v})\mathbf{v}}{\|\mathbf{v}\|_2^2}\right). \tag{6}$$

Looking at (4)–(6), we see that (4) is very similar to applying *stochastic gradient descent* (SGD) to the nonconvex problem (5), with the only difference being the scaling factor of $1/\|\mathbf{v}\|_2^2$. Nonetheless, since (5) is a nonconvex problem and we are interested in global convergence behavior of Krasulina's method, existing tools for analysis of the standard SGD problem (Bottou, 2010; Recht et al., 2011; Dekel et al., 2012; Reddi et al., 2016b;c) do not lend themselves to the fastest convergence rates for Krasulina's method. Despite its nonconvexity, however, (5) has a somewhat benign optimization landscape and a whole host of algorithmic techniques and analytical tools have been developed for such structured nonconvex problems in recent years that guarantee fast convergence to a global solution. In this paper, we leverage some of these recent developments to guarantee near-optimal global convergence of two variants of Krasulina's method in the case of high-rate streaming data.

Before proceeding further, it is worth noting that while Krasulina's method primarily focuses on the 1-PCA problem, it *can* be used to solve the $k$-PCA problem. But such an *indirect* approach, which involves repeated use of the Krasulina's method $k$ times, can be inefficient in terms of sample complexity (Allen-Zhu & Li, 2017a, Section 1). We leave investigation of a near-optimal direct method for the $k$-PCA problem involving high-rate streaming data for future work.

## 1.2 Our Contributions

In this paper, we propose and analyze two distributed variants of Krasulina's method for estimating the top eigenvector of a covariance matrix from fast streaming, independent and identically distributed (i.i.d.) data samples. Our theoretical analysis, as well as numerical experiments on synthetic and real data, establish near-optimality of the proposed algorithms. In particular, our analysis shows that the proposed algorithms can achieve the optimal convergence rate of $O(1/M)$ for 1-PCA after processing a total of $O(M)$ data samples (see (Jain et al., 2016, Theorem 1.1) and (Allen-Zhu & Li, 2017a, Theorem 6)). In terms of details, following are our key contributions:

1. Our first contribution corresponds to the scenario in which there is a mismatch of $N \in \mathbb{Z}_+ > 1$ between the data streaming rate and the processing capability of a single processor, i.e., one iteration of Krasulina's method on one processor takes as long as $N$ data arrival epochs. Our solution to this problem, which avoids discarding of samples, involves splitting the data stream into $N$ parallel streams that are then input to $N$ interconnected processors. Note that this splitting effectively reduces the streaming rate at each processor by a factor of $N$. We then propose and analyze a distributed variant of Krasulina's method—termed D-Krasulina—that solves the 1-PCA problem for this distributed setup consisting of $N$ processing nodes. Our analysis substantially improves the one in (Balsubramani et al., 2013) for Krasulina's method and shows that D-Krasulina can result in an improved convergence rate of $O(1/Nt)$ after $t$ iterations (Theorem 1), as opposed to the $O(1/t)$ rate for the classical Krasulina's method at any one of the nodes seen in isolation. Establishing this result involves a novel analysis of Krasulina's method that brings out the dependence of its convergence rate on the variance of the sample covariance matrix; this analysis coupled with a variance reduction argument leads to the convergence rate of $O(1/Nt)$ for D-Krasulina under appropriate conditions.

2. Mini-batching of data samples has long been promoted as a strategy in stochastic methods to reduce the wall-clock time. Too large of a mini-batch, however, can have an adverse effect on the algorithmic performance; see, e.g., (Shamir & Srebro, 2014, Sec. VIII). One of the challenges in mini-batched stochastic methods, therefore, is characterizing the mini-batch size that leads to near-optimal convergence rates in terms of the number of processed samples. In (Agarwal & Duchi, 2011; Cotter et al., 2011; Dekel et al., 2012; Shamir & Srebro, 2014; Ruder, 2016; Golmant et al., 2018; Goyal et al., 2017), for example, the authors have focused on this challenge for the case of mini-batch SGD for convex and nonconvex problems. In the case of nonconvex problems, however, the guarantees only hold for convergence to first-order stationary points. In contrast, our second contribution is providing a comprehensive understanding of the global convergence behavior of mini-batched Krasulina's method. In fact, our analysis of D-Krasulina is equivalent to that of a mini-batch (centralized) Krasulina's method that uses a mini-batch of $N$ samples in each iteration. This analysis, therefore, guarantees near-optimal convergence rate with arbitrarily high probability for an appropriately mini-batcheded Krasulina's method in a centralized setting. This is in contrast to (Jain et al., 2016; Yang et al., 2018), where the focus is on Oja's rule and the probability of success is upper bounded by $3/4$ for a single algorithmic run.[2] In addition, in the case of high-rate streaming data that requires splitting the data stream into $N$ parallel ones, we characterize the *global* convergence behavior of a mini-batch generalization of D-Krasulina—termed DM-Krasulina—in terms of the mini-batch size. This involves specifying the conditions under which mini-batches of size $B/N$ per node can lead to near-optimal convergence rate of $O(1/Bt)$ after $t$ iterations of DM-Krasulina (Theorem 2). An implication of this analysis is that for a fixed (network-wide) sample budget of $T$ samples, DM-Krasulina can achieve $O(1/T)$ rate after $t := T/B$ iterations provided the (network-wide) mini-batch size $B$ satisfies $B = O(T^{1-\frac{2}{c_0}})$ for some constant $c_0 > 2$ (Corollary 1).

3. Our next contribution is an extended analysis of DM-Krasulina that concerns the scenario where (computational and/or communication) resource constraints translate into individual nodes still receiving more data samples than they can process in one iteration of DM-Krasulina. This resource-constrained setting necessitates DM-Krasulina dropping $\mu \in \mathbb{Z}_+$ samples across the network in each iteration. Our analysis in this setting shows that such loss of samples need not result in sub-optimal performance. In particular, DM-Krasulina can still achieve near-optimal convergence rate as a function of the number of samples arriving in the network—for both infinite-sample and finite-sample regimes—as long as $\mu = O(B)$ (Corollary 2).

4. We provide numerical results involving both synthetic and real-world data to establish the usefulness of the proposed algorithms, validate our theoretical analysis, and understand the impact of the number of dropped samples per iteration of DM-Krasulina on the convergence rate. These results in particular corroborate our findings that increasing the mini-batch size improves the performance of DM-Krasulina up to a certain point, after which the convergence rate starts to decrease. Since the focus of this work is on systems theory, the reported results do not focus on some of the large-scale system implementation issues such as unexpected processor failures, high network coordination costs, etc. Such large-scale implementation issues, while relevant from a practical perspective, are beyond the scope of this paper and provide interesting research directions for future work.

## 1.3 Related Work

Solving the PCA problem efficiently in a number of settings has been an active area of research for decades. (Krasulina, 1969; Oja & Karhunen, 1985) are among the earliest and most popular methods to solve PCA in streaming data settings. Several variants of these methods have been proposed over the years, including (Bin Yang, 1995; Chatterjee, 2005; Doukopoulos & Moustakides, 2008). Like earlier developments in

---

[2]Note that it is possible to improve the probability of success for Oja's rule, as derived in (Jain et al., 2016; Yang et al., 2018), to $1-\delta$, $\delta \in (0,1)$, by running $O(\log(1/\delta))$ instances of the algorithm and combining their outcomes. But such a strategy, which also adds to the computational and storage overhead because of the 'combine' step, is impractical for the streaming settings considered in this paper. Additional differences between the results of this paper pertaining to the centralized PCA problem for streaming data and that of (Jain et al., 2016; Yang et al., 2018) are discussed in Section 1.3. It is, however, important to point out that this work and (Jain et al., 2016; Yang et al., 2018) are analyzing two related, *yet different*, algorithmic approaches that are based on Krasulina's method and Oja's rule, respectively, and their results are therefore complementary in nature.

stochastic approximation methods (Robbins & Monro, 1951), such variants were typically shown to converge asymptotically. Convergence rate analyses for stochastic optimization in finite-sample settings (Shapiro & Homem-de Mello, 2000; Linderoth et al., 2006) paved the way for non-asymptotic convergence analysis of different variants of the stochastic PCA problem, which is fundamentally a nonconvex optimization problem. Within the context of such works, the results that are the most relevant to the algorithmic strategy devised in this paper can be found in (Jain et al., 2016) and (Yang et al., 2018). The authors in these two papers provide variance-dependent convergence guarantees for Oja's rule in the finite-sample regime, thereby making their results also translatable to the algorithmic framework being considered in this paper for high-rate streaming PCA. However, as noted earlier, the results derived in (Jain et al., 2016; Yang et al., 2018) only hold with probability 3/4, which is in contrast to the high-probability results of this paper. And while one could increase the probability of success in (Jain et al., 2016; Yang et al., 2018) through multiple algorithmic runs, this is not a feasible strategy in the streaming settings.

In order to complement the pioneering results of (Jain et al., 2016; Yang et al., 2018) in streaming settings, we shift our focus away from Oja's rule as the base algorithm and develop a different proof strategy that substantially extends and generalizes the analysis in (Balsubramani et al., 2013) for Krasulina's method. The analysis in (Balsubramani et al., 2013) assumes that the $\ell_2$ norm of the data samples is bounded by a positive constant, but it does not take into account the variance of the sample covariance matrices. Such an analysis leads to convergence results that are independent of the variance and hence are unable to capture any improvement in the convergence rate due to mini-batching and/or distributed implementations for computing the top eigenvector of a covariance matrix. We overcome this limitation of the analysis in (Balsubramani et al., 2013) by developing a proof in this work that explicitly accounts for the variance of the sample covariance matrices. Note that this extension/generalization of the analysis in (Balsubramani et al., 2013)—despite the intuitive nature of our final set of results—is a nontrivial task. There are in particular two main technical challenges that are addressed in this paper: (*i*) Using concentration of measure results that allow for incorporation of the variance within the analysis, as opposed to the Hoeffding inequality (Boucheron et al., 2013) utilized in (Balsubramani et al., 2013); and (*ii*) Utilizing the variance-dependent concentration guarantees within two terms in Lemma 1, one of which depends on the norm bound on data samples and the other on the variance—a careful decoupling of these two quantities being critical to obtain convergence results in which the dominant term depends only on the variance. Finally, another aspect that distinguishes this paper from prior works such as (Balsubramani et al., 2013; Jain et al., 2016; Yang et al., 2018) is that it provides a formal framework for studying the communications and computation tradeoffs involved in solving the 1-PCA problem in distributed streaming settings. This framework is described in detail in Section 3.2, with theoretical characterization of the proposed framework in terms of the communications and computation costs described in Section 4.2.

Because of the vastness of literature on (stochastic) PCA, this work is also tangentially or directly related to a number of additional prior works. We review some of these works in the following under the umbrellas of different problem setups, with the understanding that the resulting lists of works are necessarily incomplete. Much of our discussion in the following focuses on solving the PCA problem in (fast) streaming and distributed data settings, which is the main theme in this paper.

**Sketching for PCA.** Sketching methods have long been studied in the literature for solving problems involving matrix computations; see (Woodruff, 2014) for a review of such methods. The main idea behind these methods is to compress data using either randomized or deterministic sketches and then perform computations on the resulting low-dimensional data. While sketching has been used as a tool to solve the PCA problem in an efficient manner (see, e.g., (Warmuth & Kuzmin, 2007; Halko et al., 2011; Liberty, 2013; Leng et al., 2015; Karnin & Liberty, 2015)), the resulting methods cannot be used to *exactly* solve (1) in the fast streaming settings of this paper.

**Online PCA.** The PCA problem has also been extensively studied in online settings. While such settings also involve streaming data, the main goal in online PCA is to minimize the cumulative subspace estimation error over the entire time horizon of the algorithm. The online PCA framework, therefore, is especially useful in situations where either the underlying subspace changes over time or there is some adversarial noise in the sampling process. Some of the recent works in this direction include (Garber et al., 2015; Allen-Zhu & Li, 2017b; Garber, 2018; Marinov et al., 2018; Kotłowski & Neu, 2019).

**Stochastic convex optimization for PCA.** One approach towards solving (3) in streaming settings is to relax the PCA problem to a convex optimization problem and then use SGD to solve the resulting stochastic convex optimization problem (Arora et al., 2013; Garber & Hazan, 2015; Nie et al., 2016). The benefit of this approach is that now one can rely on rich literature for solving stochastic convex problems using SGD. But the tradeoff is that one now needs to store an iterate of dimension $\mathbb{R}^{d \times d}$, as opposed to an iterate of dimension $\mathbb{R}^{d \times k}$ when we solve the PCA problem in its original nonconvex form. Due to these high memory requirements of $O(d^2)$, we limit ourselves to solving PCA in the nonconvex form.

**Streaming PCA and nonconvex optimization.** The PCA problem in the presence of streaming data can also be tackled as an explicit constrained nonconvex optimization program (Zhang & Balzano, 2016; De Sa et al., 2015). In (Zhang & Balzano, 2016), for instance, the problem is solved as an optimization program over the Grassmannian manifold. The resulting analysis, however, relies on the availability of a good initial guess. In contrast, the authors in (De Sa et al., 2015) analyze the use of the SGD for solving certain nonconvex problems that include PCA. The resulting approach, however, requires the step size to be a significantly small constant for eventual convergence (e.g., $10^{-12}$ for the Netflix Prize dataset); this translates into slower convergence in practice.

**Classical stochastic approximation methods for PCA.** Recent years have seen an increased interest in understanding the global convergence behavior of classical stochastic approximation methods such as Krasulina's method (Krasulina, 1969) and Oja's rule (Oja & Karhunen, 1985) for the PCA problem in non-asymptotic settings (Allen-Zhu & Li, 2017a; Chatterjee, 2005; Hardt & Price, 2014; Shamir, 2015; 2016; Jain et al., 2016; Li et al., 2016; Tang, 2019; Henriksen & Ward, 2019; Amid & Warmuth, 2019). Some of these works, such as (Shamir, 2015) and (Shamir, 2016), use variance reduction techniques to speed-up the algorithmic convergence. Such works, however, require multiple passes over the data, which makes them ill-suited for fast streaming settings. The analysis in (Shamir, 2015) and (Shamir, 2016) also requires an initialization close to the true subspace, which is somewhat unlikely in practice. Among other works, the authors in (Allen-Zhu & Li, 2017a) provide eigengap-free convergence guarantees for Oja's rule. Since the results in this work do not take into account the variance of data samples, they do not generalize to mini-batch/distributed streaming settings. As stated earlier, the authors in (Jain et al., 2016) do provide variance-dependent convergence guarantees for Oja's rule, which makes this work the most relevant to ours. In particular, the authors in (Yang et al., 2018) have extended the initial analysis in (Jain et al., 2016) to mini-batch settings. But a limitation of the analysis in (Jain et al., 2016; Yang et al., 2018) is that it guarantees convergence of Oja's rule only up to a probability of 3/4. And while the probability of success can be increased to $1 - \delta$ by running and combining the outcomes of $O(\log(1/\delta))$ instances of Oja's rule, such an approach has two major drawbacks in streaming settings. First, since new data samples arrive continuously in a streaming setting, multiple runs of an algorithm in this case can only be achieved through multiple replicas of the processing system. Such a strategy, therefore, leads to a substantial increase in system costs. Second, the outcomes of the multiple runs need to be appropriately combined. In (Jain et al., 2016), it is suggested this be done by computing the geometric median of the multiple outcomes, which requires solving an additional optimization problem. This then adds to the computational and storage overhead for the PCA problem. We conclude by remarking on two key distinctions between our results and those in (Jain et al., 2016; Yang et al., 2018). First, the arbitrarily high probability of success in our analysis requires the initial step size in Krasulina's method to decrease with an increase in the probability; we refer the reader to the discussion following Theorem 1 in this paper for further details on this point. Second, our convergence guarantees have the flavor of 'convergence in mean' as opposed to the 'convergence in probability' nature of the results in (Jain et al., 2016; Yang et al., 2018). A straightforward application of Markov's inequality, however, leads to results that are directly comparable to the ones in (Jain et al., 2016; Yang et al., 2018); we refer the reader to Corollary 3 in Appendix D as an illustrative example of this.

**Distributed PCA and streaming data.** Several recent works such as (Balcan et al., 2016; Boutsidis et al., 2016; Garber et al., 2017; De Sa et al., 2018) have focused on the PCA problem in distributed settings. Among these works, the main focus in (Balcan et al., 2016; Boutsidis et al., 2016; Garber et al., 2017) is on improving the communications efficiency. This is accomplished in (Balcan et al., 2016; Boutsidis et al., 2016) by sketching the local iterates and communicating the resulting compressed iterates to a central server in each iteration. In contrast, (Garber et al., 2017) provides a batch solution in which every node

in the network first computes the top eigenvector of its local (batch) covariance matrix and then, as a last step of the algorithm, all the local eigenvector estimates are summed up at a central server to provide an eigenvector estimate for the global covariance matrix. In contrast to these works, our focus in this paper is on establishing that distributed (mini-batch) variants of stochastic approximation methods such as Oja's rule and Krasulina's method can lead to improved convergence rates, as a function of the number of samples, for the PCA problem in fast streaming settings. In this regard, our work is more closely related to (De Sa et al., 2018), where the authors use the momentum method to accelerate convergence of power method and further extend their work to stochastic settings. However, the approach of (De Sa et al., 2018) relies on a variance reduction technique that requires a pass over the complete dataset every once in a while; this is impractical in streaming settings. In addition, theoretical guarantees in (De Sa et al., 2018) are based on the assumption of a "good" initialization; further, an implicit assumption in (De Sa et al., 2018) is that inter-node communications is fast enough that there are no communication delays.

**Decentralized PCA.** Another important practical setting under which the PCA problem has been studied is when the data are distributed across an interconnected set of nodes that form a complete graph but not a fully connected one. We refer the reader to (Blondel et al., 2005; Aysal et al., 2009; Khan et al., 2009; Dimakis et al., 2010) for a general understanding of this decentralized setting. Some contributions to the PCA problem in this setting are (Kempe et al., 2008; Li et al., 2011; Korada et al., 2011; Wu et al., 2017; 2018; Gang et al., 2021; Gang & Bajwa, 2021; 2022). Most of these contributions consider batch data settings and their extensions to the streaming setting are not evident. And while contributions such as (Li et al., 2011) do consider the streaming data setting, the convergence rates established in such works are asymptotic in nature and the theoretical analyses cannot be used to derive conditions for a linear speed-up in convergence rates in the mini-batch setting.

**Connections to low-rank matrix approximation.** The *low-rank matrix approximation* problem (Frieze et al., 2004; Clarkson & Woodruff, 2017), which involves computing a low-rank approximation of a given matrix, is closely related to the PCA problem. The overarching goal in both problems is the same: *find a subspace that best approximates the data samples / given matrix.* In the case of PCA, however, the focus is fundamentally on finding an orthogonal basis for the subspace that is precisely given by the top eigenvectors of the data covariance matrix. Notwithstanding this difference between the two problems, (Yun et al., 2015; Tropp et al., 2019) are two works within the low-rank matrix approximation literature that are the most related to this paper. The setting in (Yun et al., 2015) corresponds to a large-scale but fixed matrix whose columns are presented to a low-rank approximation algorithm in a streaming manner. This is in contrast to the streaming setting of this work that is akin to having a matrix with infinitely many columns. In addition, the algorithm being studied in (Yun et al., 2015) requires computing the top principal component directions of a random submatrix of the larger matrix in a batch setting as its first step, which is again a departure from the streaming data setting of this work. Next, the mathematical model in (Tropp et al., 2019) corresponds to a matrix that is given by the sum of a long sequence of low-rank and sparse matrices that are presented to a low-rank approximation algorithm in a streaming manner. This summation aspect of the data model in (Tropp et al., 2019) does not coincide with the mathematical model of the streaming data samples in this work. Finally, and in stark contrast to this paper, neither of these works are concerned with the interplay between the streaming data rate, data processing rate, and the number of interconnected processors.

**Connections to stochastic nonconvex optimization.** Recent years have also seen an increased focus on understanding (variants of) SGD for general (typically unconstrained) stochastic nonconvex optimization problems. Among such works, some have focused on classical SGD (Ge et al., 2015; Hazan et al., 2016; 2017; Li et al., 2021; Mokhtari et al., 2017), some have studied variance-reduction variants of SGD (Reddi et al., 2016a;b; Qureshi et al., 2021), and some have investigated accelerated variants of stochastic nonconvex optimization (Allen-Zhu, 2018b;a). In particular, works such as (Reddi et al., 2016a; Allen-Zhu & Hazan, 2016) are directly relevant to this paper since these works also use mini-batches to reduce sample variance and improve on SGD convergence rates. While (implicit, through the distributed framework, and explicit) mini-batching is one of the key ingredients of our work also, this paper differs from such related works because of its ability to prove convergence to a global optimum of the 1-PCA problem. In contrast, aforementioned works only provide guarantees for convergence to first-order stationary points of (typically unconstrained) stochastic nonconvex optimization problems.

### 1.4 Notational Convention and Paper Organization

We use lower-case ($a$), bold-faced lower-case ($\mathbf{a}$), and bold-faced upper-case ($\mathbf{A}$) letters to represent scalars, vectors, and matrices, respectively. Given a scalar $a$ and a vector $\mathbf{a}$, $\lceil a \rceil$ denotes the smallest integer greater than or equal to $a$, while $\|\mathbf{a}\|_2$ denotes the $\ell_2$-norm of $\mathbf{a}$. Given a matrix $\mathbf{A}$, $\|\mathbf{A}\|_2$ denotes its spectral norm and $\|\mathbf{A}\|_F$ denotes its Frobenius norm. In addition, assuming $\mathbf{A} \in \mathbb{R}^{d \times d}$ to be a positive semi-definite matrix, $\lambda_i(\mathbf{A})$ denotes its $i$-th largest eigenvalue, i.e., $\|\mathbf{A}\|_2 := \lambda_1(\mathbf{A}) \geq \lambda_2(\mathbf{A}) \geq \cdots \geq \lambda_d(\mathbf{A}) \geq 0$. Whenever obvious from the context, we drop $\mathbf{A}$ from $\lambda_i(\mathbf{A})$ for notational compactness. Finally, $\mathbb{E}\{\cdot\}$ denotes the expectation operator, where the underlying probability space $(\Omega, \mathcal{F}, \mathbb{P})$ is either implicit from the context or is explicitly pointed out in the body.

The rest of this paper is organized as follows. We first provide a formal description of the problem and the system model in Section 2. The two proposed variants of Krasulina's method that can be used to solve the 1-PCA problem in fast streaming settings are then presented in Section 3. In Section 4, we provide theoretical guarantees for the proposed algorithms, while proofs / outlines of the proofs of the main theoretical results are provided in Section 5. Finally, numerical results using both synthetic and real-world data are presented in Section 6, while appendices are used for detailed proofs of some of the theoretical results.

## 2 Problem Formulation and System Model

Our goal is to use some variants of Krasulina's method (cf. (4)) in order to obtain an estimate of the top eigenvector of a covariance matrix from independent and identically distributed (i.i.d.) data samples that are fast streaming into a system. The algorithms proposed in this regard and their convergence analysis rely on the following sets of assumptions concerning the data and the system.

### 2.1 Data Model

We consider a streaming data setting where a new data sample $\mathbf{x}_{t'} \in \mathbb{R}^d$ independently drawn from an unknown distribution $\mathcal{P}_x$ arrives at a system at each sampling time instance $t'$. We assume a uniform data arrival rate of $R_s$ samples per second and, without loss of generality, take the data arrival index $t' \geq 1$ to be an integer. We also make the following assumptions concerning our data, which aid in our convergence analysis.

[**A1**] *(Zero-mean, norm-bounded samples)* Without loss of generality, the data samples have zero mean, i.e., $\mathbb{E}_{\mathcal{P}_x}\{\mathbf{x}_{t'}\} = 0$. In addition, the data samples are almost surely bounded in norm, i.e., $\|\mathbf{x}_{t'}\|_2 \leq r$, where we let the bound $r \geq 1$ without loss of generality.

[**A2**] *(Spectral gap of the covariance matrix)* The largest eigenvalue of $\mathbf{\Sigma} := \mathbb{E}_{\mathcal{P}_x}\{\mathbf{x}_{t'}\mathbf{x}_{t'}^{\mathrm{T}}\}$ is strictly greater than the second largest eigenvalue, i.e., $\lambda_1(\mathbf{\Sigma}) > \lambda_2(\mathbf{\Sigma}) \geq \lambda_3(\mathbf{\Sigma}) \geq \cdots \geq \lambda_d(\mathbf{\Sigma}) \geq 0$.

Note that both Assumptions [**A1**] and [**A2**] are standard in the literature for convergence analysis of Krasulina's method and Oja's rule (cf. (Balsubramani et al., 2013; Oja & Karhunen, 1985; Allen-Zhu & Li, 2017a; Jain et al., 2016)).

We also associate with each data sample $\mathbf{x}_{t'}$ a rank-one random matrix $\mathbf{A}_{t'} := \mathbf{x}_{t'}\mathbf{x}_{t'}^{\mathrm{T}}$, which is a trivial unbiased estimate of the population covariance matrix $\mathbf{\Sigma}$. We then define the variance of this unbiased estimate as follows.

**Definition 1** (Variance of sample covariance matrix)**.** *We define the variance of the sample covariance matrix* $\mathbf{A}_{t'} := \mathbf{x}_{t'}\mathbf{x}_{t'}^{\mathrm{T}}$ *as follows:*

$$\sigma^2 := \mathbb{E}_{\mathcal{P}_x}\left\{\left\|\mathbf{A}_{t'} - \mathbf{\Sigma}\right\|_F^2\right\}.$$

Note that all moments of the probability distribution $\mathcal{P}_x$ exist by virtue of the norm boundedness of $\mathbf{x}_{t'}$ (cf. Assumption [**A1**]). The variance $\sigma^2$ of the sample covariance matrix $\mathbf{A}_{t'}$ as defined above, therefore, exists and is finite.

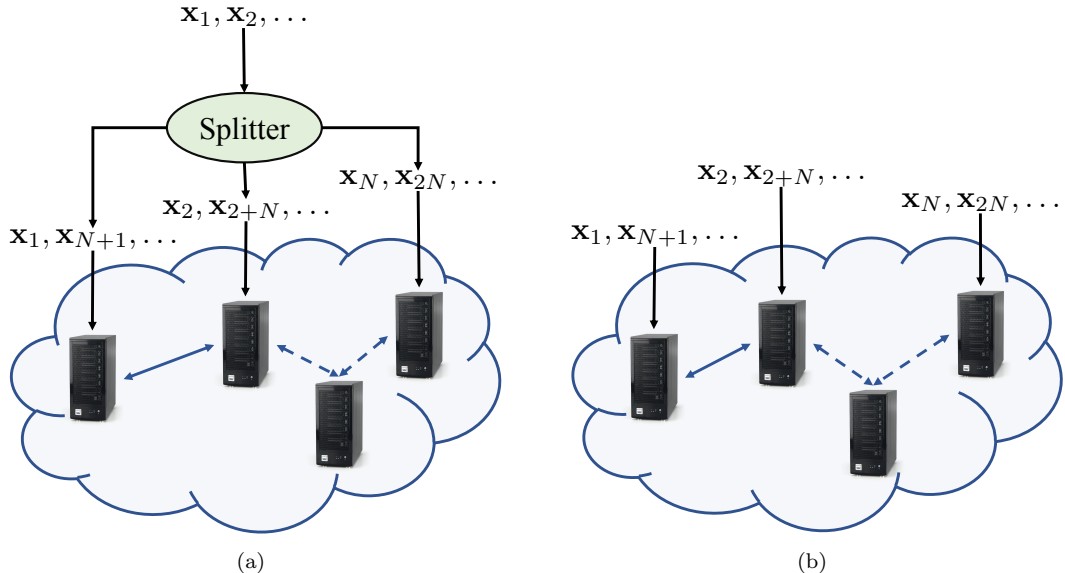

Figure 1: The distributed PCA problem, which involves distributed processing of data over a network of $N$ processors, can arise in two contexts. (a) A data splitter can split a data stream into $N$ parallel streams, one for each processor in the network. In relation to the original data stream, this effectively reduces the data arrival rate for each parallel stream by a factor of $N$. (b) Data can be inherently distributed, as in the Internet-of-Things systems, and can arrive at $N$ different processing nodes as $N$ separate data streams.

The two algorithms proposed in this paper, namely, D-Krasulina and DM-Krasulina, are initialized with a random vector $\mathbf{v}_0 \in \mathbb{R}^d$ that is randomly generated over the unit sphere in $\mathbb{R}^d$ with respect to the uniform (Haar) measure. All analysis in this paper is with respect to the natural probability space $(\Omega, \mathcal{F}, \mathbb{P})$ given by the stochastic process $(\mathbf{v}_0, \mathbf{x}_1, \mathbf{x}_2, \dots)$ and filtered versions of this probability space.

## 2.2 System Model

Let $R_p$ denote the number of data samples that a single processing node in the system can process in one second using an iteration of the form (4).[3] The focus of this paper is on the high-rate streaming setting, which corresponds to the setup in which the data arrival rate $R_s$ is strictly greater than the data processing rate $R_p$. A naive approach to deal with this computation–streaming mismatch is to discard (per second) a fraction $\alpha := R_s/R_p$ of samples in the system. Such an approach, however, leads to an equivalent reduction in the convergence rate by $\alpha$. We pursue an alternative to this approach in the paper that involves the simultaneous use of $N \geq \lceil \alpha \rceil$ interconnected processors, each individually capable of processing $R_p$ samples per second, within the system. In particular, we advocate the use of such a network of $N$ processors in the following two manners to achieve near-optimal convergence rates (as a function of the number of samples arriving at the system) for estimates of the top eigenvector of $\mathbf{\Sigma}$ in high-rate streaming settings.

### 2.2.1 Distributed Processing Over a Network of Processors

We assume the fast data stream terminates into a data splitter, which splits the original stream with data rate $R_s$ samples per second into $N$ parallel streams, each with data rate $R_s/N$ samples per second, that are then synchronously input to the interconnected network of $N$ processors; see Figure 1(a) for a schematic rendering of such splitting. In order to simplify notation in this setting, we reindex the data samples associated with the $i$-th processor / data stream in the following as $\{\mathbf{x}_{i,t}\}_{t \in \mathbb{Z}_+}$, where the reindexing map $(i, t) \mapsto t'$ is simply defined as $t' = i + (t-1)N$.

---

[3]Note that the parameter $R_p$, among other factors, is a function of the problem dimension $d$; this dependence is being suppressed in the notation for ease of exposition.

We also assume the network of processors implements some message passing protocol that allows it to compute sums of locally stored vectors, i.e., $\sum_{i=1}^{N} \mathbf{a}_i$ for the set of local vectors $\{\mathbf{a}_i\}_{i=1}^{N}$, within the network. This could, for instance, be accomplished using either Reduce or AllReduce primitives within most message passing implementations. We let $R_c(N)$ denote the number of these primitive (sum) operations that the message passing protocol can carry out per second in the network of $N$ processors. Note that, in addition to the number of nodes $N$, this parameter also depends upon the problem dimension $d$, message passing implementation, network topology, and inter-node communications bandwidth, all of which are being abstracted here through $R_c(N)$. We will also be suppressing the dependence of the parameter $R_c$ on $N$ within the notation beyond this section for ease of exposition.

Data splitting among this network of $N$ processors effectively slows down the data streaming rate at each processing node by a factor of $N$. It is under this system model that we present a distributed variant of Krasulina's method, termed D-Krasulina, in Section 3.1 that involves two key operations after each round of splitting of the $N$ samples: ($i$) per-processor computation of the form (4), which requires $1/R_p$ seconds for completion, and ($ii$) a network-wide vector-sum operation, which requires an additional $1/R_c(N)$ seconds for completion. Under such an algorithmic framework, we can express the *effective* network-wide data processing rate, denoted by $R_e$, in terms of the per-node data processing rate $R_p$ and the network-wide sum rate $R_c(N)$ by noticing that the *effective* time it takes to process one sample within the network is given by

$$T_e = \frac{1}{NR_p} + \frac{1}{R_c(N)} \text{ seconds/sample.}$$

Here, the first algorithmic operation gives rise to the first term in the above expression, since it takes $1/R_p$ seconds for computation of the form (4) for $N$ samples, and the network-wide vector-sum operation gives rise to the second term. It then follows that the effective data processing rate is $R_e = 1/T_e = \frac{NR_pR_c(N)}{NR_p+R_c(N)}$.

In the high-rate streaming setting, D-Krasulina can only operate as long as the streaming rate $R_s$ does not exceed the effective processing rate $R_e$, i.e., $R_s \leq R_e$. This gives rise to the following condition on the number of processors $N$ that enables our algorithmic framework to cope with the high-rate streaming data:

$$N\left(R_c(N) - R_s\right) \geq \frac{R_sR_c(N)}{R_p} \quad \implies \quad N \geq \frac{R_sR_c(N)}{R_p\left(R_c(N) - R_s\right)} \; ; \; (R_c(N) - R_s) > 0. \tag{7}$$

Our developments in the following will be based on the assumption that the condition on $N$ in (7) is met. The main analytical challenge for us is understanding the scenarios under which distributed processing over a network of $N$ processors using D-Krasulina still yields near-optimal performance; we address this challenge in Section 4.1 by improving on the analysis in (Balsubramani et al., 2013) for Krasulina's method.

We conclude by remarking on the nature of the lower bound in (7) on the number of processors $N$, given that the bound itself is a function of $N$ through its dependence on the parameter $R_c(N)$ that is expected to decrease with an increase in $N$. The high-performance computing community has access to copious amounts of trace data for different parallel computing environments that allows one to relate the parameter $R_c(N)$ for a given dimension $d$ to the number of processors $N$; see, e.g., (Kavulya et al., 2010). One such relationship could be, for instance, that $R_c(N) \propto 1/N^\kappa$ for some $\kappa \in (0,1]$. The final bound on the number of processors $N$ can then be obtained by plugging such a relationship into the provided bound.[4]

**Remark 1.** *It is straightforward to see that our developments in this paper are also applicable to the setting in which data naturally arrives in a distributed manner at $N$ different nodes, as in Figure 1(b). In addition, our analysis of D-Krasulina is equivalent to that of a mini-batch Krasulina's method running on a powerful-enough single processor that uses a mini-batch of $N$ samples in each iteration.*

---

[4]As an illustrative toy example, let us consider a scenario for a fixed $d$ in which $R_s = 10^3 \text{ sec}^{-1}$, $R_p = 50 \text{ sec}^{-1}$, and the parameter $R_c(N)$ scales as $R_c(N) = \frac{C_R}{N} \text{ sec}^{-1}$ for a constant $C_R$ that depends on the message passing implementation, network topology, and inter-node communications bandwidth. In a "slow" network, corresponding to $C_R \leq 10^3$, there exists no $N$ that satisfies the condition (7). In a "faster" network, corresponding to $C_R > 10^3N$, a necessary condition on the number of processors is $N < 10^{-3}C_R$. But while this necessary condition is satisfied for $1 \leq N < 80$ for $C_R$ as high as $8 \times 10^4$, there still does not exist any $N$ that satisfies (7). In the case of even faster networks, however, we are able to find feasible values of $N$ for running D-Krasulina without discarding any samples; e.g., any $28 \leq N \leq 72$ satisfies the condition (7) for $C_R = 10^5$.

### 2.2.2 Distributed Processing Coupled with Mini-batching

Mini-batching in (centralized) stochastic methods, as discussed in Section 1.2, helps reduce the wall-clock time by reducing the number of read operations per iteration. Mini-batching of samples in distributed settings has the added advantage of reduction in the average number of primitive (sum) operations per processed sample, which further reduces the wall-clock time. It is in this vein that we put forth a mini-batched variant of D-Krasulina, which is termed DM-Krasulina, in Section 3.2.

Similar to the case of D-Krasulina (cf. Figure 1), there are several equivalent system models that can benefit from the DM-Krasulina framework. In keeping with our theme of fast streaming data, as well as for the sake of concreteness, we assume the system buffers (i.e., mini-batches) $B := bN \geq \lceil R_s/R_p \rceil$ samples of the incoming data stream every $B/R_s$ seconds for some parameter $b \in \mathbb{Z}_+$. This *network-wide* mini-batch of $B$ samples is then split into $N$ parallel (local) mini-batches, each comprising $b = B/N$ samples, which are then synchronously input to the interconnected network of $N$ processors at a rate of $R_s/N$ samples per second and collaboratively processed by DM-Krasulina. In each iteration $t$ of DM-Krasulina, therefore, the network processes a total of $B \geq N$ samples, as opposed to $N$ samples for D-Krasulina. In order to simplify notation in this mini-batched distributed setting, we reindex the $b$ data samples in the mini-batch associated with the $i$-th processor in iteration $t$ of DM-Krasulina as $\{\mathbf{x}_{i,j,t}\}_{j=1,t\in\mathbb{Z}_+}^{j=b}$, where the reindexing map $(i, j, t) \mapsto t'$ is defined as $t' = j + (i-1)b + (t-1)B$.

It is straightforward to see from the discussion surrounding (7) that the DM-Krasulina framework can process all data samples arriving at the system as long as $N \geq \frac{bR_cR_s}{R_p(bR_c-R_s)}$ and $(bR_c - R_s) > 0$. However, when this condition is violated due to faster streaming rate $R_s$, slower processing rate $R_p$, slower summation rate $R_c$, or any combination thereof, it becomes necessary for DM-Krasulina to discard $\mu := \left(\frac{bR_s}{R_p} + \frac{NR_s}{R_c}\right) - B$ samples at the splitter per iteration. The main analytical challenges for DM-Krasulina are, therefore, twofold: first, assuming $\mu = 0$, characterize the mini-batch size $B$ that leads to near-optimal convergence rates for DM-Krasulina in terms of the total number of samples arriving at the system; second, when discarding of samples becomes necessary, characterize the interplay between $B$ and $\mu$ that allows DM-Krasulina to still achieve (order-wise) near-optimal convergence rates. We address both these challenges in Section 4.2.

## 3 Proposed Distributed Stochastic Algorithms

We now formally describe the two stochastic algorithms, termed D-Krasulina and DM-Krasulina, that can be used to solve the 1-PCA problem from high-rate streaming data under the two setups described in Section 2.2.1 and Section 2.2.2, respectively.

### 3.1 Distributed Krasulina's Method (D-Krasulina) for High-rate Streaming Data

Recall from the discussion in Section 2.2.1 that each node $i$ in the network receives data sample $\mathbf{x}_{i,t}$ in iteration $t$ of the distributed implementation, which comprises $N$ processing nodes. Unlike the centralized Krasulina's method (cf. (4)), therefore, any distributed variant of Krasulina's method needs to process $N$ samples in every iteration $t$. Using $\mathbf{A}_{i,t}$ as a shorthand for $\mathbf{x}_{i,t}\mathbf{x}_{i,t}^{\mathrm{T}}$, one natural extension of (4) that processes $N$ samples in each iteration is as follows:

$$\mathbf{v}_t = \mathbf{v}_{t-1} + \gamma_t \left( \frac{1}{N} \sum_{i=1}^N \mathbf{A}_{i,t}\mathbf{v}_{t-1} - \frac{1}{\|\mathbf{v}_{t-1}\|_2^2} \left( \mathbf{v}_{t-1}^{\mathrm{T}} \frac{1}{N} \sum_{i=1}^N \mathbf{A}_{i,t}\mathbf{v}_{t-1}\mathbf{v}_{t-1} \right) \right) = \mathbf{v}_{t-1} + \gamma_t \boldsymbol{\xi}_t. \tag{8}$$

One natural question here is whether (8) can be computed within our distributed framework. The answer to this is in the affirmative under the assumption $N \geq \frac{R_sR_c}{R_p(R_c-R_s)}$, with the implementation (termed D-Krasulina) formally described in Algorithm 1.[5]

Notice that unlike classical Krasulina's method, which processes a total of $t$ samples after $t$ iterations, D-Krasulina processes a total of $Nt$ samples after $t$ iterations in order to provide an estimate $\mathbf{v}_t$ of the

---

[5]Here, and in the following, the implicit assumption is that the quantity $R_c - R_s$ (resp., $bR_c - R_s$) is strictly positive for D-Krasulina (resp., DM-Krasulina).

---

**Algorithm 1** Distributed Krasulina's Method (D-Krasulina)

---

**Input:** Incoming data streams at $N$ processors, expressed as $\left\{\mathbf{x}_{i,t} \overset{\text{i.i.d.}}{\sim} \mathcal{P}_x\right\}_{i=1,t\in\mathbb{Z}_+}^{N}$, and a step-size sequence $\{\gamma_t \in \mathbb{R}_+\}_{t\in\mathbb{Z}_+}$

**Initialize:** All processors initialize with $\mathbf{v}_0 \in \mathbb{R}^d$ randomly generated over the unit sphere

1: **for** $t = 1, 2, \ldots,$ **do**
2:     **(In Parallel)** Processor $i$ receives data sample $\mathbf{x}_{i,t}$ and updates $\boldsymbol{\xi}_{i,t}$ locally as follows:

$$\forall i \in \{1, \ldots, N\}, \quad \boldsymbol{\xi}_{i,t} \leftarrow \mathbf{x}_{i,t}\mathbf{x}_{i,t}^{\mathrm{T}}\mathbf{v}_{t-1} - \frac{\mathbf{v}_{t-1}^{\mathrm{T}}\mathbf{x}_{i,t}\mathbf{x}_{i,t}^{\mathrm{T}}\mathbf{v}_{t-1}\mathbf{v}_{t-1}}{\|\mathbf{v}_{t-1}\|_2^2}$$

3:     Compute $\boldsymbol{\xi}_t \leftarrow \frac{1}{N}\sum_{i=1}^{N}\boldsymbol{\xi}_{i,t}$ in the network using a distributed vector-sum subroutine
4:     Update eigenvector estimate in the network as follows: $\mathbf{v}_t \leftarrow \mathbf{v}_{t-1} + \gamma_t\boldsymbol{\xi}_t$
5: **end for**
**Return:** An estimate $\mathbf{v}_t$ of the eigenvector $\mathbf{q}^*$ of $\boldsymbol{\Sigma}$ associated with $\lambda_1(\boldsymbol{\Sigma})$

---

top eigenvector $\mathbf{q}^*$ of $\boldsymbol{\Sigma}$. Another natural question, therefore, is whether the estimate $\mathbf{v}_t$ returned by D-Krasulina can converge to $\mathbf{q}^*$ at the near-optimal rate of $O\left(1/\# \text{ of processed samples}\right)$. Convergence analysis of D-Krasulina in Section 4 establishes that the answer to this is also in the affirmative under appropriate conditions that are specified in Theorem 1. An important interpretation of this result is that our proposed distributed implementation of Krasulina's method can lead to linear speed-up as a function of the number of processing nodes $N$ in the network.

### 3.2 Mini-batched D-Krasulina (DM-Krasulina) for High-rate Streaming Data

The distributed, mini-batched setup described in Section 2.2.2 entails each node $i$ receiving a mini-batch of $b = B/N$ data samples, $\{\mathbf{x}_{i,j,t}\}_{j=1}^{b}$, in each iteration $t$, for a total of $B = bN$ samples across the network in every iteration. Similar to (8), these $B$ samples can in principle be processed by the following variant of the original Krasulina's iteration:

$$\mathbf{v}_t = \mathbf{v}_{t-1} + \gamma_t \underbrace{\left(\frac{1}{B}\sum_{i=1}^{N}\sum_{j=1}^{b}\mathbf{A}_{i,j,t}\mathbf{v}_{t-1} - \frac{1}{\|\mathbf{v}_{t-1}\|_2^2}\left(\mathbf{v}_{t-1}^{\mathrm{T}}\frac{1}{B}\sum_{i=1}^{N}\sum_{j=1}^{b}\mathbf{A}_{i,j,t}\mathbf{v}_{t-1}\mathbf{v}_{t-1}\right)\right)}_{\boldsymbol{\xi}_t}, \tag{9}$$

where $\mathbf{A}_{i,j,t}$ is a shorthand for $\mathbf{x}_{i,j,t}\mathbf{x}_{i,j,t}^{\mathrm{T}}$. Practical computation of (9) within our distributed framework, however, requires consideration of two different scenarios.

- *Scenario 1:* The mini-batched distributed framework satisfies $N \geq \frac{bR_cR_s}{R_p(bR_c-R_s)}$. This enables incorporation of every sample arriving at the system into the eigenvector estimate.

- *Scenario 2:* The mini-batched distributed framework leads to the condition $N < \frac{bR_cR_s}{R_p(bR_c-R_s)}$. This necessitates discarding of $\mu = \left(\frac{bR_s}{R_p} + \frac{NR_s}{R_c}\right) - B$ samples per iteration in the system. Stated differently, the system receives $B + \mu$ samples per iteration in this scenario, but only $B$ samples per iteration are incorporated into the eigenvector estimate.

We now formally describe the algorithm (termed DM-Krasulina) that implements (9) under both these scenarios in Algorithm 2.

Speaking strictly in terms of implementation, the mini-batched setup of DM-Krasulina allows one to relax the condition $N \geq \frac{R_sR_c}{R_p(R_c-R_s)}$ associated with D-Krasulina to either $N \geq \frac{bR_cR_s}{R_p(bR_c-R_s)}$, which still incorporates all samples into the eigenvector estimate, or $N < \frac{bR_cR_s}{R_p(bR_c-R_s)}$, which involves discarding of $\mu > 0$ samples

---

**Algorithm 2** Distributed Mini-batch Krasulina's Method (DM-Krasulina)

---

**Input:** Incoming streams of mini-batches $\left\{\mathbf{x}_{i,j,t} \overset{\text{i.i.d.}}{\sim} \mathcal{P}_x\right\}_{i,j=1,t\in\mathbb{Z}_+}^{N,b}$ at $N$ processors, size of the network-wide mini batch $B := bN$, and a step-size sequence $\{\gamma_t \in \mathbb{R}_+\}_{t\in\mathbb{Z}_+}$

**Initialize:** All processors initialize with $\mathbf{v}_0 \in \mathbb{R}^d$ randomly generated over the unit sphere

1: **for** $t = 1, 2, \ldots,$ **do**
2:     **(In Parallel)** $\forall i \in \{1, \ldots, N\}, \quad \boldsymbol{\xi}_{i,t} \leftarrow 0$
3:     **for** $j = 1, \ldots, b$ **do**
4:         **(In Parallel)** Processor $i$ receives data sample $\mathbf{x}_{i,j,t}$ and updates $\boldsymbol{\xi}_{i,t}$ locally as follows:

$$\forall i \in \{1, \ldots, N\}, \quad \boldsymbol{\xi}_{i,t} \leftarrow \boldsymbol{\xi}_{i,t} + \mathbf{x}_{i,j,t}\mathbf{x}_{i,j,t}^{\mathrm{T}}\mathbf{v}_{t-1} - \frac{\mathbf{v}_{t-1}^{\mathrm{T}}\mathbf{x}_{i,j,t}\mathbf{x}_{i,j,t}^{\mathrm{T}}\mathbf{v}_{t-1}\mathbf{v}_{t-1}}{\|\mathbf{v}_{t-1}\|_2^2}$$

5:     **end for**
6:     Compute $\boldsymbol{\xi}_t \leftarrow \frac{1}{B}\sum_{i=1}^N \boldsymbol{\xi}_{i,t}$ in the network using a distributed vector-sum subroutine
7:     Update eigenvector estimate in the network as follows: $\mathbf{v}_t \leftarrow \mathbf{v}_{t-1} + \gamma_t \boldsymbol{\xi}_t$
8:     **if** $N < \frac{bR_cR_s}{R_p(bR_c - R_s)}$ **then**
9:         The system (e.g., data splitter/buffer) receives $(B + \mu)$ additional samples during execution of Steps 2–7, out of which $\mu \in \mathbb{Z}_+$ samples are discarded
10:     **end if**
11: **end for**

**Return:** An estimate $\mathbf{v}_t$ of the eigenvector $\mathbf{q}^*$ of $\boldsymbol{\Sigma}$ associated with $\lambda_1(\boldsymbol{\Sigma})$

---

per algorithmic iteration. While this makes DM-Krasulina particularly attractive for systems with slower communication links, the major analytical hurdle here is understanding the interplay between the different problem parameters that still allows DM-Krasulina to achieve near-optimal convergence rates in terms of the number of samples received at the system. We tease out this interplay as part of the convergence analysis of DM-Krasulina in Section 4.

### 3.3 A Note on the Processing of Non-centered and Non-i.i.d. Data

Both D-Krasulina and DM-Krasulina have been developed under the assumptions of zero-mean (i.e., *centered*) and i.i.d. data samples. In this section, we discuss one possible approach to handling non-centered data using the two algorithms and also provide a rationale for the applicability of D-Krasulina and DM-Krasulina in the face of any shifts in the data distribution.

In the case of non-centered data, one simple strategy that works for D-Krasulina and DM-Krasulina is to maintain a (network-wide) running average of the non-centered data samples, and then use it to center the data samples at each processor before applying Step 2 (resp., Step 4) in Algorithm 1 (resp., Algorithm 2). While such a modification requires an extension of the convergence analysis presented in the next section, this can be accomplished in a manner similar to the analytical extension in (Zhou & Bai, 2021) for the centralized Oja's rule with non-centered data.

Next, while the forthcoming convergence analysis for D-Krasulina and DM-Krasulina has been provided under the assumption of i.i.d. data samples, the two algorithms are expected to remain effective in the non-i.i.d. data setting. This is because D-Krasulina and DM-Krasulina first essentially compute a new gradient-like quantity using the latest batch of data samples at each time $t$ (cf. Step 2 in Algorithm 1 and Step 4 in Algorithm 2), and then update their respective eigenvector estimates using this quantity. In particular, any shifts in the data distribution can be tracked by the two algorithms because of such an update rule. It is because of this reason that algorithms such as Oja's rule and Krasulina's method are also often employed for the problem of *subspace tracking* (see, e.g., (Bin Yang, 1995; Chatterjee, 2005; Doukopoulos & Moustakides, 2008)). Since providing a formal analysis of such tracking capabilities of D-Krasulina and DM-Krasulina for non-i.i.d. data is beyond the scope of this paper, we leave it for future work.

## 4 Convergence Analysis of D-Krasulina and DM-Krasulina

Our convergence analysis of D-Krasulina and DM-Krasulina is based on understanding the rate at which the so-called *potential function* $\Psi_t$ of these methods converges to zero as a function of the number of algorithmic iterations $t$. Formally, this potential function $\Psi_t$ is defined as follows.

**Definition 2** (Potential function). *Let $\mathbf{q}^*$ be the eigenvector of $\boldsymbol{\Sigma}$ associated with $\lambda_1(\boldsymbol{\Sigma})$ and let $\mathbf{v}_t$ be an estimate of $\mathbf{q}^*$ returned by an iterative algorithm in iteration $t$. Then the quality of the estimate $\mathbf{v}_t$ can be measured in terms of the potential function $\Psi_t : \mathbf{v}_t \mapsto [0, 1]$ that is defined as*

$$\Psi_t := 1 - \frac{(\mathbf{v}_t^{\mathrm{T}} \mathbf{q}^*)^2}{\|\mathbf{v}_t\|^2}. \tag{10}$$

Notice that $\Psi_t$ is a measure of estimation error, which approaches 0 as $\mathbf{v}_t$ converges to any scalar multiple of $\mathbf{q}^*$. This measure, which essentially computes sine squared of the angle between $\mathbf{q}^*$ and $\mathbf{v}_t$, is frequently used in the literature to evaluate the performance of PCA algorithms. In particular, when one initializes an algorithm with a random vector $\mathbf{v}_0$ uniformly distributed over the unit sphere in $\mathbb{R}^d$ then it can be shown that $\mathbb{E}\{\Psi_0\} \leq 1 - 1/d$ (Balsubramani et al., 2013). While this is a statement in expectation for $t = 0$, our analysis relies on establishing such a statement in probability for *any* $t \geq 0$ for both D-Krasulina and DM-Krasulina. Specifically, we show in Theorem 3 that $\sup_{t \geq 0} \Psi_t \leq 1 - O(1/d)$ with high probability as long as $\gamma_t = c/(L + t)$ for any constant $c$ and a large-enough constant $L$.

All probabilistic analysis in the following uses a filtration $(\mathcal{F}_t)_{t \geq 0}$ of sub $\sigma$-algebras of $\mathcal{F}$ on the sample space $\Omega$, where the $\sigma$-algebra $\mathcal{F}_t$ captures the progress of the iterates of the two proposed stochastic algorithms up to iteration $t$. Mathematically, let us define the sample covariance matrix $\mathbf{A}_t$ as $\mathbf{A}_t := \frac{1}{N} \sum_{i=1}^{N} \mathbf{A}_{i,t}$ and $\mathbf{A}_t := \frac{1}{B} \sum_{i=1}^{N} \sum_{j=1}^{b} \mathbf{A}_{i,j,t}$ for D-Krasulina and DM-Krasulina, respectively. In order to simplify notation and unify some of the analysis of D-Krasulina and DM-Krasulina, we will be resorting to the use of random matrices $\mathbf{A}_t$, as opposed to $\mathbf{x}_{i,t}$ and $\mathbf{x}_{i,j,t}$, in the following. We then have the following definition of $\sigma$-algebras in the filtration.

**Definition 3** ($\sigma$-algebra $\mathcal{F}_t$). *The $\sigma$-algebra $\mathcal{F}_t \subseteq \mathcal{F}$ on sample space $\Omega$ for both D-Krasulina and DM-Krasulina is defined as the $\sigma$-algebra generated by the vector-/matrix-valued random variables $(\mathbf{v}_0, \mathbf{A}_1, \ldots, \mathbf{A}_t)$, i.e., $\mathcal{F}_t := \sigma(\mathbf{v}_0, \mathbf{A}_1, \ldots, \mathbf{A}_t)$.*

In addition to the filtration $(\mathcal{F}_t)_{t \geq 0}$, the forthcoming analysis also uses a sequence of nested sample spaces that is defined as follows.

**Definition 4** (Nested sample spaces). *Let $(t_0, \epsilon_0), (t_1, \epsilon_1), (t_2, \epsilon_2), \ldots, (t_J, \epsilon_J)$ be a sequence of pairs such that $0 = t_0 < t_1 < t_2 < \ldots < t_J$ and $\epsilon_0 > \epsilon_1 > \epsilon_2 > \ldots > \epsilon_J > 0$ for any non-negative integer $J$. We then define a sequence $(\Omega_t')_{t \in \mathbb{Z}_+}$ of nested sample spaces such that $\Omega \supset \Omega_1' \supset \Omega_2' \supset \ldots$, each $\Omega_t'$ is $\mathcal{F}_{t-1}$-measurable, and*

$$\Omega_t' := \left\{ \omega \in \Omega : \forall 0 \leq j \leq J, \sup_{t_j \leq l < t} \Psi_l(\omega) \leq 1 - \epsilon_j \right\}. \tag{11}$$

*Here, $\omega$ denotes an outcome within the sample space $\Omega$ and $\Psi_l(\omega)$ is the (random) potential function after the $l$-th iteration of D-Krasulina / DM-Krasulina that is being explicitly written as a function of the outcomes $\omega$ in the sample space.*

In words, the sample space $\Omega_t'$ corresponds to that subset of the original sample space for which the error $\Psi_l$ in all iterations $l \in \{t_j, \ldots, t-1\}$ is below $1 - \epsilon_j$, where $j \in \{0, \ldots, J\}$. In the following, we use the notation $\mathbb{E}_t\{\cdot\}$ and $\mathbb{P}_t(\cdot)$ to denote conditional expectation and conditional probability, respectively, with respect to $\Omega_t'$.

An immediate implication of Definition 4 is that, for appropriate choices of $\epsilon_j$'s, it allows us to focus on those subsets of the original sample space that ensure convergence of iterates of the proposed algorithms to the top eigenvector $\mathbf{q}^*$ at the desired rates. The main challenge here is establishing that such subsets have high probability measure, i.e., $\mathbb{P}\left(\cap_{t>0} \Omega_t'\right) \geq 1 - \delta$ for any $\delta > 0$. We obtain such a result in Theorem 4 in the following. We are now ready to state our main results for D-Krasulina and DM-Krasulina.

### 4.1 Convergence of D-Krasulina (Algorithm 1)

The first main result of this paper shows that D-Krasulina results in linear speed-up in convergence rate as a function of the number of processing nodes, i.e., the potential function for D-Krasulina converges to 0 at a rate of $O(1/Nt)$. Since the system receives a total of $Nt$ samples at the end of $t$ iterations of D-Krasulina, this result establishes that D-Krasulina is order-wise near-optimal in terms of sample complexity for the streaming PCA problem. The key to proving this result is characterizing the convergence behavior of D-Krasulina in terms of variance of the sample covariance matrix $\mathbf{A}_t := \frac{1}{N}\sum_{i=1}^{N} \mathbf{A}_{i,t}$ that is implicitly computed within D-Krasulina. We denote this variance as $\sigma_N^2$, which has the form

$$\sigma_N^2 := \mathbb{E}_{\mathcal{P}_x}\left\{\left\|\frac{1}{N}\sum_{i=1}^{N}\mathbf{x}_{i,t}\mathbf{x}_{i,t}^{\mathrm{T}} - \boldsymbol{\Sigma}\right\|_F^2\right\}. \tag{12}$$

It is straightforward to see from Definition 1 and (12) that $\sigma_N^2 = \sigma^2/N$. This reduction in variance of the sample covariance matrix within D-Krasulina essentially enables the linear speed-up in convergence. In terms of specifics, we have the following convergence result for D-Krasulina.

**Theorem 1.** *Fix any $\delta \in (0,1)$ and pick $c := c_0/2(\lambda_1 - \lambda_2)$ for any $c_0 > 2$. Next, define*

$$L_1 := \frac{64edr^4\max(1,c^2)}{\delta^2}\ln\frac{4}{\delta}, \quad L_2 := \frac{512e^2d^2\sigma_N^2\max(1,c^2)}{\delta^4}\ln\frac{4}{\delta}, \tag{13}$$

*pick any $L \geq L_1 + L_2$, and choose the step-size sequence as $\gamma_t := c/(L+t)$. Then, as long as Assumptions [**A1**] and [**A2**] hold, we have for D-Krasulina that there exists a sequence $(\Omega_t')_{t\in\mathbb{Z}_+}$ of nested sample spaces such that $\mathbb{P}\left(\cap_{t>0}\Omega_t'\right) \geq 1 - \delta$ and*

$$\mathbb{E}_t\{\Psi_t\} \leq C_1\left(\frac{L+1}{t+L+1}\right)^{\frac{c_0}{2}} + C_2\left(\frac{\sigma_N^2}{t+L+1}\right), \tag{14}$$

*where $C_1$ and $C_2$ are constants defined as*

$$C_1 := \frac{1}{2}\left(\frac{4ed}{\delta^2}\right)^{\frac{5}{2\ln 2}}e^{2c^2\lambda_1^2/L} \quad \text{and} \quad C_2 := \frac{8c^2e^{(c_0+2c^2\lambda_1^2)/L}}{(c_0-2)}.$$

**Remark 2.** *While we can obtain a similar result for the case of $c_0 \leq 2$, that result does not lead to any convergence speed-up. In particular, the convergence rate in that case becomes $O(t^{-c_0/2})$, which matches the one in (Balsubramani et al., 2013).*

**Discussion.** A proof of Theorem 1, which is influenced by the proof technique employed in (Balsubramani et al., 2013), is provided in Section 5. Here, we discuss some of the implications of this result, especially in relation to (Balsubramani et al., 2013) and (Jain et al., 2016). The different problem parameters affecting the performance of stochastic methods for streaming PCA include: ($i$) dimensionality of the ambient space, $d$, ($ii$) eigengap of the population covariance matrix, $(\lambda_1 - \lambda_2)$, ($iii$) upper bound on norm of the received data samples, $r$, and ($iv$) variance of the sample covariance matrix, $\sigma^2$ and/or $\sigma_N^2$. Theorem 1 characterizes the dependence of D-Krasulina on all these parameters and significantly improves on the related result provided in (Balsubramani et al., 2013).

First, Theorem 1 establishes D-Krasulina can achieve the convergence rate $O(\sigma_N^2/t) \equiv O(\sigma^2/Nt)$ with high probability (cf. (14)). This is in stark contrast to the result in (Balsubramani et al., 2013), which is independent of variance of the sample covariance matrix, thereby only guaranteeing convergence rate of $O(r^4/t)$ for D-Krasulina and its variants. This ability of variants of Krasulina's methods to achieve faster convergence through variance reduction is arguably one of the most important aspects of our analysis. Second, in comparison with (Balsubramani et al., 2013), Theorem 1 also results in an improved lower bound on choice of $L$ by splitting it into two quantities, viz., $L_1$ and $L_2$ (cf. (13)). This improved bound allows larger step sizes, which also results in faster convergence. In terms of specifics, $L_1$ in the theorem is on the

order of $\Omega(r^4 d/\delta^2)$, which is an improvement over $\Omega(r^4 d^2/\delta^4)$ bound of (Balsubramani et al., 2013). On the other hand, while $L_2$ has same dependence on $\delta$ and $d$ as (Balsubramani et al., 2013), it depends on $\sigma_N^2$ instead of $r^4$ and, therefore, it reduces with an increase in $N$. Third, the improved lower bound on $L$ also allows for an improved dependence on the dimensionality $d$ of the problem. Specifically, for large enough $t$ and $N$, the dependence on $d$ in (14) is due to the higher-order (first) term and is of the order $O(d^{\frac{5}{2\ln 2}+\frac{c_0}{2}})$, as opposed to $O(d^{\frac{5}{2\ln 2}+c_0})$ for (Balsubramani et al., 2013). It is worth noting here, however, that this is still loser than the result in (Jain et al., 2016) that has only $\log^2(d)$ dependence on $d$ in higher-order error terms.

Fourth, in terms of the eigengap, our analysis has optimal dependence of $1/(\lambda_1 - \lambda_2)^2$, which also matches the dependence in (Balsubramani et al., 2013) and (Jain et al., 2016). It is important to note here, however, that knowledge of the eigengap $(\lambda_1 - \lambda_2)$ is not necessary to run Oja's rule, Krasulina's method, D-Krasulina, or any of the related stochastic methods in a practical setting. Specifically, it can be seen from Theorem 1 that the eigengap is only needed to set the step size in D-Krasulina for the optimal convergence rate. In practice, however, step sizes of the form $\tilde{c}/t$ work well for D-Krasulina and the related methods, and a simple yet highly effective strategy for setting the step size in these methods is to estimate the parameter $\tilde{c}$ by running multiple instances of the method during a warm-up phase. Such an approach is akin to approximating several problem-related parameters using a single parameter $\tilde{c}$, and is the one we have followed for the numerical experiments discussed in Section 6.

Finally, we compare the recommended step-size sequence $\gamma_t = c/(L + t)$ in Theorem 1 to the ones in (Balsubramani et al., 2013) and (Jain et al., 2016). Since the step sizes in these two prior works also take the form $\gamma_t = c/(L + t)$, all three works are equivalent to each other in terms of scaling of the step size as a function of $t$. But in terms of the initial step size, and assuming small enough $\delta$ in Theorem 1, $\gamma_1$ is the largest for (Jain et al., 2016), second-largest for this work, and the smallest for (Balsubramani et al., 2013). In relation to our work, this difference in the initial step size in the case of (Balsubramani et al., 2013) is due to the improved lower bound on $L$ in Theorem 1. In the case of (Jain et al., 2016), this difference is attributable to the fact that the parameter $L$ is independent of $\delta$ in that work. Stated differently, we are able to vary the probability of success $1 - \delta$ in this work by making the parameter $L$ be a function of $\delta$, with the caveat being that the initial step size $\gamma_1$ gets smaller as $\delta$ decreases. In contrast, a fixed $L$ in (Jain et al., 2016) can be thought of as one of the reasons the probability of success is fixed at $3/4$ in that work. We conclude by noting that this dependence of the performance of D-Krasulina on different problem parameters is further highlighted through numerical experiments in Section 6.

**Remark 3.** *While Theorem 1 is for (a distributed variant of) Krasulina's method, Oja's rule can also be analyzed using similar techniques; see, e.g., the discussion in (Balsubramani et al., 2013).*

**Remark 4.** *Recall from the discussion in Section 1.1 that an iteration of Krasulina's method is similar to that for SGD applied to the optimization problem (5). A natural question to ask then is whether D-Krasulina can be "accelerated" in much the same way SGD can be accelerated by adding a momentum term to its iteration expression. The authors in (De Sa et al., 2018), however, argue that naively applying momentum to Oja's rule or the power iteration, both of which are closely related to Krasulina's method, results in worst performance since this increases the effect of the noise within the iterates. And while the noise within the iterates can be controlled through variance reduction techniques, as done in (De Sa et al., 2018) to accelerate the power iteration for eigenvector computation, such techniques typically require multiple data passes and are therefore not suited for the setting in which data samples continuously stream into the system.*

### 4.2 Convergence of DM-Krasulina (Algorithm 2)

The convergence analysis of DM-Krasulina follows from slight modifications of the proof of Theorem 1 for D-Krasulina. The final set of results, which covers the two scenarios of zero data loss ($\mu = 0$) and some data loss ($\mu > 0$) in each iteration, is characterized in terms of variance of the (mini-batched) sample covariance $\mathbf{A}_t := \frac{1}{B} \sum_{i=1}^N \sum_{j=1}^b \mathbf{A}_{i,j,t}$ associated with DM-Krasulina. We denote this variance as $\sigma_B^2$, which is given by

$$\sigma_B^2 := \mathbb{E}_{\mathcal{P}_x} \left\{ \left\| \frac{1}{B} \sum_{i=1}^N \sum_{j=1}^b \mathbf{x}_{i,j,t} \mathbf{x}_{i,j,t}^{\mathrm{T}} - \boldsymbol{\Sigma} \right\|_F^2 \right\}. \tag{15}$$

It is once again straightforward to see that $\sigma_B^2 = \sigma^2/B$. We now split our discussion of the convergence of DM-Krasulina according to the two scenarios discussed in Section 3.2.

### 4.2.1 Scenario 1—DM-Krasulina with no data loss: $N \geq \frac{bR_cR_s}{R_p(bR_c - R_s)} \implies \mu = 0$

Analytically, this scenario is similar to D-Krasulina, with the only difference being that we are now incorporating an average of $B$ sample covariances $\mathbf{x}_{i,j,t}\mathbf{x}_{i,j,t}^T$ in the estimate in each iteration (as opposed to $N$ sample covariances for D-Krasulina). We therefore have the following generalization of Theorem 1 in this scenario.

**Theorem 2.** *Let the parameters and constants be as specified in Theorem 1, except that the parameter $L_2$ is now defined as $L_2 := \frac{512e^2d^2\sigma_B^2\max(1,c^2)}{\delta^4}\ln\frac{4}{\delta}$. Then, as long as Assumptions [**A1**] and [**A2**] hold, we have for DM-Krasulina that $\mathbb{P}\left(\cap_{t>0}\Omega_t'\right) \geq 1 - \delta$ and*

$$\mathbb{E}_t\{\Psi_t\} \leq C_1\left(\frac{L+1}{t+L+1}\right)^{\frac{c_0}{2}} + C_2\left(\frac{\sigma_B^2}{t+L+1}\right). \tag{16}$$

The proof of this theorem can be obtained from that of Theorem 1 by replacing $1/N$ and $\sigma_N^2$ in there with $1/B$ and $\sigma_B^2$, respectively. Similar to the case of D-Krasulina, this theorem establishes that DM-Krasulina can also achieve linear speed-up in convergence as a function of the network-wide mini-batch size $B$ with very high probability, i.e., $\mathbb{E}_t\{\Psi_t\} = O(\sigma_B^2/t) \equiv O(\sigma^2/Bt)$.

Our discussions of D-Krasulina and DM-Krasulina have so far been focused on the infinite-sample regime, in which the number of algorithmic iterations $t$ for both algorithms can grow unbounded. We now focus on the implications of our results for the finite-sample regime, in which a final estimate is produced at the end of arrival of a total of $T \gg 1$ samples.[6] This finite-sample regime leads to an interesting interplay between $N$ (resp., $B$) and the total number of samples $T$ for linear speed-up of D-Krasulina (resp., DM-Krasulina). We describe this interplay in the following for DM-Krasulina; the corresponding result for D-Krasulina follows by simply replacing $B$ with $N$ in this result.

**Corollary 1.** *Let the parameters and constants be as specified in Theorem 2. Next, pick parameters $(L_1', L_2')$ such that $L_1' \geq L_1$ and $L_2' \geq L_2/\sigma_B^2$, and define the final number of algorithmic iterations for DM-Krasulina as $T_B := T/B$. Then, as long as Assumptions [**A1**] and [**A2**] hold and the network-wide mini-batch size satisfies $B \leq T^{1-\frac{2}{c_0}}$, we have that $\mathbb{P}\left(\cap_{0<t\leq T_B}\Omega_t'\right) \geq 1 - \delta$ and*

$$\mathbb{E}_{T_B}\{\Psi_{T_B}\} \leq c_0C_1\frac{L_1'^{c_0/2}}{T} + c_0C_1\left(\frac{\sigma^2L_2'}{T}\right)^{c_0/2} + \frac{C_2\sigma^2}{T}. \tag{17}$$

*Proof.* Substituting $t = T_B$ in (16) and using simple upper bounds yield

$$\mathbb{E}_{T_B}\{\Psi_{T_B}\} \leq C_1\left(\frac{L+1}{L+T_B}\right)^{\frac{c_0}{2}} + C_2\left(\frac{\sigma_B^2}{T_B}\right) \leq 2C_1\left(\frac{L}{T_B}\right)^{\frac{c_0}{2}} + C_2\left(\frac{\sigma_B^2}{T_B}\right).$$

Next, substituting $L = L_1' + \sigma_B^2L_2'$ in this expression gives us

$$\mathbb{E}_{T_B}\{\Psi_{T_B}\} \leq c_0C_1\left(\frac{L_1'}{T_B}\right)^{\frac{c_0}{2}} + c_0C_1\left(\frac{\sigma_B^2L_2'}{T_B}\right)^{\frac{c_0}{2}} + C_2\left(\frac{\sigma_B^2}{T_B}\right). \tag{18}$$

Since $\sigma_B^2 = \sigma^2/B$ and $T_B = T/B$, (18) reduces to the following expression:

$$\mathbb{E}_{T_B}\{\Psi_{T_B}\} \leq c_0C_1\left(\frac{BL_1'}{T}\right)^{c_0/2} + c_0C_1\left(\frac{\sigma^2L_2'}{T}\right)^{c_0/2} + \frac{C_2\sigma^2}{T}.$$

The proof now follows from the assumption that $B \leq T^{1-\frac{2}{c_0}}$. $\qquad\square$

---

[6]An implicit assumption here is that $T$ is large enough that it precludes the use of a batch PCA algorithm.

**Discussion.** Corollary 1 dictates that linear convergence speed-up for DM-Krasulina (resp., D-Krasulina) occurs in the finite-sample regime provided the network-wide mini-batch size $B$ (resp., number of processing nodes $N$) scales sublinearly with the total number of samples $T$. In particular, the proposed algorithms achieve the best (order-wise) convergence rate of $O(1/T)$ for appropriate choices of system parameters. We also corroborate this theoretical finding with numerical experiments involving synthetic and real-world data in Section 6.

### 4.2.2 Scenario 2—DM-Krasulina with data loss: $N < \frac{bR_c R_s}{R_p(bR_c - R_s)} \implies \mu > 0$

The statement of Theorem 2 for DM-Krasulina in the lossless setting immediately carries over to the resource-constrained setting that causes loss of $\mu$ ($> 0$) samples per iteration. The implication of this result is that DM-Krasulina can achieve convergence rate of $O(1/Bt)$ in the infinite-sample regime after receiving a total of $(B + \mu)t$ samples. Therefore, it trivially follows that DM-Krasulina can achieve order-wise near-optimal convergence rate in the infinite-sample regime as long as $\mu = O(B)$.

We now turn our attention to understanding the interplay between $\mu$, $B$, and the total number of samples $T$ arriving at the system for the resource-constrained finite-sample setting for DM-Krasulina. To this end, we have the following generalization of Corollary 1.

**Corollary 2.** *Let the parameters and constants be as specified in Corollary 2, and define the final number of algorithmic iterations for DM-Krasulina as $T_B^\mu := T/(B + \mu)$. Then, as long as Assumptions [**A1**] and [**A2**] hold, we have that $\mathbb{P}\left(\cap_{t>0}\Omega_t'\right) \geq 1 - \delta$ and*

$$\mathbb{E}_{T_B^\mu}\left\{\Psi_{T_B^\mu}\right\} \leq c_0 C_1 \left(\frac{(B+\mu)L_1'}{T}\right)^{c_0/2} + c_0 C_1 \left(\frac{(B+\mu)\sigma^2 L_2'}{BT}\right)^{c_0/2} + \frac{C_2 \sigma^2 (B+\mu)}{BT}. \tag{19}$$

*Proof.* The proof of this corollary follows from replacing $T_B$ with $T_B^\mu$ in (18) and subsequently substituting the values of $T_B^\mu$ and $\sigma_B^2$ in there. $\square$

**Discussion.** Recall that since the distributed framework receives a total of $T$ samples, it is desirable to achieve convergence rate of $O(1/T)$. It can be seen from Corollary 2 that the first and the third terms in (19) are the ones that dictate whether DM-Krasulina can achieve the (order-wise) optimal rate of $O(1/T)$. To this end, the first term in (19) imposes the condition $(B + \mu) \leq T^{1-2/c_0}$, i.e., the total number of samples received at the system (both processed and discarded) per iteration must scale sublinearly with the final number of samples $T$. In addition, the third term in (19) imposes the condition $\mu = O(B)$, i.e., the number of samples discarded by the system in each iteration must scale no faster than the number of samples processed by the system in each iteration. Once these two conditions are satisfied, Corollary 2 guarantees near-optimal convergence for DM-Krasulina.

## 5 Proof of the Main Result

The main result of this paper is given by Theorem 1, which can then be applied to any algorithm that (implicitly or explicitly) involves an iteration of the form (8). We develop a proof of this result in this section, which—similar to the approach taken in (Balsubramani et al., 2013) for the analysis of Krasulina's method—consists of characterizing the behavior of D-Krasulina in three different algorithmic epochs. The result concerning the *initial epoch* is described in terms of Theorem 3 in the following, the behavior of the *intermediate epoch*, which comprises multiple *sub-epochs*, is described through Theorem 4, while the behavior of D-Krasulina in the *final epoch* is captured through a formal proof of Theorem 1 at the end of this section.

Before proceeding, recall that our result requires the existence of a sequence $(\Omega_t')_{t \in \mathbb{Z}_+}$ of nested sample spaces that are defined in terms of a sequence of pairs $(t_0 \equiv 0, \epsilon_0), (t_1, \epsilon_1), \ldots, (t_J, \epsilon_J)$. Our analysis of the initial epoch involves showing that for the step size $\gamma_t$ chosen as in Theorem 1, the error for all $t \geq 0$ will be less than $(1 - \epsilon_0)$ with high probability for some constant $\epsilon_0$. We then define the remaining $\epsilon_j$'s as $\epsilon_j = 2^j \epsilon_0, j = 1, \ldots, J$, where $J$ is defined as the smallest integer satisfying $\epsilon_J \geq 1/2$. Our analysis

in the intermediate epoch then focuses on establishing lower bounds on the number of iterations $t_j$ for which D-Krasulina is guaranteed to have the error less than $1 - \epsilon_j$ with high probability. Stated differently, the intermediate epoch characterizes the sub-epochs $\{1 + t_{j-1}, t_j\}$ during which the error is guaranteed to decrease from $(1 - \epsilon_{j-1})$ to $(1 - \epsilon_j)$ with high probability.

## 5.1 Initial Epoch

Our goal for the initial epoch is to show that if we pick the step size appropriately, i.e., we set $L$ to be large enough (cf. (13)), then the error, $\Psi_t$, will not exceed a certain value with high probability. This is formally stated in the following result.

**Theorem 3.** *Fix any $\delta \in (0,1)$, define $\epsilon \in (0,1)$ as $\epsilon := \delta^2/8e$, and let*

$$L \geq \frac{8dr^4 \max(1, c^2)}{\epsilon} \ln \frac{4}{\delta} + \frac{8d^2\sigma_N^2 \max(1, c^2)}{\epsilon^2} \ln \frac{4}{\delta}. \tag{20}$$

*Then, if Assumptions* [**A1**] *and* [**A2**] *hold and we choose step size to be $\gamma_t = c/(L + t)$, we have*

$$\mathbb{P}\left( \sup_{t \geq 0} \Psi_t \geq 1 - \frac{\epsilon}{d} \right) \leq \sqrt{2e\epsilon} \equiv \frac{\delta}{2}. \tag{21}$$

In order to prove Theorem 3 we need several helping lemmas that are stated in the following. We only provide lemma statements in this section and move the proofs to Appendix A. We start by writing the recursion of error metric $\Psi_t$ in the following lemma.

**Lemma 1.** *Defining a scalar random variable*

$$z_t := 2\gamma_t \frac{(\mathbf{v}_{t-1}^{\mathsf{T}}\mathbf{q}^*)(\boldsymbol{\xi}_t^{\mathsf{T}}\mathbf{q}^*)}{\|\mathbf{v}_{t-1}\|_2^2}, \tag{22}$$

*we get the following recursion:*

*(i)* $\Psi_t \leq \Psi_{t-1} + 4\gamma_t^2 \left( \left\| \frac{1}{N} \sum_{i=1}^N \mathbf{A}_{i,t} - \boldsymbol{\Sigma} \right\|_F^2 + \lambda_1^2 \Psi_{t-1} \right) - z_t$, *and*

*(ii)* $\Psi_t \leq \Psi_{t-1} + \gamma_t^2 r^4 - z_t$.

*Proof.* See Appendix A.1. □

Part $(i)$ of this lemma will be used to analyze the algorithm in the final epoch for proof of Theorem 1, while Part $(ii)$ will be used to prove Theorem 3 for this initial epoch and Theorem 4 for the intermediate phase.

Next we will bound the moment generating function of $\Psi_t$ conditioned on $\mathcal{F}_{t-1}$ (Definition 3). For this, we need an upper bound on conditional variance of $z_t$, which is given below.

**Lemma 2.** *The conditional variance of the random variable $z_t$ is given by*

$$\mathbb{E}\{(z_t - \mathbb{E}\{z_t\})^2 | \mathcal{F}_{t-1}\} \leq 16\gamma_t^2 \sigma_N^2. \tag{23}$$

*Proof.* See Appendix A.2. □

Using this upper bound on conditional variance of $z_t$ we can now upper bound the conditional moment generating function of $\Psi_t$. In order to simplify notation, much of our discussion in the following will revolve around the moment generating function with parameter $s \in \mathbb{S} := \{d/4\epsilon, (2/\epsilon_0) \ln(4/\delta)\}$. Note, however, that similar results can be derived for any positive-valued parameter $s \in \mathbb{R}$.

**Lemma 3.** *The conditional moment generating function of $\Psi_t$ for $s \in \mathbb{S}$ is upper bounded as*

$$\mathbb{E}\{\exp(s\Psi_t) | \mathcal{F}_{t-1}\} \leq \exp\left( s\Psi_{t-1} - s\mathbb{E}\{z_t | \mathcal{F}_{t-1}\} + s\gamma_t^2 r^4 + s^2\gamma_t^2\sigma_N^2 \right). \tag{24}$$

*Proof.* See Appendix A.3. □

Note that this result is similar to (Balsubramani et al., 2013, Lemma 2.3) with the difference being that the last term here is sample variance, $\sigma_N^2$, as opposed to upper bound on input $\|\mathbf{x}_{t'}\|_2 \leq r$ in (Balsubramani et al., 2013, Lemma 2.3). This difference prompts changes in next steps of the analysis of D-Krasulina and it also enables us to characterize improvements in convergence rate of Krasulina's method using iterations of the form (8).

We are now ready to prove the statement of Theorem 3, which is based on Lemma 1 and 3.

*Proof of Theorem 3.* We start by constructing a supermartingale from sequence of errors $\Psi_t$. First, restricting ourselves to $s \in \mathbb{S}$, we define quantities

$$\beta_t := \gamma_t^2 r^4, \quad \zeta_t := s\gamma_t^2\sigma_N^2, \quad \tau_t := \sum_{l>t}(\beta_l + \zeta_l), \quad \text{and} \quad M_t := \exp\left(s\Psi_t + s\tau_t\right).$$

Now, taking expectation of $M_t$ conditioned on the filtration $\mathcal{F}_{t-1}$ we get

$$\mathbb{E}\{M_t|\mathcal{F}_{t-1}\} = \mathbb{E}\{\exp\left(s\Psi_t\right)|\mathcal{F}_{t-1}\}\exp\left(s\tau_t\right) \overset{(a)}{\leq} \exp\left(s\Psi_{t-1} + s\beta_t + s\zeta_t + s\tau_t\right)$$
$$= \exp\left(s\Psi_{t-1} + s\tau_{t-1}\right) = M_{t-1}.$$

Here, $(a)$ is due to Lemma 3 and using the fact that $\mathbb{E}\{z_t|\mathcal{F}_{t-1}\} \geq 0$ (Balsubramani et al., 2013, Theorem 2.1). These calculations show that the sequence $\{M_t\}$ forms a supermartingale. Using sequence $M_t$, we can now use Doob's martingale inequality (Durrett, 2010, pg. 231) to show that $\Psi_t$ will be bounded away from 1 with high probability. Specifically, for any $\Delta \in (0, 1)$, we have

$$\mathbb{P}\left(\sup_{t\geq 0}\Psi_t \geq \Delta\right) \leq \mathbb{P}\left(\sup_{t\geq 0}\Psi_t + \tau_t \geq \Delta\right) = \mathbb{P}\left(\sup_{t\geq 0}\exp\left(s\Psi_t + s\tau_t\right) \geq e^{s\Delta}\right)$$

$$= \mathbb{P}\left(\sup_{t\geq 0}M_t \geq e^{s\Delta}\right) \leq \frac{\mathbb{E}\{M_0\}}{e^{s\Delta}} = \exp\left(-s(\Delta - \tau_0)\right)\mathbb{E}\{e^{s\Psi_0}\}.$$

Substituting $\Delta = 1 - \epsilon/d$ and using (Balsubramani et al., 2013, Lemma 2.5) to bound $\mathbb{E}e^{s\Psi_0}$ we get

$$\mathbb{P}\left(\sup_{t\geq 0}\Psi_t \geq 1 - \frac{\epsilon}{d}\right) \leq \exp\left(-s(1 - (\epsilon/d) - \tau_0)\right)e^s\sqrt{\frac{d}{2s}}. \tag{25}$$

Next we need to bound $\sum_{l>0}\beta_l$ and $\sum_{l>0}\zeta_l$. First we get

$$\sum_{l>0}\beta_l = \sum_{l>0}\gamma_l^2 r^4 = r^4\sum_{l>0}\gamma_l^2 = r^4\sum_{l>0}\frac{c^2}{(l+L)^2} \leq \frac{r^4 c^2}{L}. \tag{26}$$

Again using a similar procedure we get

$$\sum_{l>0}\zeta_l \leq \frac{s\sigma_N^2 c^2}{L}. \tag{27}$$

Combining (26) and (27), along with the definition of $\tau_t$ at the beginning, we get

$$\tau_0 \leq \frac{c^2}{L}\left(r^4 + s\sigma_N^2\right). \tag{28}$$

Now using the lower bound on $L$, we get $\tau_0 \leq \epsilon/d$ for $s = d/4\epsilon$ as shown in Proposition 4 in Appendix D. Substituting this in (25) we get

$$\mathbb{P}\left(\sup_{t\geq 0}\Psi_t \geq 1 - \frac{\epsilon}{d}\right) \leq \exp\left(-s(1 - \epsilon/d - \epsilon/d)\right)e^s\sqrt{\frac{d}{2s}} = \exp\left(2s\epsilon/d\right)\sqrt{\frac{d}{2s}}.$$

Finally, substituting $s = d/4\epsilon$, we get $\mathbb{P}\left(\sup_{t\geq 0}\Psi_t \geq 1 - \frac{\epsilon}{d}\right) \leq \sqrt{2e\epsilon}$. □

## 5.2 Intermediate Epoch

In Theorem 3 we have shown that if we choose $L$ such that it satisfies the lower bound given in Theorem 3 then we have error $\Psi_t$ greater than $1 - \epsilon_0$ (here, $\epsilon_0 = \delta^2/8ed$) with probability $\delta$. Next, our aim is to show that if we perform enough iterations $t_J$ of D-Krasulina then for any $t \geq t_J$ the error in the iterate will be bounded by $\Psi_t \leq 1/2$ with high probability. In order to prove this, we divide our analysis into different sub-epochs that are indexed by $j \in \{1, \ldots, J\}$. Starting from $1 - \epsilon_0$, we provide a lower bound on the number of iterations $t_j$ such that we progressively increase $\epsilon_j$ in each sub-epoch until we reach $\epsilon_J$.

**Theorem 4.** *Fix any $\delta \in (0, 1)$ and pick $c := c_0/2(\lambda_1 - \lambda_2)$ for any $c_0 > 2$. Next, let the number of processing nodes $N > 1$, the parameter $L \geq \frac{8r^4 \max(1, c^2)}{\epsilon_0} \ln \frac{4}{\delta} + \frac{8\sigma_N^2 \max(1, c^2)}{\epsilon_0^2} \ln \frac{4}{\delta}$, and the step size $\gamma_t := c/(L+t)$. Finally, select a schedule $(0, \epsilon_0), (t_1, \epsilon_1), \ldots, (t_J, \epsilon_J)$ such that the following conditions are satisfied:*

[**C1**] $\epsilon_0 = \frac{\delta^2}{8ed}$, $\frac{3}{2}\epsilon_j \leq \epsilon_{j+1} \leq 2\epsilon_j$ *for* $0 \leq j < J$, *and* $\epsilon_{J-1} \leq \frac{1}{4}$, *and*

[**C2**] $\left(t_{j+1} + L + 1\right) \geq e^{5/c_0}\left(t_j + L + 1\right)$ *for* $0 \leq j < J$.

*Then* $\mathbb{P}\left(\cap_{t>0}\Omega_t'\right) \geq 1 - \delta$.

In order to prove this theorem, we need Lemmas 4–7, which are stated as follows.

**Lemma 4.** *For $t > t_j$, the moment generating function of $\Psi_t$ for $s \in \mathbb{S}$ conditioned on $\Omega_t'$ satisfies*

$$\mathbb{E}_t\left\{e^{s\Psi_t}\right\} \leq \exp\left(s\left(\Psi_{t-1}\left(1 - \frac{c_0\epsilon_j}{t+L}\right) + \frac{c^2r^4}{(t+L)^2} + \frac{sc^2\sigma_N^2}{(t+L)^2}\right)\right).$$

*Proof.* See Appendix B.1. □

**Lemma 5.** *For $t > t_j$ and $s \in \mathbb{S}$, we have*

$$\mathbb{E}_t\{e^{s\Psi_t}\} \leq \exp\left(s(1 - \epsilon_j)\left(\frac{t_j + L + 1}{t + L + 1}\right)^{c_0\epsilon_j} + \left(sc^2r^4 + s^2c^2\sigma_N^2\right)\left(\frac{1}{t_j + L} - \frac{1}{t + L}\right)\right). \tag{29}$$

*Proof.* See Appendix B.2. □

Using Lemma 5, our next result deals with a specific value of $t$, namely, $t = t_{j+1}$.

**Lemma 6.** *Suppose Conditions [**C1**]–[**C2**] are satisfied. Then for $0 \leq j < J$ and $s \in \mathbb{S}$, we get*

$$\mathbb{E}_{t_{j+1}}\left\{e^{s\Psi_{t_{j+1}}}\right\} \leq \exp\left(s(1 - \epsilon_{j+1}) - s\epsilon_j + \left(sc^2r^4 + s^2c^2\sigma_N^2\right)\left(\frac{1}{t_j + L} - \frac{1}{t_{j+1} + L}\right)\right).$$

*Proof.* See Appendix B.3. □

**Lemma 7.** *Suppose Conditions [**C1**]–[**C2**] are satisfied. Then picking any $0 < \delta < 1$, we have*

$$\sum_{j=1}^{J} \mathbb{P}_{t_j}\left(\sup_{t \geq t_j} \Psi_t > 1 - \epsilon_j\right) \leq \frac{\delta}{2}.$$

*Proof.* See Appendix B.4. □

*Proof. (Proof of Theorem 4)* Using results from Lemma 7 and Theorem 3 and applying union bound, we get the statement of Theorem 4. □

### 5.3 Final Epoch

Now that we have shown that $\Psi_t \leq 1/2$ with probability $1 - \delta$ for all $t \geq t_J$, we characterize in the final epoch how $\Psi_t$ decreases further as a function of algorithmic iterations. The following result captures the rate at which $\Psi_t$ decreases during this final epoch.

**Lemma 8.** *For any $t > t_J$ and $c := c_0/(\lambda_1 - \lambda_2)$, the (conditional) expected error in $\Psi_t$ is given by*

$$
\mathbb{E}_t\{\Psi_t\} \leq \left(1 + \frac{c_0^2 \lambda_1^2}{2(t+L)^2(\lambda_1 - \lambda_2)^2} - \frac{c_0}{2(t+L)}\right)\mathbb{E}_{t-1}\{\Psi_{t-1}\} + \frac{4c^2\sigma_N^2}{(t+L)^2}.
$$

*Proof.* See Appendix C. □

We are now ready to prove our main result, which is given by Theorem 1.

*Proof.* (*Proof of Theorem 1*) Recall the definitions of the sub-epochs corresponding to the pairs $(t_j, \epsilon_j)'s$ that satisfy the two conditions in Theorem 4. Following the same procedure as in the proof of (Balsubramani et al., 2013, Theorem 1.1), notice that $J = \log_2\left(1/(2\epsilon_0)\right)$ (since $\epsilon_J = 2\epsilon_{J-1} = \cdots = 2^J\epsilon_0 \Rightarrow 2^J = \epsilon_J/\epsilon_0 = 1/2\epsilon_0$) and therefore Condition [**C2**] implies

$$
t_J + L + 1 = (L+1)\exp\left(\frac{5J}{c_0}\right) = (L+1)\left(\frac{1}{2\epsilon_0}\right)^{5/(c_0 \ln 2)} = (L+1)\left(\frac{4ed}{\delta^2}\right)^{5/(c_0 \ln 2)}. \tag{30}
$$

Defining $a_1 := c_0^2\lambda_1^2/2(\lambda_1 - \lambda_2)^2$, $a_2 := c_0/2$, $b := 4c^2\sigma_N^2$, and using Lemma 8 for $t > t_J$, we have

$$
\mathbb{E}_t\{\Psi_t\} \leq \left(1 + \frac{a_1}{(t+L)^2} - \frac{a_2}{t+L}\right)\mathbb{E}_{t-1}\{\Psi_{t-1}\} + \frac{b}{(t+L)^2}.
$$

Now using Proposition 1 from Appendix C with $c_0 > 2$, we get

$$
\begin{aligned}
\mathbb{E}_t\{\Psi_t\} &\leq \left(\frac{t_J + L + 1}{t + L + 1}\right)^{\frac{c_0}{2}}\exp\left(\frac{a_1}{t_J + L + 1}\right)\mathbb{E}_{t_J}\{\Psi_{t_J}\} \\
&\quad + \frac{b}{a_2 - 1}\left(1 + \frac{1}{t_J + L + 1}\right)^2\exp\left(\frac{a_1}{t_J + L + 1}\right)\frac{1}{t + L + 1} \\
&\overset{(a)}{\leq} \frac{1}{2}\left(\frac{L+1}{t + L + 1}\right)^{\frac{c_0}{2}}\left(\frac{4ed}{\delta^2}\right)^{\frac{5a_2}{(c_0 \ln 2)}}\exp\left(\frac{a_1}{t_J + L + 1}\right) \\
&\quad + \frac{b}{a_2 - 1}\exp\left(\frac{2}{t_J + L + 1}\right)\exp\left(\frac{a_1}{t_J + L + 1}\right)\frac{1}{t + L + 1} \\
&= \frac{1}{2}\left(\frac{L+1}{t + L + 1}\right)^{\frac{c_0}{2}}\left(\frac{4ed}{\delta^2}\right)^{\frac{5}{(2 \ln 2)}}\exp\left(\frac{a_1}{(L+1)(4ed/\delta^2)^{(5/2 \ln 2)}}\right) \\
&\quad + \frac{8c^2\sigma_N^2}{c_0 - 2}\exp\left(\frac{2 + a_1}{(L+1)(4ed/\delta^2)^{(5/2 \ln 2)}}\right)\frac{1}{(t + L + 1)}.
\end{aligned}
$$

Here, the inequality in $(a)$ is due to (30) and we have also used the fact that $(1 + x)^a \leq \exp(ax)$ for $x < 1$. In addition, since $(4ed/\delta^2)^{(5/2 \ln 2)} \geq 1$, we get

$$
\begin{aligned}
\mathbb{E}_t\{\Psi_t\} &\leq \frac{1}{2}\left(\frac{L+1}{t + L + 1}\right)^{\frac{c_0}{2}}\left(\frac{4ed}{\delta^2}\right)^{\frac{5}{(2 \ln 2)}}\exp\left(\frac{a_1}{L+1}\right) + \frac{8c^2\sigma_N^2}{c_0 - 2}\exp\left(\frac{a_1 + 2}{L+1}\right)\frac{1}{(t+1)} \\
&\leq \frac{1}{2}\left(\frac{L+1}{t + L + 1}\right)^{\frac{c_0}{2}}\left(\frac{4ed}{\delta^2}\right)^{\frac{5}{(2 \ln 2)}}e^{a_1/L} + \frac{8c^2\sigma_N^2 e^{(a_1+2)/L}}{c_0 - 2}\frac{1}{(t + L + 1)} \\
&= C_1\left(\frac{L+1}{t + L + 1}\right)^{\frac{c_0}{2}} + C_2\left(\frac{\sigma_N^2}{t + L + 1}\right). \tag{31}
\end{aligned}
$$

This completes the proof of the theorem. □

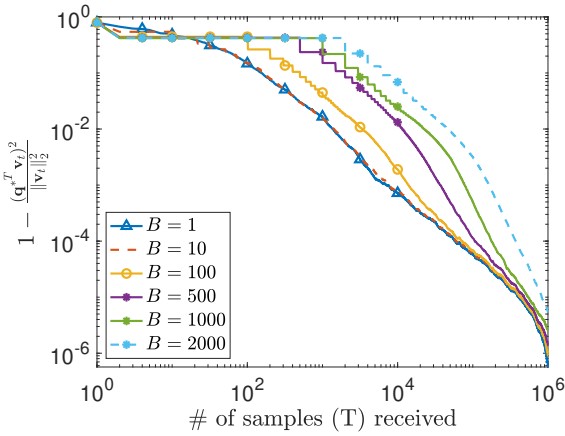 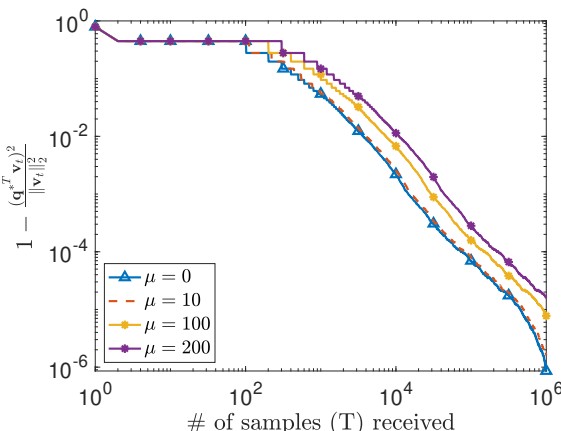

(a) Impact of the mini-batch size on the convergence rate of DM-Krasulina for the resourceful regime. Note that the $B = 1$ plot is effectively Krasulina's method.

(b) Performance of DM-Krasulina in a resource-constrained regime (i.e., $N < \frac{bR_cR_s}{R_p(bR_c - R_s)}$), which causes loss of $\mu$ samples per iteration; here, $(N, B) = (10, 100)$.

Figure 2: Convergence behavior of DM-Krasulina for the case of synthetic data under two scenarios: (a) No data loss ($\mu = 0$) and (b) loss of $\mu > 0$ samples per algorithmic iteration.

## 6 Numerical Results

In this section, we utilize numerical experiments to validate the theoretical findings of this work in terms of the ability of implicit/explicit mini-batched variants of the original Krasulina's method (Krasulina, 1969) to estimate the top eigenvector of a covariance matrix from (fast) streaming data. Instead of repeating the same set of experiments for the original Krasulina's method, D-Krasulina, and DM-Krasulina, we present our results that are parameterized by the network-wide mini-batch size $B \in \{1\} \bigcup \{bN : b \in \mathbb{Z}_+\}$ that appears in DM-Krasulina. This is because $B = 1$ trivially corresponds to the original Krasulina's iterations, while $B = N$ corresponds to iterations that characterize D-Krasulina.

Our goals for the numerical experiments are threefold: ($i$) showing the impact of (implicit/explicit) mini-batching on the convergence rate of DM-Krasulina, ($ii$) establishing robustness of DM-Krasulina against the loss of $\mu > 0$ samples per iteration for the case when $N < \frac{bR_cR_s}{R_p(bR_c - R_s)}$, and ($iii$) experimental validation for scaling of the convergence rate in terms of the problem parameters as predicted by our theoretical findings, namely, eigengap ($\lambda_1 - \lambda_2$), dimensionality ($d$), and upper bound on input samples ($\|\mathbf{x}_{t'}\|_2 \leq r$). In the following, we report results of experiments on both synthetic and real-world data to highlight these points. Since the main purpose is to corroborate the scaling behaviors within the main results—and not to investigate additional system-related issues concerned with large-scale implementations—the real-world datasets are chosen to facilitate their processing on low-cost compute machines.

### 6.1 Experiments on Synthetic Data

In the following experiments we generate $T = 10^6$ samples from some probability distribution (specified for each experiment later) and for each experiment we perform 200 Monte-Carlo trials. In all the experiments in the following we use step size of the form $\gamma_t = c/t$. We performed experiments with multiple values of $c$ and here we are reporting the results for the value of $c$ which achieves the best convergence rate. Further details about each experiment are provided in the following sections.

#### 6.1.1 Impact of mini-batch size on the performance of DM-Krasulina

For a covariance matrix $\mathbf{\Sigma} \in \mathbb{R}^{5 \times 5}$ with $\lambda_1 = 1$ and eigengap $\lambda_1 - \lambda_2 = 0.2$, we generate $T = 10^6$ samples from $\mathcal{N}(\mathbf{0}, \mathbf{\Sigma})$ distribution. The first set of experiments here deals with the resourceful regime, i.e., $N \geq \frac{bR_cR_s}{R_p(bR_c - R_s)}$, with mini-batches of sizes $B \in \{1, 10, 100, 500, 1000, 2000\}$. Note that these values of $B$ can be

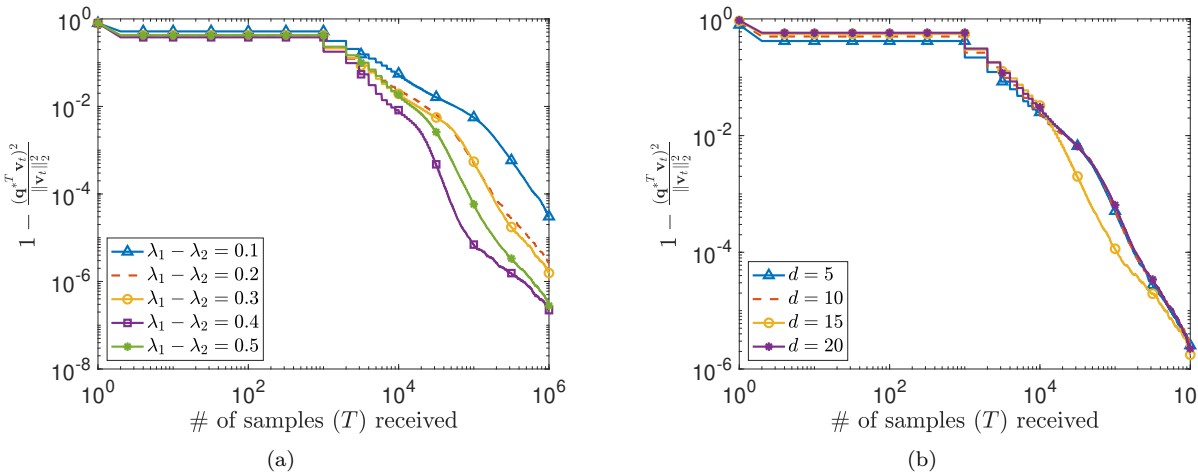

Figure 3: Understanding the impact of (a) eigengap ($\lambda_1 - \lambda_2$) and (b) dimensionality $d$ on the convergence behavior of DM-Krasulina, corresponding to $B = 1000$ and $\mu = 0$.

factored into any positive integers $b$ and $N$ as long as the condition $N \geq \frac{bR_cR_s}{R_p(bR_c-R_s)}$ that is governed by the application scenario and the physical system is satisfied. It is, therefore, unnecessary to specify $b$ and $N$ for these experiments, whose results are shown in Figure 2(a). These results are obtained for step-size parameter $c \in \{70, 80, 80, 90, 110, 100\}$, which are the values of $c$ resulting in the best convergence rate. As predicted by Corollary 1, we can see that after $T/B$ iterations of DM-Krasulina, the error $\Psi_{T/B}$ is on the order of $O(1/T)$ for $B \in \{1, 10, 100, 500, 1000\}$, while for $B = 2000$, the error $\Psi_{T/B}$ is not optimal anymore.

Next, we demonstrate the performance of DM-Krasulina for resource constrained settings, i.e., $N < \frac{bR_sR_c}{R_p(bR_c-R_s)}$, which causes the algorithm to discard $\mu := \left( \frac{bR_s}{R_p} + \frac{NR_s}{R_c} \right) - B$ samples per iteration. Using the same data generation setup as before, we run DM-Krasulina for a network of 10 nodes ($N = 10$) with network-wide mini-batch of size $B = 100$ (i.e., $b = 10$). We consider different mismatch factors between streaming, processing, and communication rates in this experiment, which result in the number of samples being discarded as $\mu \in \{0, 10, 100, 200\}$. The results are plotted in Figure 2(b), which shows that the error $\Psi_{T/(B+\mu)}$ for $\mu = 10$ is comparable to that for $\mu = 0$, but the error for $\mu = 200$ is an order of magnitude worse than the nominal error.

### 6.1.2 Impact of the eigengap on the performance of DM-Krasulina

For this set of experiments, we again generate data in $\mathbb{R}^5$ from a normal distribution $\mathcal{N}(\mathbf{0}, \mathbf{\Sigma})$, where the covariance matrix $\mathbf{\Sigma}$ has the largest eigenvalue $\lambda_1 = 1$. We then vary the remaining eigenvalues to ensure an eigengap that takes values from the set $\{0.1, 0.2, 0.3, 0.4, 0.5\}$. The corresponding values of $c$ that give the best convergence rate for each unique eigengap satisfy $c \in \{180, 110, 90, 70, 60\}$. The final results for these experiments are plotted in Figure 3(a) for the case of $B = 1000$ and $\mu = 0$. These results establish that the final gap in error after observing $T = 10^6$ data samples is indeed on the order of $O(1/(\lambda_1 - \lambda_2)^2)$, as suggested by the theoretical analysis.

### 6.1.3 Impact of dimensionality on the performance of DM-Krasulina

For this set of experiments, we generate data in $\mathbb{R}^d$ from a normal distribution $\mathcal{N}(\mathbf{0}, \mathbf{\Sigma})$ whose dimensionality is varied such that $d \in \{5, 10, 15, 20\}$. In addition, we fix the largest eigenvalue of $\mathbf{\Sigma}$ to be $\lambda_1 = 1$ and its eigengap to be 0.2. The values of $c$ corresponding to each unique value of $d$ that provide the best convergence rate in these experiments satisfy $\{110, 110, 100, 100\}$; contrary to our theoretical analysis, this seems to suggest that the optimal step-size sequence does not have a strong dependence on $d$, at least for small values of $d$. We also plot the potential function for each $d$ as a function of the number of received samples in Figure 3(b) for the case of $B = 1000$ and $\mu = 0$. Once again, we observe little dependence of

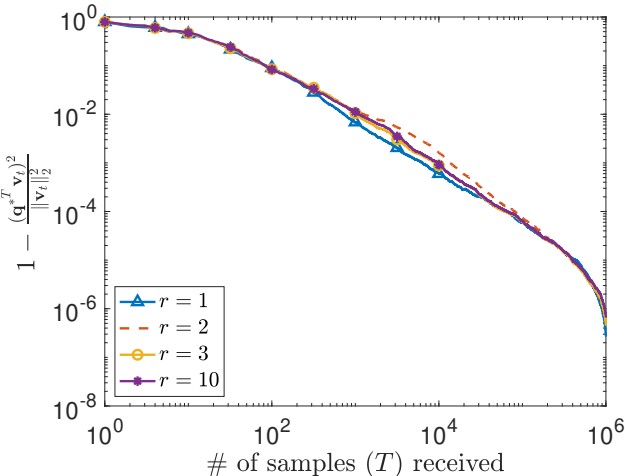

Figure 4: Performance of DM-Krasulina for varying upper bound on the norm of the streaming data.

the performance of DM-Krasulina on $d$. Both these observations suggest that our theoretical analysis is not tight in terms of its dependence on dimensionality $d$ of the streaming data.

### 6.1.4 Impact of upper bound on the performance of DM-Krasulina

In order to understand the impact of the upper bound $\|\mathbf{x}_{t'}\|_2 \leq r$ on the convergence behavior of DM-Krasulina, we generate $\mathbf{x}_{t'} \in \mathbb{R}^5$ as $\mathbf{x}_{t'} = \mathbf{C}\mathbf{u}_{t'}$ with $\mathbf{u}_{t'} \in \mathbb{R}^5$ having independent entries drawn from uniform distribution $\mathcal{U}(-a, a)$ and $\mathbf{C}$ chosen to ensure an eigengap of 0.2 for the covariance matrix. As we vary the value of $a$ within the set $\{1, 2, 3, 10\}$, we generate four different datasets of $T = 10^6$ samples for which the resulting $r \in \{1.45, 2.9, 4.5, 14.5\}$. The values of $c$ that provide best convergence for these values of $r$ satisfy $c \in \{8, 2, 1, 0.08\}$. The final set of results are displayed in Figure 4 for $B = 1$ and $\mu = 0$. It can be seen from this figure that changing $r$ does not affect the convergence behavior of DM-Krasulina. This behavior can be explained by noticing that the parameter $r$ appears in our convergence results in terms of a lower bound on $L$ (cf. (13)) and within the non-dominant term in the error bound. The dependence of $L$ on the parameter $r$ is already being reflected here in our choice of the step-size parameter $c$ that results in the best convergence result. In addition, we hypothesize that the non-dominant error term in our experiments, compared to the dominant one, is significantly small that it masks the dependence of the final error on $r$.

### 6.2 Experiments on Real-world Datasets

In this section, we evaluate the performance of DM-Krasulina on two real-world datasets, namely, the MNIST dataset (LeCun, 1998) and the Higgs dataset (Baldi et al., 2014). The MNIST dataset corresponds to $d = 784$ and has a total of $T = 6 \times 10^4$ samples, while the Higgs dataset is $d = 28$ dimensional and comprises $1.1 \times 10^7$ samples. It is worth noting here that since it is straightforward to store all the samples in these datasets at a single machine, one can always solve the 1-PCA problem for these datasets without resorting to the utilization of a distributed streaming framework. Nonetheless, it is still possible to utilize these dataset in a simulated distributed streaming setting in order to highlight the agreement between the scaling behavior predicted by our theoretical results and the scaling behavior observed using real-world datasets; this is indeed the purpose of the following sets of experiments.

Our first set of experiments is for the MNIST dataset, in which we use the step size $\gamma = c/t$ with $c \in \{0.6, 0.9, 1.1, 1.5, 1.6\}$ for network-wide mini-batch sizes $B \in \{1, 10, 100, 300, 1000\}$ in the resourceful regime ($\mu = 0$). The results, which are averaged over 200 random initializations and random shuffling of data, are given in Figure 5(a). It can be seen from this figure that the final error relatively stays the same as $B$ increases from 1 to 100, but it starts getting affected significantly as the network-wide mini-batch size is further increased to $B = 300$ and $B = 1000$. Our second set of experiments for the MNIST dataset corresponds to

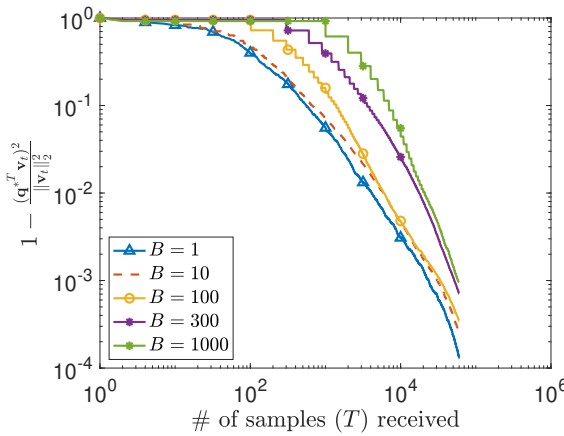

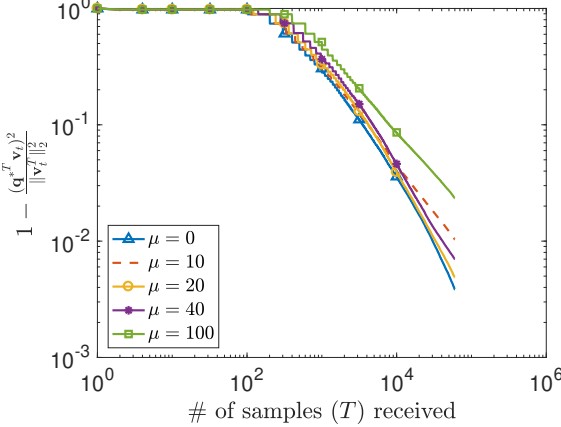

(a) MNIST Data ($\mu = 0$): Impact of network-wide mini-batch size $B$ on the convergence behavior of DM-Krasulina for the resourceful regime.

(b) MNIST Data ($N = 10$; $B = 100$): Convergence behavior of DM-Krasulina in a resource-constrained regime, which causes loss of $\mu$ samples per iteration.

Figure 5: Performance of DM-Krasulina for the MNIST dataset under two scenarios: (a) No data loss ($\mu = 0$) and (b) loss of $\mu > 0$ samples per algorithmic iteration.

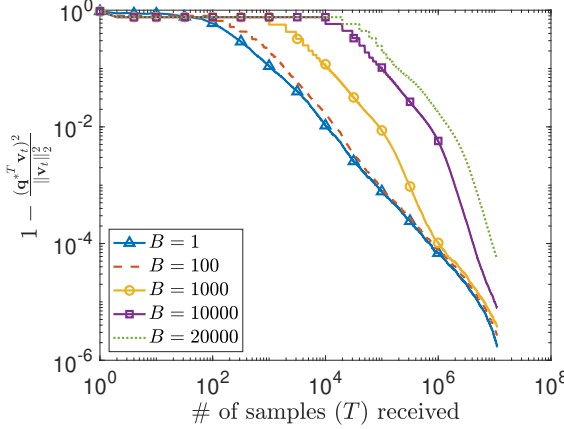

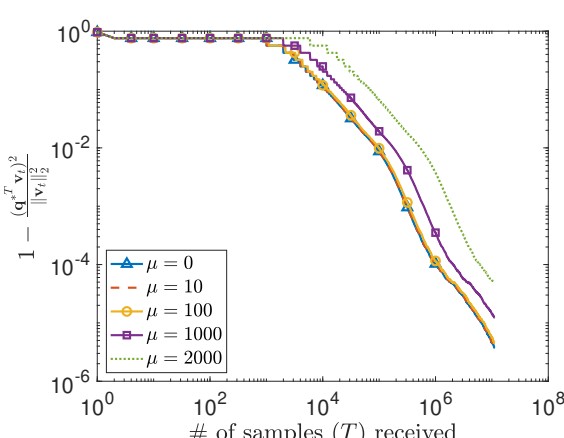

(a) Higgs Data ($\mu = 0$): Impact of network-wide mini-batch size $B$ on the convergence behavior of DM-Krasulina for the resourceful regime.

(b) Higgs Data ($N = 10$; $B = 1000$): Convergence behavior of DM-Krasulina in a resource-constrained regime, which causes loss of $\mu$ samples per iteration.

Figure 6: Performance of DM-Krasulina for the Higgs dataset under two scenarios: (a) No data loss ($\mu = 0$) and (b) loss of $\mu > 0$ samples per algorithmic iteration.

the resource-constrained regime with $(N, B) = (10, 100)$ and step-size parameter $c \in \{0.6, 0.9, 1.1, 1.5, 1.6\}$ for the number of discarded samples $\mu \in \{0, 10, 20, 40, 100\}$. The results, averaged over 200 trials and given in Figure 5(b), show that the system can tolerate loss of some data samples per iteration without significant increase in the final error; the increase in error, however, becomes noticeable as $\mu$ approaches $B$. Both these observations are in line with the insights of our theoretical analysis.

We now turn our attention to the Higgs dataset. Our results for this dataset, averaged over 200 trials and using $c = 0.07$, for the resourceful and resource-constrained settings are given in Figure 6(a) and Figure 6(b), respectively. In the former setting, corresponding to $B \in \{1, 10^2, 10^3, 10^4, 2 \times 10^4\}$, we once again see that the error relatively stays the same for values of $B$ that are significantly smaller than $T$; in particular, since $T$ for the Higgs dataset is larger than for the MNIST dataset, it can accommodate a larger value of $B$ without significant loss in performance. In the latter resource-constrained setting, corresponding to $N = 10$,

$B = 1000$ and $\mu \in \{0, 10, 100, 1000, 2000\}$, we similarly observe that small (relative to $B$) values of $\mu$ do not impact the performance of DM-Krasulina in a significant manner. Once again, these results corroborate our research findings.

## 7  Conclusion

In this paper, we studied the problem of estimating the principal eigenvector of a covariance matrix from independent and identically distributed data samples. Our particular focus in here was developing and analyzing two variants, termed D-Krasulina and DM-Krasulina, of a classical stochastic algorithm that can estimate the top eigenvector in a near-optimal fashion from fast streaming data that overwhelms the processing capabilities of a single processor. Unlike the classical algorithm that must discard data samples in high-rate streaming settings, and thus sacrifice the convergence rate, the proposed algorithms manage the high-rate streaming data by trading off processing capabilities with computational resources and communications infrastructure. Specifically, both D-Krasulina and DM-Krasulina virtually slow down the rate of streaming data by spreading the processing of data samples across of a network of processing nodes. In addition, DM-Krasulina can overcome slower communication links and/or lack of sufficient number of processing nodes through a network-wide mini-batching strategy, coupled with discarding of a small number of data samples per iteration.

Our theoretical analysis, which fundamentally required a characterization of the error incurred by the proposed algorithms as a function of the variance of the sample covariance matrix, substantially improved the variance-agnostic analysis in (Balsubramani et al., 2013) and established the conditions under which near-optimal convergence rate is achievable in the fast streaming setting, even when some data samples need to be discarded due to lack of sufficient computational and/or communication resources. We also carried out numerical experiments on both synthetic and real-world data to validate our theoretical findings.

In terms of future work, extension of our algorithmic and analytical framework for estimation of the principal subspace comprising multiple eigenvectors remains an open problem. In addition, tightening our theoretical analysis to better elucidate the role of dimensionality of data in the performance of the proposed algorithmic framework is an interesting problem. Finally, investigation of additional practical issues (e.g., processor failures, variable compute costs, and network coordination costs) concerning processing of data in large-scale systems provides another avenue for future research.

### Funding Acknowledgements

This work has been supported in part by the National Science Foundation under Awards CCF-1907658, OAC-1940074, and CNS-2148104, and by the Army Research Office under Awards W911NF-17-1-0546 and W911NF-21-1-0301.

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

# Appendix A    Proofs of Lemmas for the Initial Epoch

## A.1    Proof of Lemma 1

In order to prove Lemma 1, we first need the following result.

**Lemma 9.** *The second moment of the update vector $\boldsymbol{\xi}_t$ in D-Krasulina is upper bounded as*

$$\mathbb{E}\left\{\frac{\|\boldsymbol{\xi}_t\|_2^2}{\|\mathbf{v}_{t-1}\|_2^2}\right\} \leq \frac{\mathbb{E}\left\{\|\boldsymbol{\xi}_t - \mathbb{E}\boldsymbol{\xi}_t\|_2^2\right\}}{\|\mathbf{v}_{t-1}\|_2^2} + 2\lambda_1^2 \Psi_{t-1}.$$

*Proof.* We start by writing $\mathbb{E}\left\{\|\boldsymbol{\xi}_t - \mathbb{E}\{\boldsymbol{\xi}_t\}\|_2^2\right\}$ in terms of $\mathbb{E}\left\{\|\boldsymbol{\xi}_t\|_2^2\right\}$ as follows:

$$\mathbb{E}\left\{\|\boldsymbol{\xi}_t - \mathbb{E}\{\boldsymbol{\xi}_t\}\|_2^2\right\} = \mathbb{E}\left\{\boldsymbol{\xi}_t^{\mathrm{T}}\boldsymbol{\xi}_t + (\mathbb{E}\{\boldsymbol{\xi}_t\})^{\mathrm{T}}\mathbb{E}\{\boldsymbol{\xi}_t\} - \boldsymbol{\xi}_t^{\mathrm{T}}\mathbb{E}\{\boldsymbol{\xi}_t\} - (\mathbb{E}\{\boldsymbol{\xi}_t\})^{\mathrm{T}}\boldsymbol{\xi}_t\right\}$$

$$= \mathbb{E}\{\|\boldsymbol{\xi}_t\|_2^2\} - \mathbb{E}\{\boldsymbol{\xi}_t^{\mathrm{T}}\}\mathbb{E}\{\boldsymbol{\xi}_t\}.$$

Now defining $C_t := \mathbb{E}\{\boldsymbol{\xi}_t^{\mathrm{T}}\}\mathbb{E}\{\boldsymbol{\xi}_t\}$ and rearranging the above equation, we get

$$\mathbb{E}\{\|\boldsymbol{\xi}_t\|_2^2\} = \mathbb{E}\{\|\boldsymbol{\xi}_t - \mathbb{E}\{\boldsymbol{\xi}_t\}\|_2^2\} + C_t.$$

Next, substituting value of $\boldsymbol{\xi}_t$ from (8) we get

$$\frac{C_t}{\|\mathbf{v}_{t-1}\|_2^2} = \frac{\mathbb{E}\{\boldsymbol{\xi}_t^{\mathrm{T}}\}\mathbb{E}\{\boldsymbol{\xi}_t\}}{\|\mathbf{v}_{t-1}\|_2^2} = \frac{1}{\|\mathbf{v}_{t-1}\|_2^2}\left(\boldsymbol{\Sigma}\mathbf{v}_{t-1} - \frac{\mathbf{v}_{t-1}^{\mathrm{T}}\boldsymbol{\Sigma}\mathbf{v}_{t-1}\mathbf{v}_{t-1}}{\mathbf{v}_{t-1}^{\mathrm{T}}\mathbf{v}_{t-1}}\right)^{\mathrm{T}}\left(\boldsymbol{\Sigma}\mathbf{v}_{t-1} - \frac{\mathbf{v}_{t-1}^{\mathrm{T}}\boldsymbol{\Sigma}\mathbf{v}_{t-1}\mathbf{v}_{t-1}}{\mathbf{v}_{t-1}^{\mathrm{T}}\mathbf{v}_{t-1}}\right)$$

$$= \frac{\mathbf{v}_{t-1}^{\mathrm{T}}\boldsymbol{\Sigma}^2\mathbf{v}_{t-1}}{\|\mathbf{v}_{t-1}\|_2^2} - \left(\frac{\mathbf{v}_{t-1}^{\mathrm{T}}\boldsymbol{\Sigma}\mathbf{v}_{t-1}}{\|\mathbf{v}_{t-1}\|_2^2}\right)^2. \tag{32}$$

Since $\boldsymbol{\Sigma}$ is a positive semi-definite matrix, we can write its eigenvalue decomposition as $\boldsymbol{\Sigma} = \sum_{i=1}^{d}\lambda_i\mathbf{q}_i\mathbf{q}_i^{\mathrm{T}}$, where $\lambda_1 > \lambda_2 \geq \cdots \geq \lambda_d \geq 0$ and $\mathbf{q}_1(\equiv \mathbf{q}^*), \mathbf{q}_2, \ldots, \mathbf{q}_d$ are the eigenvalues and corresponding eigenvectors of $\boldsymbol{\Sigma}$, respectively. It follows that

$$\frac{C_t}{\|\mathbf{v}_{t-1}\|_2^2} = \sum_{i=1}^{d}\lambda_i^2\frac{(\mathbf{v}_{t-1}^{\mathrm{T}}\mathbf{q}_i)^2}{\|\mathbf{v}_{t-1}\|_2^2} - \left(\sum_{i=1}^{d}\lambda_i\frac{(\mathbf{v}_{t-1}^{\mathrm{T}}\mathbf{q}_i)^2}{\|\mathbf{v}_{t-1}\|_2^2}\right)^2$$

$$= \lambda_1^2\frac{(\mathbf{v}_{t-1}^{\mathrm{T}}\mathbf{q}^*)^2}{\|\mathbf{v}_{t-1}\|_2^2} + \sum_{i=2}^{d}\lambda_i^2\frac{(\mathbf{v}_{t-1}^{\mathrm{T}}\mathbf{q}_i)^2}{\|\mathbf{v}_{t-1}\|_2^2} - \left(\lambda_1\frac{(\mathbf{v}_{t-1}^{\mathrm{T}}\mathbf{q}^*)^2}{\|\mathbf{v}_{t-1}\|_2^2} + \sum_{i=2}^{d}\lambda_i\frac{(\mathbf{v}_{t-1}^{\mathrm{T}}\mathbf{q}_i)^2}{\|\mathbf{v}_{t-1}\|_2^2}\right)^2$$

$$\leq \lambda_1^2\frac{(\mathbf{v}_{t-1}^{\mathrm{T}}\mathbf{q}^*)^2}{\|\mathbf{v}_{t-1}\|_2^2} + \lambda_2^2\sum_{i=2}^{d}\frac{(\mathbf{v}_{t-1}^{\mathrm{T}}\mathbf{q}_i)^2}{\|\mathbf{v}_{t-1}\|_2^2} - \lambda_1^2\frac{(\mathbf{v}_{t-1}^{\mathrm{T}}\mathbf{q}^*)^4}{\|\mathbf{v}_{t-1}\|_2^4}$$

$$= \lambda_1^2\frac{(\mathbf{v}_{t-1}^{\mathrm{T}}\mathbf{q}^*)^2}{\|\mathbf{v}_{t-1}\|_2^2}\left(1 - \frac{(\mathbf{v}_{t-1}^{\mathrm{T}}\mathbf{q}^*)^2}{\|\mathbf{v}_{t-1}\|_2^2}\right) + \lambda_2^2\left(1 - \frac{(\mathbf{v}_{t-1}^{\mathrm{T}}\mathbf{q}^*)^2}{\|\mathbf{v}_{t-1}\|_2^2}\right).$$

Finally, we get from definition of $\Psi_{t-1}$ that

$$\frac{C_t}{\|\mathbf{v}_{t-1}\|_2^2} \leq \Psi_{t-1}\big((1-\Psi_{t-1})\lambda_1^2 + \lambda_2^2\big) \leq \Psi_{t-1}\big(\lambda_1^2 + \lambda_2^2\big) \leq 2\lambda_1^2\Psi_{t-1}.$$

This completes the proof of the lemma. □

Using Lemma 9, we can now prove Lemma 1 in the following.

*Proof of Lemma 1.* From (10), we have $\Psi_t = \frac{\|\mathbf{v}_t\|_2^2 - (\mathbf{v}_t^{\mathrm{T}}\mathbf{q}^*)^2}{\|\mathbf{v}_t\|_2^2}$. Substituting $\mathbf{v}_t$ from (8), we get

$$
\begin{aligned}
\Psi_t &= \frac{\|\mathbf{v}_{t-1} + \gamma_t\boldsymbol{\xi}_t\|_2^2 - ((\mathbf{v}_{t-1} + \gamma_t\boldsymbol{\xi}_t)^{\mathrm{T}}\mathbf{q}^*)^2}{\|\mathbf{v}_t\|_2^2} \overset{(a)}{=} \frac{\|\mathbf{v}_{t-1}\|_2^2 + \gamma_t^2\|\boldsymbol{\xi}_t\|_2^2 - ((\mathbf{v}_{t-1} + \gamma_t\boldsymbol{\xi}_t)^{\mathrm{T}}\mathbf{q}^*)^2}{\|\mathbf{v}_t\|_2^2} \\
&\overset{(b)}{\leq} \frac{\|\mathbf{v}_{t-1}\|_2^2 + \gamma_t^2\|\boldsymbol{\xi}_t\|_2^2 - ((\mathbf{v}_{t-1} + \gamma_t\boldsymbol{\xi}_t)^{\mathrm{T}}\mathbf{q}^*)^2}{\|\mathbf{v}_{t-1}\|_2^2} = 1 + \gamma_t^2\frac{\|\boldsymbol{\xi}_t\|_2^2}{\|\mathbf{v}_{t-1}\|_2^2} - \frac{((\mathbf{v}_{t-1} + \gamma_t\boldsymbol{\xi}_t)^{\mathrm{T}}\mathbf{q}^*)^2}{\|\mathbf{v}_{t-1}\|_2^2} \\
&= 1 + \gamma_t^2\frac{\|\boldsymbol{\xi}_t\|_2^2}{\|\mathbf{v}_{t-1}\|_2^2} - \frac{(\mathbf{v}_{t-1}^{\mathrm{T}}\mathbf{q}^*)^2 + \gamma_t^2(\boldsymbol{\xi}_t^{\mathrm{T}}\mathbf{q}^*)^2 + 2\gamma_t(\mathbf{v}_{t-1}^{\mathrm{T}}\mathbf{q}^*)(\boldsymbol{\xi}_t^{\mathrm{T}}\mathbf{q}^*)}{\|\mathbf{v}_{t-1}\|_2^2} \\
&= 1 - \frac{(\mathbf{v}_{t-1}^{\mathrm{T}}\mathbf{q}^*)^2}{\|\mathbf{v}_{t-1}\|_2^2} + \gamma_t^2\frac{\|\boldsymbol{\xi}_t\|_2^2 - (\boldsymbol{\xi}_t^{\mathrm{T}}\mathbf{q}^*)^2}{\|\mathbf{v}_{t-1}\|_2^2} - 2\gamma_t\frac{(\mathbf{v}_{t-1}^{\mathrm{T}}\mathbf{q}^*)(\boldsymbol{\xi}_t^{\mathrm{T}}\mathbf{q}^*)}{\|\mathbf{v}_{t-1}\|_2^2} \\
&= \Psi_{t-1} + \gamma_t^2\frac{\|\boldsymbol{\xi}_t\|_2^2}{\|\mathbf{v}_{t-1}\|_2^2} - 2\gamma_t\frac{(\mathbf{v}_{t-1}^{\mathrm{T}}\mathbf{q}^*)(\boldsymbol{\xi}_t^{\mathrm{T}}\mathbf{q}^*)}{\|\mathbf{v}_{t-1}\|_2^2}.
\end{aligned}
\tag{33}
$$

Here $(a)$ and $(b)$ are due to (Balsubramani et al., 2013, Lemma A.1), where $(a)$ is true because $\mathbf{v}_{t-1}$ is perpendicular to $\boldsymbol{\xi}_t$ and $(b)$ is true because $\|\mathbf{v}_{t-1}\|_2 \leq \|\mathbf{v}_t\|_2$. The second term in the above inequality can be bounded as

$$
\begin{aligned}
\frac{\|\boldsymbol{\xi}_t\|_2^2}{\|\mathbf{v}_{t-1}\|_2^2} &= \frac{\|\boldsymbol{\xi}_t - \mathbb{E}\{\boldsymbol{\xi}_t\}\|_2^2 + \mathbb{E}\{\boldsymbol{\xi}_t^{\mathrm{T}}\}\mathbb{E}\{\boldsymbol{\xi}_t\}}{\|\mathbf{v}_t\|_2^2} \overset{(c)}{\leq} \frac{\mathbb{E}\left\{\|\boldsymbol{\xi}_t - \mathbb{E}\{\boldsymbol{\xi}_t\}\|_2^2\right\}}{\|\mathbf{v}_{t-1}\|_2^2} + 2\lambda_1^2\Psi_{t-1} \\
&= \frac{1}{\|\mathbf{v}_{t-1}\|_2^2}\mathbb{E}\left\{\left\|\frac{1}{N}\sum_{i=1}^{N}\mathbf{A}_{i,t}\mathbf{v}_{t-1} - \frac{1}{\|\mathbf{v}_{t-1}\|_2^2}\left(\mathbf{v}_{t-1}^{\mathrm{T}}\frac{1}{N}\sum_{i=1}^{N}\mathbf{A}_{i,t}\mathbf{v}_{t-1}\mathbf{v}_{t-1}\right)\right.\right. \\
&\qquad\qquad \left.\left. - \mathbb{E}\left\{\frac{1}{N}\sum_{i=1}^{N}\mathbf{A}_{i,t}\mathbf{v}_{t-1} - \frac{1}{\|\mathbf{v}_{t-1}\|_2^2}\left(\mathbf{v}_{t-1}^{\mathrm{T}}\frac{1}{N}\sum_{i=1}^{N}\mathbf{A}_{i,t}\mathbf{v}_{t-1}\mathbf{v}_{t-1}\right)\right\}\right\|_2^2\right\} \\
&= \frac{1}{\|\mathbf{v}_{t-1}\|_2^2}\mathbb{E}\left\{\left\|\frac{1}{N}\sum_{i=1}^{N}\mathbf{A}_{i,t}\mathbf{v}_{t-1} - \frac{1}{\|\mathbf{v}_{t-1}\|_2^2}\left(\mathbf{v}_{t-1}^{\mathrm{T}}\frac{1}{N}\sum_{i=1}^{N}\mathbf{A}_{i,t}\mathbf{v}_{t-1}\mathbf{v}_{t-1}\right)\right.\right. \\
&\qquad\qquad \left.\left. - \boldsymbol{\Sigma}\mathbf{v}_{t-1} + \frac{1}{\|\mathbf{v}_{t-1}\|_2^2}\left(\mathbf{v}_{t-1}^{\mathrm{T}}\boldsymbol{\Sigma}\mathbf{v}_{t-1}\mathbf{v}_{t-1}\right)\right\|_2^2\right\} \\
&= \frac{1}{\|\mathbf{v}_{t-1}\|_2^2}\mathbb{E}\left\{\left\|\left(\frac{1}{N}\sum_{i=1}^{N}\mathbf{A}_{i,t} - \boldsymbol{\Sigma}\right)\mathbf{v}_{t-1} - \frac{1}{\|\mathbf{v}_{t-1}\|_2^2}\mathbf{v}_{t-1}^{\mathrm{T}}\left(\frac{1}{N}\sum_{i=1}^{N}\mathbf{A}_{i,t} - \boldsymbol{\Sigma}\right)\mathbf{v}_{t-1}\mathbf{v}_{t-1}\right\|_2^2\right\} \\
&\leq 4\left\|\frac{1}{N}\sum_{i=1}^{N}\mathbf{A}_{i,t} - \boldsymbol{\Sigma}\right\|_2^2 + 2\lambda_1^2\Psi_{t-1} \leq 4\left\|\frac{1}{N}\sum_{i=1}^{N}\mathbf{A}_{i,t} - \boldsymbol{\Sigma}\right\|_F^2 + 2\lambda_1^2\Psi_{t-1},
\end{aligned}
\tag{34}
$$

where $(c)$ is due to Lemma 9. Substituting (34) in (33) completes the proof of Part $(i)$ of Lemma 1. Next, we prove Part $(ii)$ of the lemma by defining $\widehat{\mathbf{v}}_{t-1} = \mathbf{v}_{t-1}/\|\mathbf{v}_{t-1}\|_2$ and noting that

$$
\begin{aligned}
\frac{\|\boldsymbol{\xi}_t\|_2^2}{\|\mathbf{v}_{t-1}\|_2^2} &= \frac{\|(1/N)\sum_{i=1}^{N}\boldsymbol{\xi}_{i,t}\|_2^2}{\|\mathbf{v}_{t-1}\|_2^2} = \frac{(1/N^2)\|\sum_{i=1}^{N}\boldsymbol{\xi}_{i,t}\|_2^2}{\|\mathbf{v}_{t-1}\|_2^2} \\
&\overset{(d)}{\leq} \frac{(1/N^2)\sum_{i=1}^{N}N\|\boldsymbol{\xi}_{i,t}\|_2^2}{\|\mathbf{v}_{t-1}\|_2^2} = \frac{\sum_{i=1}^{N}(\mathbf{x}_{i,t}^{\mathrm{T}}\mathbf{v}_{t-1})^2\|\mathbf{x}_{i,t} - (\mathbf{x}_{i,t}^{\mathrm{T}}\widehat{\mathbf{v}}_{t-1})\widehat{\mathbf{v}}_{t-1}\|_2^2}{N\|\mathbf{v}_{t-1}\|_2^2} \\
&\leq \frac{1}{N}\sum_{i=1}^{N}\|\mathbf{x}_{i,t}\|_2^2\|\mathbf{x}_{i,t} - (\mathbf{x}_{i,t}^{\mathrm{T}}\widehat{\mathbf{v}}_{t-1})\widehat{\mathbf{v}}_{t-1}\|_2^2 = \frac{1}{N}\sum_{i=1}^{N}\|\mathbf{x}_{i,t}\|_2^2(\|\mathbf{x}_{i,t}\|_2^2 - (\mathbf{x}_{i,t}^{\mathrm{T}}\widehat{\mathbf{v}}_{t-1})^2) \\
&\leq \sum_{i=1}^{N}\frac{\|\mathbf{x}_{i,t}\|_2^4}{N} \leq \max_i\|\mathbf{x}_{i,t}\|_2^4 \leq r^4.
\end{aligned}
\tag{35}
$$

Here, ($d$) is by using Cauchy–Schwartz inquality and the last inequality is due to Assumption [**A1**]. Now substituting this in (33) completes the proof. $\qquad\square$

## A.2 Proof of Lemma 2

We begin by writing

$$\mathbb{E}\{(z_t - \mathbb{E}\{z_t\})^2|\mathcal{F}_{t-1}\} = \mathbb{E}\left\{\left(\frac{2\gamma_t(\mathbf{v}_{t-1}^{\mathrm{T}}\mathbf{q}^*)(\boldsymbol{\xi}_t^{\mathrm{T}}\mathbf{q}^*)}{\|\mathbf{v}_{t-1}\|_2^2} - \mathbb{E}\left\{\frac{2\gamma_t(\mathbf{v}_{t-1}^{\mathrm{T}}\mathbf{q}^*)(\boldsymbol{\xi}_t^{\mathrm{T}}\mathbf{q}^*)}{\|\mathbf{v}_{t-1}\|_2^2}\right\}\right)^2\Bigg|\mathcal{F}_{t-1}\right\}$$

$$= \frac{4\gamma_t^2(\mathbf{v}_{t-1}^{\mathrm{T}}\mathbf{q}^*)^2}{\|\mathbf{v}_{t-1}\|_2^4}\mathbb{E}\left\{\left(\boldsymbol{\xi}_t^{\mathrm{T}}\mathbf{q}^* - \mathbb{E}\{\boldsymbol{\xi}_t^{\mathrm{T}}\mathbf{q}^*\}\right)^2\right\}.$$

Substituting value of $\boldsymbol{\xi}_t$ in this, we get

$$\mathbb{E}\{(z_t - \mathbb{E}\{z_t\})^2|\mathcal{F}_{t-1}\} = \frac{4\gamma_t^2(\mathbf{v}_{t-1}^{\mathrm{T}}\mathbf{q}^*)^2}{\|\mathbf{v}_{t-1}\|_2^4}\mathbb{E}\left\{\left(\left(\frac{1}{N}\sum_{i=1}^N \mathbf{A}_{i,t}\mathbf{v}_{t-1} - \frac{\mathbf{v}_{t-1}^{\mathrm{T}}\frac{1}{N}\sum_{i=1}^N \mathbf{A}_{i,t}\mathbf{v}_{t-1}\mathbf{v}_{t-1}}{\|\mathbf{v}_{t-1}\|_2^2}\right)^{\mathrm{T}}\mathbf{q}^*\right.\right.$$

$$\left.\left.- \mathbb{E}\left\{\left(\frac{1}{N}\sum_{i=1}^N \mathbf{A}_{i,t}\mathbf{v}_{t-1} - \frac{\mathbf{v}_{t-1}^{\mathrm{T}}\frac{1}{N}\sum_{i=1}^N \mathbf{A}_{i,t}\mathbf{v}_{t-1}\mathbf{v}_{t-1}}{\|\mathbf{v}_{t-1}\|_2^2}\right)^{\mathrm{T}}\mathbf{q}^*\right\}\right)^2\right\}$$

$$= \frac{4\gamma_t^2(\mathbf{v}_{t-1}^{\mathrm{T}}\mathbf{q}^*)^2}{\|\mathbf{v}_{t-1}\|_2^4}\mathbb{E}\left\{\left(\left(\frac{1}{N}\sum_{i=1}^N \mathbf{A}_{i,t}\mathbf{v}_{t-1} - \frac{\mathbf{v}_{t-1}^{\mathrm{T}}\frac{1}{N}\sum_{i=1}^N \mathbf{A}_{i,t}\mathbf{v}_{t-1}\mathbf{v}_{t-1}}{\|\mathbf{v}_{t-1}\|_2^2}\right)^{\mathrm{T}}q_1\right.\right.$$

$$\left.\left.- \mathbf{v}_{t-1}^{\mathrm{T}}\mathbb{E}\left\{\frac{1}{N}\sum_{i=1}^N \mathbf{A}_{i,t}\right\}\mathbf{q}^* + \frac{\mathbf{v}_{t-1}^{\mathrm{T}}\mathbf{v}_{t-1}^{\mathrm{T}}\mathbb{E}\{\frac{1}{N}\sum_{i=1}^N \mathbf{A}_{i,t}\}\mathbf{v}_{t-1}}{\|\mathbf{v}_{t-1}\|_2^2}\mathbf{q}^*\right)^2\right\}.$$

Since $\mathbb{E}\left\{\frac{1}{N}\sum_{i=1}^N \mathbf{A}_{i,t}\right\}$ is the covariance matrix $\boldsymbol{\Sigma}$, we get

$\mathbb{E}\{(z_t - \mathbb{E}\{z_t\})^2|\mathcal{F}_{t-1}\}$

$$= \frac{4\gamma_t^2(\mathbf{v}_{t-1}^{\mathrm{T}}\mathbf{q}^*)^2}{\|\mathbf{v}_{t-1}\|_2^4}\mathbb{E}\left\{\left(\left((\frac{1}{N}\sum_{i=1}^N \mathbf{A}_{i,t} - \boldsymbol{\Sigma})\mathbf{v}_{t-1} - \frac{\mathbf{v}_{t-1}^{\mathrm{T}}(\frac{1}{N}\sum_{i=1}^N \mathbf{A}_{i,t} - \boldsymbol{\Sigma})\mathbf{v}_{t-1}\mathbf{v}_{t-1}}{\|\mathbf{v}_{t-1}\|_2^2}\right)^{\mathrm{T}}\mathbf{q}^*\right)^2\right\},$$

$$= \frac{4\gamma_t^2(\mathbf{v}_{t-1}^{\mathrm{T}}\mathbf{q}^*)^2}{\|\mathbf{v}_{t-1}\|_2^4}\mathbb{E}\left\{\left(\left(\mathbf{q}^{*\mathrm{T}}(\frac{1}{N}\sum_{i=1}^N \mathbf{A}_{i,t} - \boldsymbol{\Sigma})\mathbf{v}_{t-1} - \frac{\left(\mathbf{v}_{t-1}^{\mathrm{T}}(\frac{1}{N}\sum_{i=1}^N \mathbf{A}_{i,t} - \boldsymbol{\Sigma})\mathbf{v}_{t-1}\right)\mathbf{q}^{*\mathrm{T}}\mathbf{v}_{t-1}}{\|\mathbf{v}_{t-1}\|_2^2}\right)\right)^2\right\}$$

$$\le \frac{8\gamma_t^2(\mathbf{v}_{t-1}^{\mathrm{T}}\mathbf{q}^*)^2}{\|\mathbf{v}_{t-1}\|_2^4}\mathbb{E}\left\{\left(\mathbf{q}^{*\mathrm{T}}(\frac{1}{N}\sum_{i=1}^N \mathbf{A}_{i,t} - \boldsymbol{\Sigma})\mathbf{v}_{t-1}\right)^2 + \left(\frac{\left(\mathbf{v}_{t-1}^{\mathrm{T}}(\frac{1}{N}\sum_{i=1}^N \mathbf{A}_{i,t} - \boldsymbol{\Sigma})\mathbf{v}_{t-1}\right)\mathbf{q}^{*\mathrm{T}}\mathbf{v}_{t-1}}{\|\mathbf{v}_{t-1}\|_2^2}\right)^2\right\}$$

$$= \frac{8\gamma_t^2(\mathbf{v}_{t-1}^{\mathrm{T}}\mathbf{q}^*)^2}{\|\mathbf{v}_{t-1}\|_2^2}\mathbb{E}\left\{\left(\frac{\mathbf{q}^{*\mathrm{T}}(\frac{1}{N}\sum_{i=1}^N \mathbf{A}_{i,t} - \boldsymbol{\Sigma})\mathbf{v}_{t-1}}{\|\mathbf{v}_{t-1}\|_2}\right)^2 + \left(\frac{\mathbf{v}_{t-1}^{\mathrm{T}}(\frac{1}{N}\sum_{i=1}^N \mathbf{A}_{i,t} - \boldsymbol{\Sigma})\mathbf{v}_{t-1}}{\|\mathbf{v}_{t-1}\|_2^2}\right)^2\left(\frac{\mathbf{q}^{*\mathrm{T}}\mathbf{v}_{t-1}}{\|\mathbf{v}_{t-1}\|_2}\right)^2\right\}$$

$$\le 8\gamma_t^2\mathbb{E}\left\{\left(\frac{\mathbf{q}^{*\mathrm{T}}(\frac{1}{N}\sum_{i=1}^N \mathbf{A}_{i,t} - \boldsymbol{\Sigma})\mathbf{v}_{t-1}}{\|\mathbf{v}_{t-1}\|_2}\right)^2 + \left(\frac{\mathbf{v}_{t-1}^{\mathrm{T}}(\frac{1}{N}\sum_{i=1}^N \mathbf{A}_{i,t} - \boldsymbol{\Sigma})\mathbf{v}_{t-1}}{\|\mathbf{v}_{t-1}\|_2^2}\right)^2\right\}, \qquad (36)$$

where the last inequality in (36) is due to the fact that $\left(\frac{\mathbf{q}^{*\mathrm{T}}\mathbf{v}_{t-1}}{\|\mathbf{v}_{t-1}\|_2}\right)^2 \le 1$. We can see that both the remaining terms in (36) are Rayleigh quotients of matrix $(\boldsymbol{\Sigma} - \frac{1}{N}\sum_{i=1}^N \mathbf{A}_{i,t})$ and hence the largest eigenvalue of $(\boldsymbol{\Sigma} - \frac{1}{N}\sum_{i=1}^N \mathbf{A}_{i,t})$ maximizes both the terms. Using this fact we get

$$\mathbb{E}\{(z_t - \mathbb{E}\{z_t\})^2|\mathcal{F}_{t-1}\} \le 16\gamma_t^2\mathbb{E}\{\|\boldsymbol{\Sigma} - \frac{1}{N}\sum_{i=1}^N \mathbf{A}_{i,t}\|_2^2\} \le 16\gamma_t^2\mathbb{E}\{\|\boldsymbol{\Sigma} - \frac{1}{N}\sum_{i=1}^N \mathbf{A}_{i,t}\|_F^2\}.$$

Using (12), we get $\mathbb{E}\{(z_t - \mathbb{E}\{z_t\})^2|\mathcal{F}_{t-1}\} \le 16\gamma_t^2\sigma_N^2$, which completes the proof. $\qquad\square$

### A.3 Proof of Lemma 3

Using Lemma 1, we can write the moment generating function of $\Psi_t$ as follows:

$$\mathbb{E}\{\exp(s\Psi_t)|\mathcal{F}_{t-1}\} \leq \mathbb{E}\Big\{\exp\Big(s\Psi_{t-1} + s\gamma_t^2 r^4 - sz_t\Big)\Big|\mathcal{F}_{t-1}\Big\} = \exp(s\Psi_{t-1} + s\gamma_t^2 r^4)\mathbb{E}\Big\{\exp\Big(-sz_t\Big)\Big|\mathcal{F}_{t-1}\Big\}$$

$$= \exp(s\Psi_{t-1} + s\gamma_t^2 r^4 - s\mathbb{E}\{z_t|\mathcal{F}_{t-1}\})\mathbb{E}\Big\{\exp\Big(-s(z_t - \mathbb{E}\{z_t\})\Big)\Big|\mathcal{F}_{t-1}\Big\}. \tag{37}$$

We can bound this using Bennett's inequality (Proposition 2 in Appendix D), which requires the variance and range of the random variable $z_t$. We have already computed the variance of $z_t$ in Lemma 2. Next we compute the boundedness of $(z_t - \mathbb{E}\{z_t\})$ as follows:

$$\Big|z_t - \mathbb{E}\{z_t\}\Big| \leq 2|z_t| \leq 2\gamma_t\|\mathbf{x}_{i,t}\|_2^2 \leq 2\gamma_t r^2 =: h. \tag{38}$$

Here, the last inequality is due to Assumption [**A1**]. Using parameters $\sigma_N^2$ and $h$ with Bennett's inequality, we get

$$\mathbb{E}\{\exp(s\Psi_t)|\mathcal{F}_{t-1}\} \leq \exp\Big(s\Psi_{t-1} - s\mathbb{E}\{z_t|\mathcal{F}_{t-1}\} + s\gamma_t^2 r^4 + s^2\gamma_t^2\sigma_N^2\Big(\frac{e^{sh}-1-sh}{(sh)^2}\Big)\Big). \tag{39}$$

For $L \geq L_1 + L_2$, where $L_1$ and $L_2$ are given by (13), we show in Proposition 3 in Appendix D that $(\frac{e^{sh}-1-sh}{(sh)^2}) \leq 1$ for $s \in \mathbb{S}$. This implies

$$\mathbb{E}\{\exp(s\Psi_t)|\mathcal{F}_{t-1}\} \leq \exp\Big(s\Psi_{t-1} - s\mathbb{E}\{z_t|\mathcal{F}_{t-1}\} + s\gamma_t^2 r^4 + s^2\gamma_t^2\sigma_N^2\Big),$$

which completes the proof of the lemma. $\qquad\square$

## Appendix B  Proofs of Lemmas for the Intermediate Epoch

### B.1 Proof of Lemma 4

Using Lemma 3, we have

$$\mathbb{E}\{e^{s\Psi_t}|\mathcal{F}_{t-1}\} \leq \exp\Big(s\Big(\Psi_{t-1} + \gamma_t^2 r^4 - \mathbb{E}\{z_t|\mathcal{F}_{t-1}\} + s\gamma_t^2\sigma_N^2\Big)\Big)$$

$$\overset{(a)}{\leq} \exp\Big(s\Big(\Psi_{t-1} - 2\gamma_t\Big(\lambda_1 - \lambda_2\Big)\Psi_{t-1}\Big(1 - \Psi_{t-1}\Big) + \gamma_t^2 r^4 + s\gamma_t^2\sigma_N^2\Big)\Big)$$

$$\overset{(b)}{\leq} \exp\Big(s\Big(\Psi_{t-1} - \frac{c_0\Psi_{t-1}\Big(1 - \Psi_{t-1}\Big)}{t+L} + \frac{c^2 r^4}{(t+L)^2} + \frac{sc^2\sigma_N^2}{(t+L)^2}\Big)\Big). \tag{40}$$

Here, $(a)$ is due to (Balsubramani et al., 2013, Lemma A.3) and $(b)$ is by substituting $\gamma_t = c/(t+L) = c_0/2(\lambda_1 - \lambda_2)(t+L)$. Finally, for $\omega \in \Omega_t'$ we have $\Psi_{t-1}(\omega) \leq 1 - \epsilon_j$. Now taking expectation over $\Omega_t'$, we get the desired result. $\qquad\square$

### B.2 Proof of Lemma 5

Define $\alpha_t := 1 - \frac{c_0\epsilon_j}{t+L}$ and $\zeta_t(s) := \frac{sc^2 r^4}{(t+L)^2} + \frac{s^2 c^2\sigma_N^2}{(t+L)^2}$. Substituting $\alpha_t$ and $\zeta_t(s)$ in Lemma 4, we get

$$\mathbb{E}_t\{e^{s\Psi_t}\} \leq \mathbb{E}_t\{e^{s\alpha_t\Psi_{t-1}}\}\exp\Big(\zeta_t(s)\Big) \leq \mathbb{E}_{t-1}\{e^{s\alpha_t\Psi_{t-1}}\}\exp\Big(\zeta_t(s)\Big). \tag{41}$$

Note that the second inequality in (41) is due to (Balsubramani et al., 2013, Lemma 2.8). Applying this procedure repeatedly yields

$$\mathbb{E}_t\{e^{s\Psi_t}\} \leq \mathbb{E}_{t_j+1}\{\exp\left(s\Psi_{t_j}\alpha_t\dots\alpha_{t_j+1}\right)\}\exp\left(\zeta_t(s)\right)\dots\exp\left(\zeta_{t_j+1}\left(s\alpha_t\dots\alpha_{t_j+1}\right)\right)$$
$$\leq \mathbb{E}_{t_j+1}\{\exp\left(s\Psi_{t_j}\alpha_t\dots\alpha_{t_j+1}\right)\}\exp\left(\zeta_t(s)\right)\dots\exp\left(\zeta_{t_j+1}(s)\right).$$

Substituting values of $\alpha_t$ and $\zeta_t(s)$ in the above, we get

$$\mathbb{E}_t\{e^{s\Psi_t}\} \leq \mathbb{E}_{t_j+1}\Big\{\exp\Big(s\Psi_{t_j}\Big(1 - \frac{c_0\epsilon_j}{t+L}\Big)\dots\Big(1 - \frac{c_0\epsilon_j}{t_j+L+1}\Big)\Big)\Big\}$$
$$\exp\Big(\big(sc^2r^4 + s^2c^2\sigma_N^2\big)\Big(\frac{1}{(t+L)^2} + \dots + \frac{1}{(t_j+L+1)^2}\Big)\Big)$$
$$\leq \exp\Big(s(1-\epsilon_j)\exp\Big(-c_0\epsilon_j\Big(\frac{1}{t+L} + \dots + \frac{1}{t_j+L+1}\Big)\Big)\Big)$$
$$\exp\Big(\big(sc^2r^4 + s^2c^2\sigma_N^2\big)\Big(\frac{1}{(t+L)^2} + \dots + \frac{1}{(t_j+L+1)^2}\Big)\Big). \tag{42}$$

Here, the last inequality is true because $\Psi_{t_j}(\omega) \leq 1 - \epsilon_j$ for $\omega \in \Omega'_{t_j+1}$ and $1 - x \leq e^{-x}$ for $x \leq 1$. Next we bound the summations in (42) as follows:

$$\frac{1}{t+L} + \dots + \frac{1}{t_j+L+1} \geq \int_{t_j+1}^{t+1} \frac{dx}{x+L} = \ln\frac{t+L+1}{t_j+L+1},$$

$$\frac{1}{(t+L)^2} + \dots + \frac{1}{(t_j+L+1)^2} \leq \int_{t_j}^{t} \frac{dx}{(x+L)^2} = \frac{1}{t_j+L} - \frac{1}{t+L}.$$

Substituting these bounds in (42), we get the desired result. $\qquad\square$

### B.3  Proof of Lemma 6

This lemma uses Lemma 5 and deals with a specific value of $t = t_{j+1}$. For $t = t_{j+1}$, (29) gives

$$\mathbb{E}_{t_j+1}\{e^{s\Psi_{t_j+1}}\} \leq \exp\Big(s(1-\epsilon_j)\Big(\frac{t_j+L+1}{t_{j+1}+L+1}\Big)^{c_0\epsilon_j} + \big(sc^2r^4 + s^2c^2\sigma_N^2\big)\Big(\frac{1}{t_j+L} - \frac{1}{t_{j+1}+L}\Big)\Big). \tag{43}$$

Using conditions [**C1**] and [**C2**] and the fact that $e^{-2x} \leq 1 - x$ for $0 \leq x \leq 3/4$, we get

$$(1-\epsilon_j)\Big(\frac{t_j+L+1}{t_{j+1}+L+1}\Big)^{c_0\epsilon_j} \leq e^{-\epsilon_j}(e^{-5/c_0})^{c_0\epsilon_j} = e^{-6\epsilon_j} \leq 1 - 3\epsilon_j \leq 1 - \epsilon_{j+1} - \epsilon_j.$$

Substituting this in (43), we obtain the desired result. $\qquad\square$

### B.4  Proof of Lemma 7

Constructing a supermartingale sequence $M_t$ in the same way as we did in Theorem 3 for $s \in \mathbb{S}$ and applying Doob's martingale inequality, we get

$$\mathbb{P}_{t_j}\Big(\sup_{t\geq t_j}\Psi_t \geq 1 - \epsilon_j\Big) \leq \mathbb{P}_{t_j}\Big(\sup_{t\geq t_j}M_t \geq e^{s(1-\epsilon_j)}\Big) \leq \frac{\mathbb{E}\{M_{t_j}\}}{e^{s(1-\epsilon_j)}}$$
$$= \frac{\mathbb{E}\{\exp\left(s\Psi_{t_j} + s\tau_{t_j}\right)\}}{e^{s(1-\epsilon_j)}} = \frac{\mathbb{E}\{\exp\left(s\Psi_{t_j}\right)\}\exp\left(s\tau_{t_j}\right)}{e^{s(1-\epsilon_j)}}.$$

Using Lemma 6 then results in

$$\mathbb{P}_{t_j}\Big(\sup_{t\geq t_j}\Psi_t \geq 1 - \epsilon_j\Big) \leq \frac{1}{e^{s(1-\epsilon_j)}}\exp\Big(s(1-\epsilon_j) - s\epsilon_{j-1} + \big(sc^2r^4 + s^2c^2\sigma_N^2\big)\Big(\frac{1}{t_{j-1}+L} - \frac{1}{t_j+L}\Big) + s\tau_{t_j}\Big).$$

Substituting a bound on $\tau_{t_j}$ from Theorem 3 (see, e.g., the discussion around (28)), we get

$$\mathbb{P}_{t_j}\Big(\sup_{t \geq t_j} \Psi_t \geq 1 - \epsilon_j\Big) \leq \exp\Big(-s\epsilon_{j-1} + \Big(sc^2r^4 + s^2c^2\sigma_N^2\Big)\Big(\frac{1}{t_{j-1}+L} - \frac{1}{t_j+L}\Big) + s\Big(c^2r^4 + sc^2\sigma_N^2\Big)\frac{1}{t_j+L}\Big)$$

$$= \exp\Big(-s\epsilon_{j-1} + s\Big(c^2r^4 + sc^2\sigma_N^2\Big)\frac{1}{t_{j-1}+L}\Big).$$

Substituting $s = (2/\epsilon_0)\ln(4/\delta)$ and using the lower bound on $L$, we get (see Proposition 5 in Appendix D for formal verification)

$$\mathbb{P}_{t_j}\Big(\sup_{t \geq t_j} \Psi_t \geq 1 - \epsilon_j\Big) \leq \exp\Big(-\frac{s\epsilon_{j-1}}{2}\Big) = \Big(\frac{\delta}{4}\Big)^{\epsilon_{j-1}/\epsilon_0} \leq \frac{\delta}{2^{j+1}}.$$

Summing over $j$ completes the proof of the lemma. $\qquad\square$

## Appendix C    Proofs for the Final Epoch

*Proof of Lemma 8.* From Lemma 1, Part $(i)$, we have

$$\Psi_t \leq \Psi_{t-1} + 4\gamma_t^2\Big(\Big\|\frac{1}{N}\sum_{i=1}^N \mathbf{A}_{i,t} - \mathbf{\Sigma}\Big\|_F^2 + \lambda_1^2\Psi_{t-1}\Big) - z_t.$$

Taking expectation conditioned on $\mathcal{F}_{t-1}$, we get

$$\mathbb{E}\{\Psi_t|\mathcal{F}_{t-1}\} \leq \Psi_{t-1}(1 + \gamma_t^2\lambda_1^2) + 4\gamma_t^2\sigma_N^2 - \mathbb{E}\{z_t|\mathcal{F}_{t-1}\},$$

where the second term is due to Lemma 9. Now using upper bound on $-\mathbb{E}\{z_t|\mathcal{F}_{t-1}\}$ from (Balsubramani et al., 2013, Lemma A.4), we get the following:

$$\mathbb{E}\{\Psi_t|\mathcal{F}_{t-1}\} \leq \Psi_{t-1}(1 + \gamma_t^2\lambda_1^2) + 4\gamma_t^2\sigma_N^2 - 2\gamma_t(\lambda_1 - \lambda_2)\Psi_{t-1}(1 - \Psi_{t-1})$$

$$= \Psi_{t-1}\Big(1 + \gamma_t^2\lambda_1^2 - 2\gamma_t(\lambda_1 - \lambda_2)(1 - \Psi_{t-1})\Big) + 4\gamma_t^2\sigma_N^2.$$

Finally, taking expectation over $\Omega_t'$, substituting $\gamma_t = c_0/(2(t+L)(\lambda_1 - \lambda_2))$, and using the facts that $\Omega_t'$ is $\mathcal{F}_{t-1}$-measurable and for $t > t_J$, $\Psi_{t-1} \leq 1/2$ and we lie in sample space $\Omega_t'$ with probability greater than $1 - \delta$ (Theorem 3), we obtain

$$\mathbb{E}_t\{\Psi_t\} \leq \mathbb{E}_t\Big\{\Psi_{t-1}\Big(1 + \frac{c_0^2\lambda_1^2}{2(t+L)^2(\lambda_1 - \lambda_2)^2} - \frac{c_0}{2(t+L)}\Big)\Big\} + \frac{4c^2\sigma_N^2}{(t+L)^2}$$

$$= \Big(1 + \frac{c_0^2\lambda_1^2}{2(t+L)^2(\lambda_1 - \lambda_2)^2} - \frac{c_0}{2(t+L)}\Big)\mathbb{E}_t\{\Psi_{t-1}\} + \frac{4c^2\sigma_N^2}{(t+L)^2}$$

$$\leq \Big(1 + \frac{c_0^2\lambda_1^2}{2(t+L)^2(\lambda_1 - \lambda_2)^2} - \frac{c_0}{2(t+L)}\Big)\mathbb{E}_{t-1}\{\Psi_{t-1}\} + \frac{4c^2\sigma_N^2}{(t+L)^2}.$$

This completes the proof of the lemma. $\qquad\square$

**Proposition 1.** *Let $a_1, b > 0$ and $a_2 > 1$ be some constants. Consider a nonnegative sequence $(u_t : t > t_J)$ that satisfies*

$$u_t \leq \Big(1 + \frac{a_1}{(t+L)^2} - \frac{a_2}{t+L}\Big)u_{t-1} + \frac{b}{(t+L)^2}.$$

*Then we have:*

$$u_t \leq \Big(\frac{L+1}{t+L+1}\Big)^{a_2}\exp\Big(\frac{a_1}{L+1}\Big)u_0 + \frac{1}{(t+L+1)}\exp\Big(\frac{a_1}{L+1}\Big)\Big(\frac{L+2}{L+1}\Big)^2\frac{b}{a_2-1}.$$

*Proof.* Recursive application of the bound on $u_t$ gives:

$$u_t \leq \left( \prod_{i=t_J+1}^{t} \left(1 + \frac{a_1}{(i+L)^2} - \frac{a_2}{i+L}\right)\right) u_{t_0} + \sum_{i=t_J+1}^{t} \frac{b}{(i+L)^2}\left( \prod_{j=i+1}^{t} \left(1 + \frac{a_1}{(j+L)^2} - \frac{a_2}{j+L}\right)\right). \quad (44)$$

Using (Balsubramani et al., 2013, Lemma D.1) we can bound the product terms as

$$\prod_{j=i+1}^{t} \left(1 + \frac{a_1}{(j+L)^2} - \frac{a_2}{j+L}\right) \leq \exp\left( \sum_{j=i}^{t} \frac{a_1}{(j+L)^2} - \sum_{j=i}^{t} \frac{a_2}{j+L} \right)$$

$$\leq \left(\frac{i+L+1}{t+L+1}\right)^{a_2} \exp\left( \sum_{j=i}^{t} \frac{a_1}{(j+L)^2} \right). \quad (45)$$

Next, we bound the last term here as

$$\exp\left( \sum_{j=i}^{t} \frac{a_1}{(j+L)^2} \right) \leq \exp\left( \int_{i+1}^{t+1} \frac{a_1}{(x+L)^2} dx \right) = \exp\left( \frac{a_1}{i+L+1} - \frac{a_1}{t+L+1} \right) \leq \exp\left( \frac{a_1}{i+L+1} \right).$$

Substituting this in (45) we get

$$\prod_{j=i+1}^{t} \left(1 + \frac{a_1}{(j+L)^2} - \frac{a_2}{j+L}\right) \leq \left(\frac{i+L+1}{t+L+1}\right)^{a_2} \exp\left( \frac{a_1}{i+L+1} \right).$$

Substituting this in (44) we get

$$u_t \leq \left(\frac{t_J+L+1}{t+L+1}\right)^{a_2} \exp\left( \frac{a_1}{t_J+L+1} \right) u_{t_J} + \sum_{i=t_J+1}^{t} \frac{b}{(i+L)^2} \left( \prod_{j=i+1}^{t} \left(1 + \frac{a_1}{(j+L)^2} - \frac{a_2}{j+L}\right)\right)$$

$$\leq \left(\frac{t_J+L+1}{t+L+1}\right)^{a_2} \exp\left( \frac{a_1}{t_J+L+1} \right) u_{t_J} + \sum_{i=t_J+1}^{t} \frac{b}{(i+L)^2} \left(\frac{i+L+1}{t+L+1}\right)^{a_2} \exp\left( \frac{a_1}{i+L+1} \right)$$

$$\leq \left(\frac{t_J+L+1}{t+L+1}\right)^{a_2} \exp\left( \frac{a_1}{t_J+L+1} \right) u_{t_J} + \exp\left( \frac{a_1}{t_J+L+1} \right) \frac{b}{(t+L+1)^{a_2}} \sum_{i=1}^{t} \frac{(i+L+1)^{a_2}}{(i+L)^2}$$

$$\leq \left(\frac{t_J+L+1}{t+L+1}\right)^{a_2} \exp\left( \frac{a_1}{t_J+L+1} \right) u_{t_J} + \exp\left( \frac{a_1}{t_J+L+1} \right) \frac{b}{(t+L+1)^{a_2}} \left(\frac{L+2}{L+1}\right)^2 \sum_{i=1}^{t} (i+L+1)^{a_2-2}.$$

Again applying (Balsubramani et al., 2013, Lemma D.1), we get the final result as follows

$$u_t \leq \left(\frac{t_J+L+1}{t+L+1}\right)^{a_2} \exp\left( \frac{a_1}{t_J+L+1} \right) u_{t_J} + \exp\left( \frac{a_1}{t_J+L+1} \right) \frac{b}{(t+L+1)^{a_2}} \left(\frac{L+2}{L+1}\right)^2 \frac{(t+L+1)^{a_2-1}}{a_2-1}$$

$$= \left(\frac{t_J+L+1}{t+L+1}\right)^{a_2} \exp\left( \frac{a_1}{t_J+L+1} \right) u_{t_J} + \frac{1}{(t+L+1)} \exp\left( \frac{a_1}{t_J+L+1} \right) \left(\frac{L+2}{L+1}\right)^2 \frac{b}{a_2-1}.$$

This completes the proof of the proposition. $\qquad\square$

## Appendix D  Other Auxiliary Results

**Proposition 2** (Bennett's Inequality (Boucheron et al., 2013))**.** *Consider a zero-mean, bounded random variable $X_i \in \mathbb{R}$ (i.e., $|X_i| \leq h$ almost surely) with variance $\sigma_i^2$. Then for any $s \in \mathbb{R}$, we have*

$$\mathbb{E}\{e^{sX_i}\} \leq \exp\left( \sigma_i^2 s^2 \left(\frac{e^{sh}-1-sh}{(sh)^2}\right) \right).$$

**Proposition 3.** *Let $h := 2\gamma_t r^2$ and $s \in \{d/4\epsilon, (2/\epsilon_0) \ln(4/\delta)\}$. It then follows that $\frac{e^{sh}-1-sh}{(sh)^2} \leq 1$.*

*Proof.* It is straightforward to see that $\frac{e^{sh}-1-sh}{(sh)^2} \leq 1$ as long as $sh \leq 7/4$. Therefore, in order to prove this proposition, it suffices to show that the lower bound on $L$ implies $sh \leq 7/4$ for $s \in \{d/4\epsilon, (2/\epsilon_0) \ln(4/\delta)\}$. We establish this claim as two separate cases for the two values of $s$.

*Case I:* For $s = d/4\epsilon$, substituting the value of $h$ gives us

$$sh = \frac{d\gamma_t r^2}{2\epsilon} = \frac{dcr^2}{2(t+L)\epsilon} \leq \frac{dcr^2}{2L\epsilon} \leq \frac{dcr^2}{2\epsilon L_1} \leq \frac{dcr^2}{2\epsilon} \frac{\epsilon}{8dr^4 \max(1,c^2) \ln(4/\delta)} \leq \frac{1}{16\ln(4/\delta)} \leq \frac{7}{4}.$$

*Case II:* For $s = (2/\epsilon_0) \ln(4/\delta)$, we obtain

$$sh = \frac{2\ln(4/\delta)cr^2}{\epsilon_0(t+L)} \leq \frac{2\ln(4/\delta)cr^2}{\epsilon_0 L_1} \leq \frac{2\ln(4/\delta)cr^2}{\epsilon_0} \frac{\epsilon_0}{8r^4 \max(1,c^2) \ln\frac{4}{\delta}} \leq \frac{1}{4} \leq \frac{7}{4}.$$

This completes the proof of the proposition. $\qquad\square$

**Proposition 4.** *Assuming $L \geq \frac{8dr^4 \max(1,c^2)}{\epsilon} \ln\frac{4}{\delta} + \frac{8d^2\sigma_N^2 \max(1,c^2)}{\epsilon^2} \ln\frac{4}{\delta}$ and the parameter $s = d/4\epsilon$, we have $\frac{c^2}{L}\left(r^4 + s\sigma_N^2\right) \leq \frac{\epsilon}{d}$.*

*Proof.* We prove this by proving the following two statements:

$$\frac{c^2 r^4}{L} \leq \frac{c^2 r^4}{L_1} \leq \frac{\epsilon}{2d} \quad \text{and} \quad \frac{sc^2\sigma_N^2}{L} \leq \frac{sc^2\sigma_N^2}{L_2} \leq \frac{\epsilon}{2d}.$$

We start by proving the first statement: $\frac{c^2 r^4}{L_1} \leq c^2 r^4 \frac{\epsilon}{8dr^4 \max(1,c^2) \ln\frac{4}{\delta}} \leq \frac{\epsilon}{2d}$. Next, we prove the second statement as follows: $\frac{c^2 s\sigma_N^2}{L_2} \leq \frac{c^2 d\sigma_N^2}{4\epsilon} \frac{\epsilon^2}{8d^2\sigma_N^2 \max(1,c^2) \ln\frac{4}{\delta}} \leq \frac{\epsilon}{2d}$. This completes the proof. $\qquad\square$

**Proposition 5.** *For $L \geq \frac{8r^4 \max(1,c^2)}{\epsilon_0} \ln\frac{4}{\delta} + \frac{8\sigma_N^2 \max(1,c^2)}{\epsilon_0^2} \ln\frac{4}{\delta}$, we have*

(i) $\frac{c^2 r^4}{(t_{j-1}+L)} \leq \frac{\epsilon_0}{4}$, *and*

(ii) $\frac{2c^2\sigma_N^2}{\epsilon_0(t_{j-1}+L)} \ln\frac{4}{\delta} \leq \frac{\epsilon_0}{4}$.

*Proof.* We begin by noting that

$$\frac{c^2 r^4}{(t_{j-1}+L)} \leq \frac{2c^2 r^4}{L} \leq \frac{2c^2 r^4}{L_1} \leq 2c^2 r^4 \frac{\epsilon_0}{8r^4 \max(1,c^2) \ln\frac{4}{\delta}} \leq \frac{\epsilon_0}{4}.$$

Next we prove the second statement as follows:

$$\frac{2c^2\sigma_N^2}{\epsilon_0(t_{j-1}+L)} \ln\frac{4}{\delta} \leq \frac{2c^2\sigma_B^2}{\epsilon_0 L} \ln\frac{4}{\delta} \leq \frac{2c^2\sigma_N^2}{\epsilon_0 L_2} \ln\frac{4}{\delta} \leq \frac{2c^2\sigma_N^2}{\epsilon_0} \ln\frac{4}{\delta} \frac{\epsilon_0^2}{8\sigma_N^2 \max(1,c^2) \ln(4/\delta)} \leq \frac{\epsilon_0}{4}.$$

This completes the proof of the proposition. $\qquad\square$

**Corollary 3** (Restatement of the Main Result (Theorem 1): Convergence in Probability). *Fix any $\delta \in (0,1)$ and pick $c := c_0/2(\lambda_1 - \lambda_2)$ for any $c_0 > 2$. Next, define*

$$L_1 := \frac{64edr^4 \max(1,c^2)}{\delta^2} \ln\frac{4}{\delta}, \quad L_2 := \frac{512e^2d^2\sigma_N^2 \max(1,c^2)}{\delta^4} \ln\frac{4}{\delta},$$

*pick any* $L \geq L_1 + L_2$, *and choose the step-size sequence as* $\gamma_t := c/(L+t)$. *Then, as long as Assumptions* [**A1**] *and* [**A2**] *hold, we have for D-Krasulina that there exists a sequence* $(\Omega'_t)_{t \in \mathbb{Z}_+}$ *of nested sample spaces such that* $\mathbb{P}\left(\cap_{t>0} \Omega'_t\right) \geq 1 - \delta$ *and*

$$\mathbb{P}\left\{\Psi_t \geq C'_1\left(\frac{L+1}{t+L+1}\right)^{\frac{c_0}{2}} + C'_2\left(\frac{\sigma_N^2}{t+L+1}\right)\right\} \leq 2\delta, \tag{46}$$

*where* $C'_1$ *and* $C'_2$ *are constants defined as*

$$C'_1 := \frac{1}{2\delta}\left(\frac{4ed}{\delta^2}\right)^{\frac{5}{2\ln 2}} e^{2c^2\lambda_1^2/L} \quad \text{and} \quad C'_2 := \frac{8c^2 e^{(c_0 + 2c^2\lambda_1^2)/L}}{\delta(c_0 - 2)}.$$

*Proof.* The proof follows by first using Markov's inequality to bound the conditional probability

$$\mathbb{P}_t\left\{\Psi_t \geq C'_1\left(\frac{L+1}{t+L+1}\right)^{\frac{c_0}{2}} + C'_2\left(\frac{\sigma_N^2}{t+L+1}\right)\right\},$$

then using the bound on $\mathbb{E}_t\{\Psi_t\}$ in Theorem 1 to further bound the conditional probability by $\delta$, and finally removing the conditioning on the nested sample spaces by using the facts that $(i)$ $\mathbb{P}(A) \leq \mathbb{P}(A|B) + \mathbb{P}(B^c)$ for any two events $A$ and $B$, and $(ii)$ $\mathbb{P}\left(\cup_{t>0}\Omega'_t{}^c\right) \leq \delta$. $\qquad\square$

