# OpenReview forum: "Distributed Stochastic Algorithms for High-rate Streaming Principal Component Analysis"
_TMLR — Accepted by TMLR_

### Review · Reviewer_Z1FS · 2022-07-06

**Summary Of Contributions:**

The paper looks at computing the top singular vector from a set of iid vector samples arriving at a very high rate.  In particular, it is concerned with when the rate is so high that directly adapting the updates via Krasulina's (or Oja's) algorithm is not possible.  The main result of the paper is that this computation can be spread across distributed machines, and the updates averaged still achieve the optimal linear convergence rate.  Batch variants and when a limited amount of data must be dropped, even in the distributed setting, are also analyzed with similar results.

The main technical idea is revising existing results showing the convergence based on worst case bound on each update to one that achieves the same linear convergence rate using a variance bound.  This analysis can then be adapted to distributed and batch analysis where averaging estimates across sets decreases variance at the desired linear rate.

The paper's writing takes great care to discuss extensive related work, explain the computational model and when and why they are appropriate, and to provide intuition for how the analysis fits together and the main ideas.

The topic is well-motivated, as dealing with massive data is an important challenge and the streaming model considered is aligned with how such data is dealt with.  Moreover, estimating the top singular vector is an important challenge in data analysis.
Hence, this paper is appropriate for TMLR, and I recommend acceptance.


Given that, I have a couple critiques.

The experimental section does a fine job of illustrating how well the bounds and analysis of parameters in the data assumptions and algorithms align with empirical performance.  However, the data sets are significantly smaller than the sizes needed to motivate the sort of model studied.
Nevertheless, I think this is fine, since one needs to verify the accuracy, and running on such motivating sizes would require extensive computing cost.  Also the main claims are still well illustrated.  There are caveats that other unforeseen issues may occur when scaling to sizes discussed in the paper, such as node failure or larger network cost than expected.  However, I rate this concern as minor, and do not think the authors need to extend the empirical study in this way.

Second, the paper uses the term PCA, and finding the top principal component, to mean finding the vector with maximum variance.  However, it assumes the data has mean zero.  Typically PCA is more general, and does not require the data to have zero mean, and often the process is to center the data before finding the top singular vector.  As such, the problem the paper actually solves is finding the top singular vector, which does not require centering.
That said the paper does not hide this assumption, and this assumed equivalence between PCA and SVD (by assuming the data is centered) is not uncommon in this sort of analysis.  However, I would appreciate if the authors spent a bit more time discussing this distinction, and the challenges in addressing the full case where the data cannot be assumed to be centered.



**Broader Impact Concerns:**

This is analysis and efficient algorithm design for a very standard and generic technique.  There are no significant broader impact concerns.

**Requested Changes:**

Please discuss challenges associated with not assuming the data is centered.

**Strengths And Weaknesses:**

The paper's writing takes great care to discuss extensive related work, explain the computational model and when and why they are appropriate, and to provide intuition for how the analysis fits together and the main ideas.

The topic is well-motivated, as dealing with massive data is an important challenge and the streaming model considered is aligned with how such data is dealt with.  Moreover, estimating the top singular vector is an important challenge in data analysis.
Hence, this paper is appropriate for TMLR, and I recommend acceptance.


Given that, I have a couple critiques.

The experimental section does a fine job of illustrating how well the bounds and analysis of parameters in the data assumptions and algorithms align with empirical performance.  However, the data sets are significantly smaller than the sizes needed to motivate the sort of model studied.
Nevertheless, I think this is fine, since one needs to verify the accuracy, and running on such motivating sizes would require extensive computing cost.  Also the main claims are still well illustrated.  There are caveats that other unforeseen issues may occur when scaling to sizes discussed in the paper, such as node failure or larger network cost than expected.  However, I rate this concern as minor, and do not think the authors need to extend the empirical study in this way.

Second, the paper uses the term PCA, and finding the top principal component, to mean finding the vector with maximum variance.  However, it assumes the data has mean zero.  Typically PCA is more general, and does not require the data to have zero mean, and often the process is to center the data before finding the top singular vector.  As such, the problem the paper actually solves is finding the top singular vector, which does not require centering.
That said the paper does not hide this assumption, and this assumed equivalence between PCA and SVD (by assuming the data is centered) is not uncommon in this sort of analysis.  However, I would appreciate if the authors spent a bit more time discussing this distinction, and the challenges in addressing the full case where the data cannot be assumed to be centered.

---

> ### Author Response · Authors · 2022-09-04
> **Response to Reviewer Z1FS**
>
> We thank the reviewer for their careful reading of our manuscript and the valuable feedback. In the following we explain how we plan on addressing the feedback of the reviewer as well as the “requested changes” in the revised version of the manuscript.
>
> **Revision addressing the “Requested Changes”**
>
> A new subsection (Section 3.3: A Note on the Processing of Non-centered Data) will be added to the manuscript, which will provide a discussion of the distinction between PCA and SVD in the case of non-centered data, and one possible modification of Algorithm 1 and Algorithm 2 that can lead to their utilization for the PCA problem involving non-centered data.
>
> Briefly, one can maintain a (network-wide) running average of the non-centered data samples and then use it to center the data samples at each processor before applying Step 2 (resp., Step 4) in Algorithm 1 (resp., Algorithm 2). While such a modification requires an extension of the analysis presented in this manuscript, this can be accomplished in a manner similar to the analytical extension in “Convergence analysis of Oja’s iteration for solving online PCA with nonzero-mean samples (2021)” for the centralized Oja’s rule with non-centered data.
>
> **Revision addressing the general feedback**
>
> We agree with the reviewer that there could be implementation caveats when it comes to running the proposed methods within a large-scale system. In order to highlight any such caveats, we will make the following edits to Section 1.2 Section 6, and Section 7 of the revised manuscript:
> - Item 4 in Section 1.2 will be edited to: *“We provide numerical results involving both synthetic and real-world data to establish the usefulness of the proposed algorithms, validate our theoretical analysis, and understand the impact of the number of dropped samples per iteration of DM-Krasulina on the convergence rate. These results in particular corroborate our findings that increasing the mini-batch size improves the performance of DM-Krasulina up to a certain point, after which the convergence rate starts to decrease. Since the focus of this work is on systems theory, the numerical experiments reported in this paper do not focus on some of the large-scale systems implementation issues such as unexpected processor failures, high network coordination costs, etc. Such large-scale implementation issues, while relevant from a practical perspective, are beyond the scope of this paper and provide interesting research directions for future work.”*
> - The end of the second paragraph in Section 6 will be edited to: *“In the following, we report results of experiments on both synthetic and real-world data to highlight these points. Since the main purpose is to corroborate the scaling behaviors within the main results—and not to investigate additional systems-related issues concerned with large-scale implementations—the real-world datasets are chosen to facilitate their processing on low-cost compute machines.”*
> - The following sentence will be added at the end of Section 7: *“Finally, investigation of additional practical issues (e.g., processor failures, variable compute costs, and network coordination costs) concerning processing of data in large-scale systems provides another avenue for future research.”*

---

> > ### Comment · Reviewer_Z1FS · 2022-09-06
> > **plan**
> >
> > Thank you.  Your plan would sufficiently address my concerns.

---

> > > ### Author Response · Authors · 2022-09-17
> > > **Revision submitted**
> > >
> > > Thank you; we have submitted the revised manuscript as per our stated plan.

---

### Review · Reviewer_rnHX · 2022-07-10

**Summary Of Contributions:**

This paper studies the PCA problem with streaming data. It focuses on a setting where the streaming data arrives at a higher rate than the processing power of a single processor. Therefore it utilizes multiple processors that communicate via a server-worker structure or a decentralized general network structure to process the streaming data in parallel. Theoretical analysis of the Krasulina method-based algorithms in the distributed setting are presented, and they show the convergence rate of $O(1/T)$ when the number of samples observed is $T$.

**Requested Changes:**

The paper is in general OK but the authors should address the issues mentioned in "Weaknesses" and make corresponding revisions. There are some additional (but perhaps minor) issues:

1. In (10), the random variable $\Psi(w)$ is not defined.
2. In Lemma 8, what is the constant $c$?
3. It seems strange to impose a new definition in Definition 5 for the variance of distributed samples, while it is clear that the variances are related by $\sigma_N^2 = \sigma^2/N$.
4. At the end of p.17, the upper bound should be $M_{0}$ instead of $M_{t_0}$?

**Strengths And Weaknesses:**

**Strengths**: The proof seems to be correct and the paper has provided a good discussion for the proof technique used (which is a key contribution). Overall, the paper has the following strengths:

1. It provides a tighter convergence analysis than (Balsubramani et al., 2013) and show that the asymptotic convergence rate of the Krasulina's algorithm is $O(\sigma_N^2/t)$ rather than $O(r^2/t)$ as in (Balsubramani et al., 2013), where $t$ is the iteration number. As such, the result shows that the distributed Krasulina's algorithm is able to take advantage of parallel processing to speed up computation of PCA since the simple aggregation of samples allows one to reduce variance of the estimate.

2. It analyzed the effects of skipping data in the multi-batch Krasulina method and show that as long as the skipping data rate does not exceed $\mu = O(B)$, then the optimal asymptotic convergence rate of $O(1/T)$ can be maintained.

**Weaknesses**: There are some weaknesses with the current submission as listed below

1. The algorithms analyzed are standard and can be considered as the direct extension of classical Krasulina method to the distributed setting. Moreover, important aspects to distributed algorithms such as communication efficiency of the distributed algorithms has not been addressed. In this regard, the paper seems to be better positioned as presenting an improved analysis of Krasulina over (Balsubramani et al., 2013).

2. Compared to prior work (Jain et al. 2016) which has a variance dependent bound unlike (Balsubramani et al., 2013), one of the main claims in this paper is that it provides a high probability bound ($1-\delta$) to arbitrary $\delta$. However, the claim is slightly unfair:

- (a) as mentioned in (Jain et al. 2016), it is possible to run $O( log(1/\delta) )$ of their algorithms to obtain convergence w.p. at least $1-\delta$;

- (b) to ensure convergence for the Krasulina algorithm in this paper with probability at least $1-\delta$, it requires setting the step size parameter to be $L \asymp \ln(1/\delta)$ - in other words, to ensure convergence with an arbitrarily high probability, it is required to set an arbitrarily small step size.

- Moreover, it should be noted that the derived convergence results are **expected** convergence rate which holds with high probability, while (Jain et al. 2016) derived a high probability convergence rate.

---

> ### Author Response · Authors · 2022-09-11
> **Response to Reviewer rnHX**
>
> We thank the reviewer for their careful reading of the manuscript and the valuable suggestions for a revision. We clarify in the following the edits that will be made in the revised manuscript in response to the “requested changes” by the reviewer.
>
> **Revision addressing the “Requested Changes”**
>
> - *Positioning of the paper:* We agree with the reviewer that one of the major contributions of this work is in providing an improved analysis of Krasulina’s method over (Balsubramani et al., 2013), and the paper does not address some practical aspects of a large-scale system implementation such as processor failures, variable compute costs, and network coordination costs. (Note, however, that the “communication efficiency” is implicitly accounted for in the manuscript through the parameter $R_c$ in Section 2.2.1.) In order to emphasize / clarify these points, especially in relation to this work providing an improved analysis of Krasulina’s method over (Balsubramani et al., 2013), we will make edits to the Abstract, Section 1.2 (Our Contributions), Section 2.2.1 (Distributed Processing Over a Network of Processors), Section 4 (Convergence Analysis of D-Krasulina and DM-Krasulina), Section 5 (Proof of the Main Result), and Section 7 (Conclusion) of the revised manuscript.
> - *Comparison with (Jain et al., 2016):* The reviewer has raised three important points related to our discussion of (Jain et al., 2016) in the manuscript. The revised manuscript will address these points as follows:
>   - *Running (Jain et al., 2016) $O(\log(1/\delta)$ times to obtain $1-\delta$ probability of success:* This is indeed a correct observation, which was briefly discussed at the end of the first paragraph of Section 1.3 in the original manuscript, and we will clarify this point as well as its limitations further in Section 1.2 (Our Contributions) and Section 1.3 (Related Work) of the revised manuscript. Briefly, there are two main limitations of running an algorithm such as Oja’s rule and Krasulina’s multiple times, as required by (Jain et al., 2016), in streaming settings. First, since new data samples arrive continuously in a streaming setting, multiple runs of an algorithm in this case can only be achieved through multiple replicas of the processing system. Such a strategy, therefore, leads to a substantial increase in system costs. Second, the outcomes of the multiple runs need to be appropriately combined. In (Jain et al., 2016), it is suggested this be done by computing the geometric median of the multiple outcomes, which requires solving an additional optimization problem. This then adds to the computational and storage overhead for the PCA problem.
>   - *Step size comparison:* This is an important point of discussion, which will be clarified in Section 1.3 (Related Work) and Section 4.1 (Convergence of D-Krasulina (Algorithm 1)) of the revised manuscript. Briefly, the step size in both this work and (Jain et al., 2016) takes the form $O(\frac{1}{L+t})$ for some constant $L$, where the exact form of the constant $L$ differs in the two works. In terms of scaling with $t$, therefore, the step size in both these works is equivalent. By making the constant $L$ be a function of $\delta$ in this work, we are able to vary the probability of success $1-\delta$, with the caveat being that the initial step size $\gamma_1$ gets smaller as $\delta$ decreases. In contrast, a fixed $L$ in (Jain et al., 2016) can be thought of as one of the reasons the probability of success is fixed at $3/4$ in that work.
>   - *Nature of the high-probability results:* This is yet another important point, which we will clarify in Section 1.2 (Our Contributions) of the revised manuscript. Briefly, the difference between our results and that of (Jain et al., 2016) can be thought of as similar to the one between “convergence in mean” and “convergence in probability” while conditioned on a high-probability event. In particular, our results imply convergence in probability (i.e., similar to the ones in (Jain et al., 2016))  through the Markov inequality.
> - *Undefined $\Psi_l(\omega)$ in (10):* Here, $\omega$ denotes an outcome within the sample space $\Omega$ and $\Psi_l(\omega)$ is the (random) potential function after the $l$-th iteration of D-Krasulina / DM-Krasulina that is being explicitly written as a function of the outcomes $\omega$ in the sample space. We will make this clearer in Definition 4 of the revised manuscript.
> - *Constant $c$ in Lemma 8:* The constant $c$ should have been defined in the lemma as $c := \frac{c_0}{\lambda_1-\lambda_2}$. We will update this in the revised manuscript.
> - *Definition 5 for the variance of distributed samples:* We agree with the reviewer’s comment. The corresponding discussion will be moved out of the definition environment.
> - *Upper bound at the end of p. 17:* The reviewer is indeed correct and we thank them for pointing out this typo; it will be corrected in the revised manuscript.

---

> > ### Comment · Reviewer_rnHX · 2022-09-13
> > **Re: Response**
> >
> > Thank you for the response. The revision plan seems reasonable and I'm looking forward to the revised manuscript. In the meanwhile, I suggest including an exact calculation for the high probability convergence result mentioned in *"Nature of the high-probability results"* to enable a fair comparison to (Jain et al., 2016).

---

> > > ### Author Response · Authors · 2022-09-17
> > > **Revision submitted**
> > >
> > > Thank you; we have submitted the revised manuscript. The requested calculations have been provided as part of Corollary 3 in Appendix D.

---

### Review · Reviewer_Ly9o · 2022-09-01

**Summary Of Contributions:**

This paper studies a decentralized version of Krasulina's method for streaming PCA (i.e., eigenvector estimation), with and without mini-batching. The paper provides conditions under which D-Krasulina achieves a linear speedup in the number of distributed workers. In the mini-batch setting, conditions are also provided where a linear speedup is achieved. For the extreme case where, even with mini-batching, the system of N workers is not able to process samples  faster than they are arriving, further analysis quantifies the rate of convergence when some samples are dropped, with an assumption that all samples are iid. In this case, DM-Krasulina still achieves near-optimal convergence rates as long as the fraction of dropped samples is not too large. These results are validated with numerical experiments.

**Broader Impact Concerns:**

None noted

**Requested Changes:**

## Requested changes
These are the main points I view as critical.
* Please clarify/justify the timing model used in the paper, given that $R_c$ depends on $N$ and that this is also used to determine a bound on the number of processors $N$.
* While the related work section covers most relevant related work, it would be good to also discuss the relationship to [Yun et al., "Fast and memory optimal low-rank matrix approximation," NeurIPS 2015.](https://proceedings.neurips.cc/paper/2015/hash/21be9a4bd4f81549a9d1d241981cec3c-Abstract.html)
* In the statement of Corollary 1, please clarify whether the event occurring with probability $1 - \delta$ involves the intersection over $0 < t \le T_B$, or all $t > 0$ (i.e., the probability $1-\delta$ is still only achieved asymptotically).
* Setting a schedule to satisfy the conditions of Theorem 4 requires knowing the gap $\lambda_1 - \lambda_2$, among other problem-dependent constants. Please add a discussion of how one might approach this in practice when applying the method.
* One typical characteristic of streaming data settings (in practice) is that the data distribution may not be iid; rather over time, there may be some distribution shift. It would be good to add some discussion about this iid assumption and ways it may be relaxed in future work.

## Additional questions
* Given the relationship of Krasulina's method to SGD, might one expect to achieve a superior method by incorporating some sort of momentum or acceleration term?

**Strengths And Weaknesses:**

## Strengths
* The paper is generally well-written and easy to follow
* The related work is fairly comprehensive, nicely framing the contribution of this paper
* Theoretical guarantees are rigorous and establish strong performance bounds for the proposed approach

## Weaknesses
* The system timing model considered in the paper has some limitations. The processing and communication times $R_p$ and $R_c$ implicitly depend on the problem dimension $d$. The communication time $R_c$ also depends very explicitly on $N$, the number of processors aggregating results. Although this is briefly mentioned in Sec 2.2.1 when $R_c$ is introduced, this brings into question the way that it is used to obtain conditions on $N$ relative to $R_s$, $R_p$, $R_c$, and $b$ throughout the paper.
* A few older relevant related works are missing (see below).
* Some aspects of how the main theoretical results are presented could be slightly improved/clarified (see below).
* Setting a schedule to satisfy the conditions of Theorem 4 requires knowing the gap $\lambda_1 - \lambda_2$, among other problem-dependent constants, which may not be possible in some applications.
* The introduction motivates the problem considered by discussing challenges with high-dimensional data, while the experiments only consider problems with dimension at most 784. The paper would be stronger if the method were demonstrated on much higher-dimensional datasets.
* The experimental section does not include comparisons to any other method from the literature.

---

> ### Author Response · Authors · 2022-09-12
> **Response to Reviewer Ly9o: Part I**
>
> We thank the reviewer for their careful reading of our manuscript and the helpful feedback / suggestions for a revision. In the following, we elaborate on the edits that we plan on making to the manuscript in response to the “requested changes” as well as the general feedback from the reviewer. Please note that this **response is being broken into two parts** because of the limit of 5000 characters in a comment.
>
> **Revision addressing the “Requested Changes”**
>
> - *Clarification of the timing model:* We will edit Section 2.2 and Section 2.2.1 to emphasize that the parameters $R_p$ and $R_c$ depend on the problem dimension $d$, while $R_c$ also depends on the number of processors $N$. We will also provide additional details and some toy examples to help the reader better interpret the bound on the number of processors $N$, given that the bound itself is a function of $N$. Briefly, the high-performance computing community has access to copious amounts of trace data for different parallel computing environments that allows one to relate the parameter $R_c$ for a given dimension $d$ to the number of processors $N$; see, e.g., “An Analysis of Traces from a Production MapReduce Cluster (2010)”. One such relationship could be, for example, that $R_c \propto 1/N^\kappa$ for some $\kappa \in (0,1]$. The final bound on the number of processors $N$ can then be obtained by plugging such a relationship into the provided bound.
> - *Relationship to the literature on low-rank matrix approximation:* We will add a paragraph to Section 1.3 (Related Work) after the paragraph titled “Decentralized PCA”. The new paragraph will have the title “Connections to low-rank matrix approximation” and it will discuss the relationship between the PCA and low-rank matrix approximation problems, and will also compare and contrast the streaming setting of our work to that of works “Fast and Memory Optimal Low-Rank Matrix Approximation (2015)” and “Streaming Low-Rank Matrix Approximation with an Application to Scientific Simulation (2019)”. Briefly, the setting in the 2015 paper corresponds to a large-scale but fixed matrix whose columns are presented to a low-rank approximation algorithm in a streaming manner. This is in contrast to the streaming setting of our work that is akin to having a matrix with infinitely many columns. In addition, the algorithm being studied in the 2015 paper requires computing the top principal component directions of a random submatrix of the larger matrix in a batch setting as its first step, which is again a departure from the streaming data setting of our work. Further, the mathematical model in the 2019 paper corresponds to a matrix that is given by the sum of a long sequence of low-rank and sparse matrices that are presented to a low-rank approximation algorithm in a streaming manner. This summation aspect of the data model in the 2019 paper does not coincide with the mathematical model of the streaming data samples in our work. Finally, and in stark contrast to our work, neither of these works are concerned with the interplay between the streaming data rate, data processing rate, and the number of interconnected processors.
> - *Clarification of Corollary 1:* The high probability event in Corollary 1 indeed involves the intersection over $0 < t \leq T_B$ and this will be clarified in the revision.
> - *Knowledge of the eigen gap $\lambda_1 - \lambda_2$ in Theorem 4:* Theorem 4 is just part of the analytical machinery that is used to prove the main results (Theorems 1 and 2) in the manuscript, and the schedule is not utilized within Algorithms 1 and 2. Nonetheless, and as the reviewer has correctly noted, the fact remains that the main results in this work rely upon choosing a step size of the form $\frac{c_0}{(\lambda_1-\lambda_2)(L+t)}$ for the stated convergence rates. But such dependence of the step size on the underlying set of eigenvalues for the optimal convergence rates in PCA is unavoidable and ubiquitous in related works, as briefly noted in the discussion after Theorem 1. We will expand on this discussion in the revised manuscript and will provide a simple yet highly effective strategy for setting the step size for the proposed methods in a practical setting. Briefly, step sizes of the form $c/t$ work well for these methods in practice and the parameter $c$ can be estimated by running multiple instances of the methods during a warm-up phase. Note that such an approach is akin to approximating several problem-related parameters using a single parameter $c$ and is the one that we followed for the numerical experiments in Section 6 (Numerical results) of the manuscript.
>
> ***--- response continued in Part II ---***

---

> > ### Author Response · Authors · 2022-09-12
> > **Response to Reviewer Ly9o: Part II**
> >
> > ***--- continuation of the response from Part I ---***
> >
> > - *Discussion for non-iid data:* We will add such a discussion to the revised manuscript in a new subsection (Section 3.3: A Note on the Processing of Non-centered and Non-iid Data). Briefly, while the theoretical guarantees in this manuscript have been provided under the assumption of iid data samples, the same methods will work in the non-iid data setting. This is because Krasulina's method effectively computes a new gradient-like quantity using the latest batch of data samples at each time $t$ and then updates the eigenvector estimate using this quantity. In particular, any shifts in the data distribution can be tracked by the proposed methods because of such an update rule. It is because of this reason that algorithms such as Oja's rule and Krasulina's method are also often employed for the problem of "subspace tracking" (see, e.g., Yang 1995, Chatterjee 2005, and Doukopoulos et al., 2008 cited in the original manuscript). Since providing a formal analysis of such tracking capabilities of the proposed methods for non-iid data is beyond the scope of this work, we leave it for future work.
> > - *Accelerating Krasulina’s method:* This is an interesting observation and we will address this as a remark in Section 4.1 (Convergence of D-Krasulina (Algorithm 1)) of the revised manuscript. Briefly, naively applying momentum to Oja’s rule or the power iteration, both of which are closely related to Krasulina’s method, is known to result in worst performance since this increases the effect of the noise variance within the iterates (see, e.g., “Accelerated Stochastic Power Iteration (2018)”). And while the noise variance can be controlled through variance reduction techniques, such techniques typically require multiple data passes and are therefore not suited for the setting in which data samples continuously stream into the system.
> >
> > **Revision addressing the general feedback**
> >
> > *Dimensionality of data in the experiments:* We agree with the reviewer that the experiments could have been run using datasets that are higher dimensional than 784. However, our main focus was understanding and highlighting the interplay between the streaming rate, network-wide mini-batch size, and number of processors. Accordingly, since the purpose of the experiments was to corroborate this interplay predicted by theory, we limited ourselves to real-world datasets that facilitated their processing on low-cost compute machines. We will emphasize this point further in Section 6 of the revised manuscript.
> >
> > *Comparison with other methods in the experimental section:* The sample complexity of both Oja’s rule and Krasulina’s method in the case of streaming data is known to be optimal (Balsubramani et al., 2013; Jain et al., 2016; Allen-Zhu & Li, 2017a), and the results derived in our work show how the same optimal sample complexity can be achieved in the case of distributed processing of fast streaming data using Krasulina’s method. To the best of our knowledge, and as discussed in Section 1.3 (Related Work) under the titles of “Distributed PCA and streaming data” and “Decentralized PCA”, none of the existing works guarantee such optimal rates in a non-centralized setting. In light of this, we hope the reviewer will agree with us that there is no clear basis for comparison with other works within the experimental section. We will however clarify this point further in Section 6 of the revised manuscript.

---

> > > ### Author Response · Authors · 2022-09-17
> > > **Revision submitted**
> > >
> > > Dear Reviewer Ly9o, please know that we have submitted a revision of our original manuscript as per the plan stated in the response letter. We look forward to further feedback from you. Thank you.

---

> > > > ### Comment · Reviewer_Ly9o · 2022-09-23
> > > > **Acknowledging responses and updates**
> > > >
> > > > Thank you for your responses to my review and for the updates you have made to the paper. These address my concerns

---

### Decision · Action_Editors · 2022-09-27

**Recommendation:** Accept as is

**Comment:**

The writing looks nice. The reviewers comments were all addressed, and I'm pleased to recommend Acceptance.

**Audience:**

A mix of theory (researchers interested in streaming PCA results, and some general math audience), as well as practice (people interested in running streaming PCA). In terms of practical impact, the paper isn't really about new algorithms (it's more about analysis), but it does offer some useful information for anyone who wants to implement streaming PCA in a distributed setting.

**Claims Support:**

Let's break this down into new categories:

1. **Correctness of theorems**.  Neither I nor the reviewers see any issues; that's not to say that every claim has been checked in detail, but there was nothing suspicious about the math.  There were some numerical experiments to make a few points, not to provide definitive evidence that this method is superior to competing methods. I think this is reasonable for this type of paper.
2. **Improvement of the theorem over prior results in literature** (for example, reviewer rnHX pointed out that in (Jain et al. '16), one can get a delta failure probability by running log(1/delta) independent copies of the algorithm, a standard technique). In this case, I think the authors successfully point out that these repeated runs make less sense in a stream setting, and furthermore, I might add that a direct high-probability analysis is clearly preferable.  There were a few other minor points the reviewers mentioned. Reviewer rnHX is satisfied with the revision, as are the other reviewers and myself.
3.  **Usefulness of streaming PCA as a topic**.  The authors do not directly add their own contributions here, but rely on a clear chain of literature that has studied the same problem. I find that this is substantial evidence.

**Main Claims:**

The first claim is an improved analysis of Krasulina’s method, specifically an analysis that takes into account variance and therefore improves when using more nodes, resulting in an improvement over (Balsubramani et al., 2013).

The second claim is that this analysis is relevant, and the problem is interesting.